# Buoyancy forcing: A key role for northern North Atlantic SST variability across multiple time scales

Bjørg Risebrobakken[1], Mari F. Jensen[2], Helene R. Langehaug[3], Tor Eldevik[4], Anne Britt Sandø[5], Camille Li[4], Andreas Born[2], Erin L. McClymont[6], Ulrich Salzmann[7], Stijn De Schepper[1]

[1]NORCE Norwegian Research Center, Bjerknes Centre for Climate Research, Bergen, Norway

[2]Department of Earth Sciences, University of Bergen and Bjerknes Centre for Climate Research, Bergen, Norway

[3]Nansen Environmental and Remote Sensing Center, Bjerknes Centre for Climate Research, Bergen, Norway

[4]Geophysical Institute, University of Bergen and Bjerknes Centre for Climate Research, Bergen, Norway

[5]Institute of Marine Research, Bjerknes Centre for Climate Research, Bergen, Norway

[6]Department of Geography, Durham University, Durham, UK

[7]Department of Geography and Environmental Sciences, Northumbria University, UK

*Correspondence to: Bjørg Risebrobakken (bjri@norceresearch.no)*

**Abstract:** Analyses of observational data (from year 1870 AD) show that Sea Surface Temperature (SST) anomalies along the pathway of Atlantic Water transport in the North Atlantic, the Norwegian Sea and the Iceland Sea are spatially coherent at multidecadal time scales. Spatially coherent SST anomalies are also observed over hundreds of thousands of years during parts of the Pliocene (5.23–5.03, 4.63–4.43 and 4.33–4.03 Ma). However, when investigating CMIP6 SSP126 future scenario runs (next century) and other Pliocene time intervals, three additional SST relations emerge: 1) The Norwegian Sea SST anomaly is dissimilar to the North Atlantic and the Iceland Sea SST anomalies (Pliocene; 4.93-4.73 and 3.93–3.63 Ma); 2) The Iceland Sea SST anomaly is dissimilar to the North Atlantic and the Norwegian Sea SST anomalies (Pliocene; 3.43–3.23 Ma); 3) The North Atlantic SST anomaly is dissimilar to the Norwegian and Iceland Seas SST anomalies (future trend). Hence, spatially noncoherent SST anomalies may occur in equilibrium climates (Pliocene) as well as in response to transient forcing (CMIP6 SSP 126 low-emission future scenario). Since buoyancy is a key forcing for inflow of Atlantic Water to the Norwegian Sea, we investigate the impacts of buoyancy forcing on spatial relations between SST anomalies seen in the North Atlantic, Norwegian and Iceland Seas. This is done by performing a range of idealized experiments using the Massachusetts Institute of Technology general circulation model (MITgcm). Through these idealized experiments we can reproduce three out of four of the documented SST anomaly relations: spatially coherent under weak to intermediate freshwater forcing over the Nordic Seas; the Iceland Sea dissimilar to the North Atlantic and the Norwegian Sea under weak atmospheric warming over the Nordic Seas; and the North Atlantic dissimilar to the Norwegian and Iceland Seas under strong atmospheric warming over the Nordic Seas. We suggest that the unexplained SST anomaly relation, when the Norwegian Sea is dissimilar to the North Atlantic and the Iceland Sea, may reflect a response to a weakened Norwegian Atlantic Current compensated by a strong Irminger Current, or an expanded East Greenland Current.

## 1 Introduction

The North Atlantic Current transports warm and saline Atlantic Water northward, through the subpolar North Atlantic and into the Norwegian Sea where the Norwegian Atlantic Current continues the transport towards the Arctic (Fig. 1). A smaller fraction of Atlantic Water also enters the Nordic Seas west of Iceland, through the North Iceland Irminger current.

While branches of the Norwegian Atlantic Current are deflected into the Arctic Ocean and the Barents Sea through the Fram Strait and Barents Sea Opening (Blindheim and Østerhus, 2005; Smedsrud et al., 2022), a large fraction of the Atlantic Water recirculates in the Fram Strait and joins the southward flowing deeper branch of the East Greenland Current (Bourke et al., 1988). Cold and fresh water branches off from the East Greenland Current, towards the Iceland Sea. While part of this water re-joins the East Greenland Current, some will continue eastward in the East Icelandic Current and into the Norwegian Sea (Macrander et al., 2014). Along its way through the Nordic Seas (i.e. the Norwegian, Greenland and the Iceland Seas) and the Arctic Ocean, the warm and saline Atlantic Water is gradually transformed as it loses heat and gains freshwater (Mauritzen, 1996).

Wind and buoyancy are the two key factors that impact the inflow of Atlantic Water to the Norwegian Sea. Wind forcing is important for the inflow of Atlantic Water across the Greenland Scotland Ridge at seasonal and interannual time scales (Bringedal et al., 2018). However, buoyancy forcing, changing seawater density due to heat (heating/cooling) and/or freshwater (evaporation/precipitation/runoff) fluxes and associated production of dense overflow water that must be compensated, is key at longer time scales (Furevik et al., 2007; Smedsrud et al., 2022; Talley et al., 2011). We will investigate SST anomaly relations in the North Atlantic, Norwegian and Iceland Seas region, at multidecadal and longer time scales, hence, timescales when buoyancy is considered most important. Therefore, our focus is on how northern North Atlantic SST anomalies are impacted by changes in buoyancy.

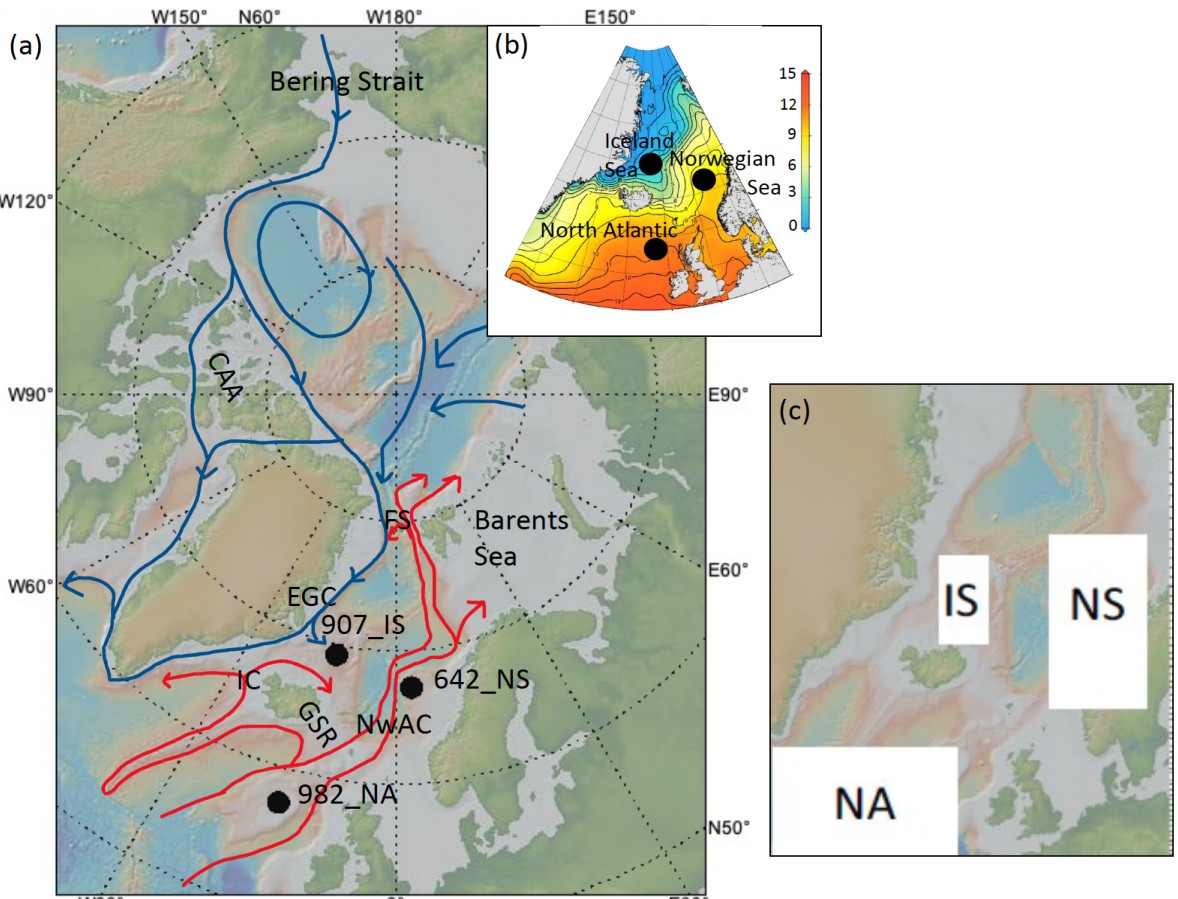

**Figure 1** (a) Map of the subpolar North Atlantic, Nordic Seas and Arctic Ocean with a schematic illustration of the main surface currents (currents carrying Atlantic Water are shown in red while currents carrying Polar Water are shown in blue). NwAC: Norwegian Atlantic Current. IC: Irminger Current. EGC: East Greenland Current. FS: Fram Strait. CAA: Canadian Arctic Archipelago. GSR: Greenland Scotland Ridge. 982_NA: Location of ODP Site 982 from the North Atlantic. 642_NS: Location of ODP Site 642 from the Norwegian Sea. 907_IS: Location of ODP Site 907 from the Iceland Sea. (b) The objectively analysed mean regional ocean climatology

for 1995 to 2004, represented by annual temperature (°C) at the surface (10 m) using a quarter-degree grid (Seidov et al., 2013; Seidov et al., 2018). Contour interval is 1°C, ranging from 13°C to -1°C. (c) show the domains over which the CMIP6 data are analysed (NA: North Atlantic; NS: Norwegian Sea and IS: Iceland Sea). The same domains are used throughout the paper for conceptual representation of the CMIP6 and Pliocene results (Fig. 6, 7, 8 and 10). The figure base is made with GeoMapApp (www.geomapapp.org) / CC BY / CC BY (Ryan et al., 2009)).

Heat is continuously transported northwards from the North Atlantic towards the Arctic. Due to the continuous northward transport, it is expected that a warm North Atlantic will entail warm SSTs both in the Norwegian and Iceland Seas. Alternatively, if it is cold in the North Atlantic, the Norwegian and Iceland Seas are also expected to be cold. It takes 3-4 years for Sea Surface Temperature (SST)/heat anomalies to travel from the North Atlantic through the Norwegian Sea (Holliday et al., 2008). Therefore, spatially incoherent SST anomalies between the seas may exist at interannual-to-decadal time scales. This feature has been documented in observations and Earth System Models (Årthun and Eldevik, 2016; Årthun et al., 2017). Beyond decadal time scales, however, this propagation-driven lag should in theory no longer be of importance, and the default expectation is of a spatially coherent SST relationship between the North Atlantic, the Norwegian Sea and the Iceland Sea, in line with observations (Årthun and Eldevik, 2016; Årthun et al., 2017).

Contrasting the expectation of spatially coherent SST anomalies, spatially noncoherency between SST anomalies of the North Atlantic and Norwegian Sea emerge at multidecadal time scales in the strongly forced Coupled Model Intercomparison Project (CMIP)5 RCP 8.5 scenario runs for future climate change (Alexander et al., 2018; Nummelin et al., 2017). These CMIP5 studies (Alexander et al., 2018; Nummelin et al., 2017; Keil et al., 2020) suggest that the expectation of spatially coherent SST anomalies between the North Atlantic, Norwegian and Iceland Seas is not valid under strongly forced, high emission scenarios, and for the associated transient changes expected to take place within the next century.

We question whether the spatially noncoherent SST response seen in the CMIP5 studies is restricted to the high emission scenario, or if spatially noncoherent SST responses may also occur under less extreme atmospheric $CO_2$ concentrations. If it turns out that spatially noncoherent SST anomalies are seen also under less extreme atmospheric $CO_2$ concentrations, the validity of our expectation of spatial coherence may be limited to the observational period. To investigate whether spatially noncoherent SST responses may also occur under less extreme atmospheric $CO_2$ concentrations, we will use CMIP6 Shared Socioeconomic Pathways (SSP)126 experiments and Pliocene alkenone SST reconstruction from the North Atlantic, Norwegian and Iceland Seas. During the Pliocene (5.3 to 2.6 Ma), atmospheric $CO_2$ concentrations were close to 400 ppm in average (De La Vega et al., 2020; Bartoli et al., 2011), comparable to the present (ca. 410 ppm) and the future low emission scenarios such as ssp126 (445 ppm by the end of the century) (Meinshausen et al., 2020; IPCC, 2021). However, it is important to keep in mind is that the Pliocene climate was not forced by an abrupt $CO_2$ increase, as the future scenarios are. Rather, relatively high $CO_2$ values existed for millions of years through the Pliocene. The SST anomaly relations in the North Atlantic, Norwegian and Iceland Seas during the Pliocene may therefore be seen as various equilibrium responses to an atmospheric $CO_2$ content comparable to todays, in contrast to the transient responses given by the CMIP model scenarios. Analysing both the SSP126 experiments and the Pliocene reconstructions therefore allow us to explore potential differences in equilibrium versus transient SST responses to a ca. 400 ppm $CO_2$ forcing.

Furthermore, we address why spatially noncoherent SST relations may emerge and exist across different climate states, time scales and atmospheric $CO_2$ forcing scenarios. As mentioned above, at multidecadal and longer timescales the inflow of Atlantic Water to the Norwegian Sea over the Greenland Scotland Ridge is tightly connected to the density difference

between the two basins (Furevik et al., 2007; Smedsrud et al., 2022; Talley et al., 2011), while wind forcing may dominate at shorter time scales (Bringedal et al., 2018). We hypothesise that, for the time scales of interest here, changing the buoyancy may be enough to push the system from spatially coherent to spatially noncoherent SST anomalies. To test this hypothesis, we perform a range of idealized sensitivity experiments using the Massachusetts Institute of Technology general circulation model (MITgcm) to investigate impacts of changes in buoyancy forcing on the SSTs in the North Atlantic, Norwegian Sea, and Iceland Sea. These idealized experiments provide potential physical explanations for the different spatial SST relations see in the investigated region.

By analysing variability across a wide range of time scales, we provide new perspectives on which spatio-temporal structure of SST patterns may exist under different background climate states and $CO_2$ forcing regimes.

## 2. Data and method

To confirm the basis for our expectation of spatial coherency in SST anomalies, we use data from the Met Office Hadley Centre (version 1.1) (Section 2.1). The future responses to SSP126 are investigated in three CMIP6 models (Section 2.2), while the Pliocene equilibrium response to a ca. 400 ppm $CO_2$ forcing is documented from three ODP sites representing the North Atlantic, Norwegian and Iceland Seas (Section 2.3). The MITgcm experiments testing the impacts of changes in buoyancy forcing are introduced in Section 2.4.

### 2.1 Observation-based data: HadlSST

The expectation of spatially coherent SST anomalies between the North Atlantic and the Norwegian and Iceland Seas is rooted in the observational period and investigations of SST anomalies at specific stations along the pathway of the North Atlantic Current (Årthun and Eldevik, 2016; Årthun et al., 2017). A comparable analysis of the observational record is done here to confirm if the expectation of spatial coherence holds when looking at averages over larger domains encompassing the North Atlantic, Norwegian and Iceland Seas. The data is averaged over three box domains, as shown in Fig. 1, to represent the three sites in the Pliocene reconstructions (Section 2.3). The same domains are used in the CMIP6 model analysis (Section 2.2). We consider that SST averaged over the domains better represents the variability than a single grid point. The domains are chosen as follows: to represent the site in the NE North Atlantic, we use a domain covering the northeastern part of the Subpolar North Atlantic (49-57°N, 35-14°W); to represent the site along the NwAC, we use a box over the eastern Norwegian Sea (62.5-73°N, 0-16°E), and finally, to represent the site in the Iceland Sea, we use a box covering the major part of the Iceland Sea (66-72°N, 18-10°W). The analysed data set is from the Met Office Hadley Centre (version 1.1) and provides monthly global SST on a 1-degree latitude-longitude grid over the period from 1870 to 2012. A detailed description of the dataset is given in (Rayner et al., 2003). In this study we use the annual mean SST to document the existing SST anomalies and the spatial relation of these between the North Atlantic, Norwegian and Iceland Seas. To investigate multidecadal time scales over the HadlSST data set, we apply a 5-year running mean. SST anomalies are calculated relative to the mean of the respective records (1870-2012).

### 2.2 Transient simulations: CMIP6exp ssp126

The current generation of global climate models is available through CMIP6. CMIP6 provides a range of climate change experiments to the end of this century and beyond. To address whether or not the spatially noncoherent SST response may also occur in the future under less extreme atmospheric $CO_2$ forcing scenarios than for RCP8.5 we use monthly gridded SST data from the SSP126 experiment covering the time period from 2021 to 2100, with an approximate radiative forcing of 2.6 W/m$^2$ and a relatively low level of global warming (it is called the "2°C-scenario") by 2100. $CO_2$

concentrations reach 445 ppm by 2100 (Meinshausen et al., 2020), which is at the high end of the Pliocene $CO_2$ range (Bartoli et al., 2011; De La Vega et al., 2020). Similar to how we treat the HadISST data, we assess the annual mean SST from each of the model simulations. A 5-year running mean have been used to smooth the interannual variability. Because of the 5-year running mean filter, the time series are shown for the period 2023 to 2098. SST anomalies are calculated relative to the mean of the respective records (2021-2100).

The CMIP6 archive offers model output from many models. In this study, we have chosen to analyse three different models that have different equilibrium climate sensitivity (ECS; Meehl et al., 2020; Seland et al., 2020), leading to different amounts of warming by 2100: CNRM-ESM2-1 having the highest sensitivity of the three models (ECS=4.8), NorESM2-MM the lowest sensitivity (ECS=2.5), and MPI-ESM1-2-LR in between (ECS=3). In the analysis herein, we use one member from each model (i.e., one simulation from each of the three selected models). Some additional model simulations have been included, to check whether the responses to a more aggressive warming scenario (SSP585 experiment; NorESM2-MM), different resolution in the atmosphere (1 degree versus 2 degrees; NorESM2-MM vs. NorESM2-LM), and more members (10 members compared to one single member; MPI-ESM1-2-LR). The model data is averaged over the same domains as analysed for the observational data: NE North Atlantic (49-57°N, 35-14°W), Norwegian Sea (62.5-73°N, 0-16°E), and Iceland Sea (66-72°N, 18-10°W). SST anomalies relative to the mean of the 5-year running mean filtered time series are calculated for CNRM-ESM-2-1, one of the 10 MPI-ESM1-2-LR members, NorESM2-MM (SSP126) and NorESM2-MM (SSP585). We focus on the SST anomalies for the three domains and the relation between these at the end of the century (last three decades).

**2.3 Pliocene SST reconstructions**
To see if spatially noncoherent SST anomalies also occur between the North Atlantic and Norwegian and Iceland Seas under a past warm climate period in equilibrium with atmospheric $CO_2$ concentrations comparable to the SSP126 experiments we use a compilation of previously published Pliocene alkenone $U^K_{37}$' SST data from three sites from the northern NE North Atlantic (ODP Site 982; 57.5167°N, 15.8667°W; 1134.2 m water depth) (Herbert et al., 2016; Lawrence et al., 2009), the Norwegian Sea (ODP Site 642; 67.255°N, 2.928333°E; 1280.9 m water depth) (Bachem et al., 2016; Bachem et al., 2017) and the Iceland Sea (ODP Site 907; 69.24815°N, 12.69°W; 1801.5 m water depth) (Herbert et al., 2016) (Fig. 1). Each dataset covers the time interval between 5.23 and 3.13 Ma.

The UK37' index records the relative abundance of specific lipids (alkenones) synthesized by selected unicellular haptophyte algae living at or near the sea surface (e.g. Marlowe et al., 1984). Through the study of cultures, water samples and surface sediments, it has been shown that the UK37' index changes with temperature (Prahl and Wakeham, 1987; Müller et al., 1998; Conte et al., 2006; Tierney and Tingley, 2018). All records are presented here as previously published (Herbert et al., 2016; Lawrence et al., 2009; Bachem et al., 2016; Bachem et al., 2017), using established age models and the Müller et al. (1998) $U^K_{37}$'-SST calibration. The near-global and linear relationship between surface-sediment UK'37 values and mean annual SSTs (Muller et al., 1998) aligns closely to a culture study (Prahl and Wakeham, 1987), and has been used to calibrate and reconstruct mean annual SSTs. The standard error of estimate using this calibration is ±1.5°C (Müller et al., 1998). As a biological temperature proxy, it is important to consider both the environmental and biological influences over this UK37'-SST relationship. Marked local or regional differences in the timing of alkenone production and flux to the seafloor may impart a seasonal bias to the sedimentary record (e.g. Rosell-Mele and Prahl, 2013). In a recent expansion and Bayesian analysis of the global surface sediment calibration, a stronger correlation to August-October SSTs was identified in the North Atlantic (Tierney and Tingley, 2018) i.e. in the region of our study, which may

be supported by overlap between reconstructed SSTs and autumn multi-model means for ODP Site 982 and ODP Site 642 during the KM5c interglacial at 3.205 Ma (McClymont et al., 2020). In the Nordic Seas, low salinity or high sea ice have been linked to elevated production of the C37:4 alkenone (e.g. Bendle and Rosell-Mele, 2004; Wang et al., 2021). However, this alkenone is not included in the UK37' index, and was not recorded at values of concern at ODP Site 642 (Bachem et al., 2017).

The sampling resolution of the original records varies; for Site 907 (Iceland Sea) the mean original temporal resolution was ca. 2600 years, however, from 3.33 to 3.16 Ma the spacing between measurements ranges from 4000 to 70,000 years; for Site 642 (Norwegian Sea) the mean resolution was ca. 2500 years between 3.13 and 3.49 Ma and ca. 7200 years between 3.49 and 5.23 Ma; and for Site 982 it was ca. 2100 years between 3.13 and 4.03 Ma and ca 40800 years between 4.03 and 5.23 Ma. To enable direct comparison between sites, independent of differences in temporal resolution and absolute ages for the raw data points, each dataset has been resampled every 100 kyr between 5.23 and 3.13 Ma, using a linear integration function in *AnalySeries* (Paillard et al., 1996). SST anomalies are calculated relative to the mean of the respective resampled records (5.23-3.13 Ma). The 100-kyr resampling interval is chosen to put focus on the long-term trends of each record and the background climate state upon which the shorter-term orbital variability is superimposed (Fig. 2). The shorter-term orbital variability is not well enough resolved by all records throughout the investigated time interval to allow for a higher resolution resampling. Hence, given the time scales considered here, the SST anomaly relations are unlikely to be orbitally forced. Furthermore, focusing on the mean state of longer intervals (the shortest time interval is 200 000 years), rather than point to point comparison, also minimizes the impact of uncertainties from age models. Pliocene chronologies are mostly constrained by tuning to LR04 and/or using tie points from magnetic reversals. The tuning error is generally considered to be no more than a few thousand years, but may exceed 10 ka prior to 4.3 Ma due to less certain obliquity variance (Lisiecki and Raymo, 2005). At the time scales considered here such errors are acceptable.

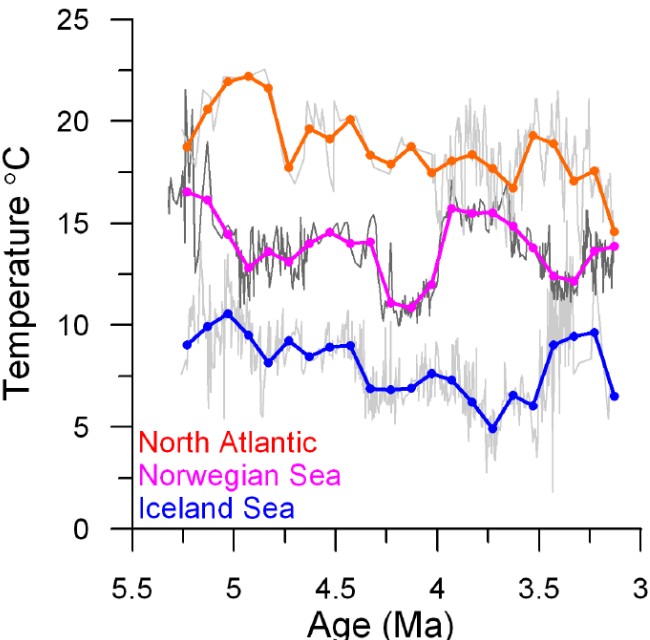

**Figure 2** Original UK'37 SST reconstructions from the North Atlantic, Norwegian and Iceland Seas (grey tones) (Bachem et al., 2016; Bachem et al., 2017; Herbert et al., 2016; Lawrence et al., 2009) with 100 kyr resampled datasets superimposed (North Atlantic (red), Norwegian (magenta) and Iceland Seas (blue)).

220

## 2.4 Model set up and reference experiments

We use an idealized-topography configuration of the MITgcm (Marshall et al., 1997) to investigate the SST relationships
in the study area under a range of buoyancy forcings (Fig. 1). The set-up (Fig. 3) is a Nordic Seas-like basin separated
from a truncated North Atlantic-like source water region by a 1000 m deep ridge. Both basins are flat-bottomed with 2000
225 m depth and surrounded by sloping sides. The model domain is closed. The boundary conditions and prescribed forcings
are the following: There is a restoring boundary condition in the south maintaining the reservoir of Atlantic source water.
The temperature is restored to a temperature of $T_A = 6°C$ and salinity 35 psu with a restoring strength of 40 W/m2 and a
time scale of one month. In addition, SSTs are restored toward atmospheric temperatures (SAT) through the surface heat
flux which is parameterized as "Q=(SST-SAT)*G" where G=40W/m²C. There is no interactive atmosphere. There is a
230 constant uniform freshwater input in the form of precipitation north of the ridge. Mechanical forcing is provided by a
constant-in-time prescribed wind field (W) with westerlies over the North Atlantic and easterlies over the Nordic Seas
field. The latitude of zero wind-stress curl is in the middle of the Atlantic region, with cyclonic wind stress to the north
and anti-cyclonic to the south.

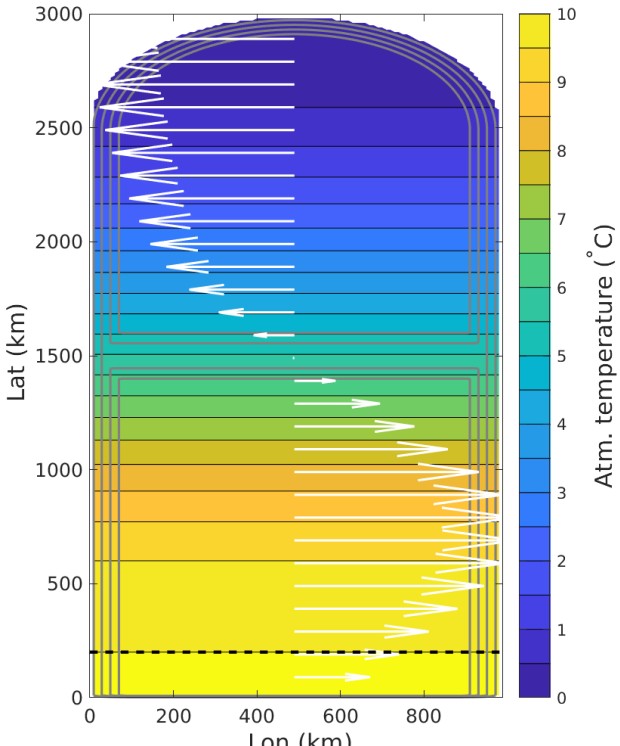

**Figure 3** The model setup. Grey contours outline the bathymetry (every 300 m), colours show the SAT (profile G1 in Fig. 4), and
white arrows represent the wind forcing. Fig. 4 provides an overview of the different buoyancy (SAT and freshwater) forcing used for
the experiments. South of the black dashed line salinity is restored to 35 psu and oceanic temperature is restored to 6°C.

240 The horizontal resolution is 10 km and there are 30 vertical layers; the upper 20 layers are 50 m thick and the deepest
layers are 100 m thick. Water density is calculated using the formula from Jackett and Mcdougall (1995), with a constant
Coriolis parameter, $f$ =1.2*10⁻⁴s⁻¹, and vertical diffusivity and viscosity of 1*10⁻⁵m²s⁻¹. Convection is parameterized with
implicit vertical diffusion; the diffusivity increases to 1000m²s⁻¹ for statically unstable conditions. Horizontal viscosity is

parameterized using the Smagorinsky closure (Smagorinsky, 1963): Typical values are $30m^2s^{-1}$ for the boundary current region. Temperature and salinity are advected using a third-order flux-limiting scheme.

Since one of the key drivers for inflow of Atlantic Water to the Norwegian Sea is buoyancy forcing and production of dense overflow water that must be compensated (Furevik et al., 2007), we change the SAT (G) and the freshwater (in form of precipitation) (P) north of the ridge to study the impact of buoyancy forcing on the relationships between SSTs in the North Atlantic, Norwegian Sea, and Iceland Sea. The SAT and freshwater are changed as shown in Fig. 4 and Table 1. Note that the SAT over the restoring region is the same for all experiments. The idealized model is run for 30 years, to near steady-state, and we present results from the last 5 years of the runs which are compared to the results from the relevant reference experiment. We are therefore not studying transient changes, but differences between equilibrium states.

Buoyancy changes are forced north of the ridge due to the nature of the model set up. The restoring boundary conditions in the south is also a forcing, both representing the (infinite) source of Atlantic water and the experiment's energy source (heat and buoyancy input). As the surface forcing is applied north of the ridge, water mass transformation takes place and a consistent ocean circulation is set up, including setting the hydrography of the different regions. The southern boundary energy input and northern surface heat loss balance when the model Has recached (quasi)-equilibrium. The northern and southern regions are accordingly equally important for the experiments.

We want to investigate the responses to changes in buoyancy caused by either a SAT or a freshwater change. In addition, we want to see if the initial state of the ocean impacts the response to a SAT change, specifically testing if the response differs if we start out from a fresher Nordic Seas. Therefore, we define three reference experiments, REF-1 (G0 and P1), REF-2 (G1 and P1) and REF-3 (G0 and P3) (Table 1). REF-1 is set up to investigate the oceanographic responses to a gradually decreasing SAT gradient between the North Atlantic and the Nordic Seas, under constant freshwater forcing, by increasing the SATs over the Nordic Seas. For the REF-2 experiments, the buoyancy is changed by gradually increasing the freshwater over the Nordic Seas while SAT is kept constant. The REF-3 experiments are similar to the REF-1 experiments in the sense that SAT over the Nordic Seas are increased, however, the initial state of the Nordic Seas is fresher for REF-3 than for REF-1. Hence the REF-3 experiments are set up to see how the initial state of the ocean may impact the responses to increased SAT over the Nordic Seas.

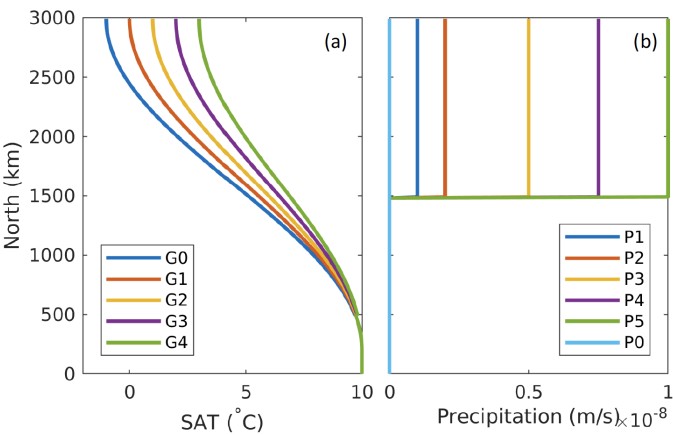

**Figure 4** (a) SAT forcing and (b) freshwater forcing for experiments.

The combination of the prescribed wind stress and the steep yet sloping coastal boundary supports a cyclonic boundary-intensified circulation around the Nordic Seas. The reference experiments have an ocean circulation which represents the main characteristics of the North Atlantic (south of the ridge), with an anticyclonic gyre in the "subpolar" latitudes and a cyclonic gyre further south. The buoyancy forcing from the prescribed surface temperature and salinity results in a gradual meridional temperature decrease and similar salinity decrease mimicking northern heat loss and freshwater input (Fig. 5). The thermal forcing dominates, and there is net northern buoyancy loss. There is warm and saline inflow to the Nordic Seas-like basin, and a colder and fresher outflow. The densest and coldest water is found in the relatively motionless and weakly stratified interior of the Nordic Seas. The overall temperature contrast between the dense-water interior and the Atlantic source water region reflects the temperature range of the prescribed surface air temperature. Waters are continuously exchanged between the buoyant boundary current and the interior by lateral eddy-mixing. Heat is thus not only lost from the boundary current by air-sea interaction but also by lateral heat loss to the interior, where it is also given up to the atmosphere.

When presenting MITgcm results, the North Atlantic domain is defined as the North Atlantic restoring region (Fig. 5c), set to be 6°C for all experiments (Fig. 4a). The Norwegian Sea domain is defined as a box in the eastern boundary current region, while the Iceland Sea domain is represented by the interior ocean north of the ridge (Fig. 5c). The definition of these domains, as the domains used for the observations and CMIP6 results (Section 2.1; Fig. 1), are directed by the location of the Pliocene sites, representing the Norwegian boundary current and interior Iceland Sea. Since the MITgcm setup is idealized it is not possible to set the exact same domains as used for the observations and CMIP6 results. However, within the limitations set by the individual data sources, all information extracted from the reconstructions, observations, CMIP6 and MITgcm experiments represents the Norwegian Sea boundary current and the interior Iceland Sea. Regional inference can therefore be made from the MITgcm experiments.

We identify spatially coherent/noncoherent SST anomaly relationships between the North Atlantic, Norwegian and Iceland Seas by comparing the temperature of the sensitivity experiment with the relevant reference experiment. A significant temperature change in one region is defined as a temperature change where the change between two experiments exceeds $2\sigma(\text{SST}_{\text{reference\_experiment}})$. The North Atlantic is restored to constant temperatures. Temperatures are not necessarily constant in the restoring region but restores towards constant temperatures. The restoring will dampen the potential temperature change, and therefore no significant temperature change is ever seen for this region. Thus, a change in SST anomaly relationship between the three regions exists if there is a significant temperature change in either the Norwegian Sea or the Iceland Sea, or both.

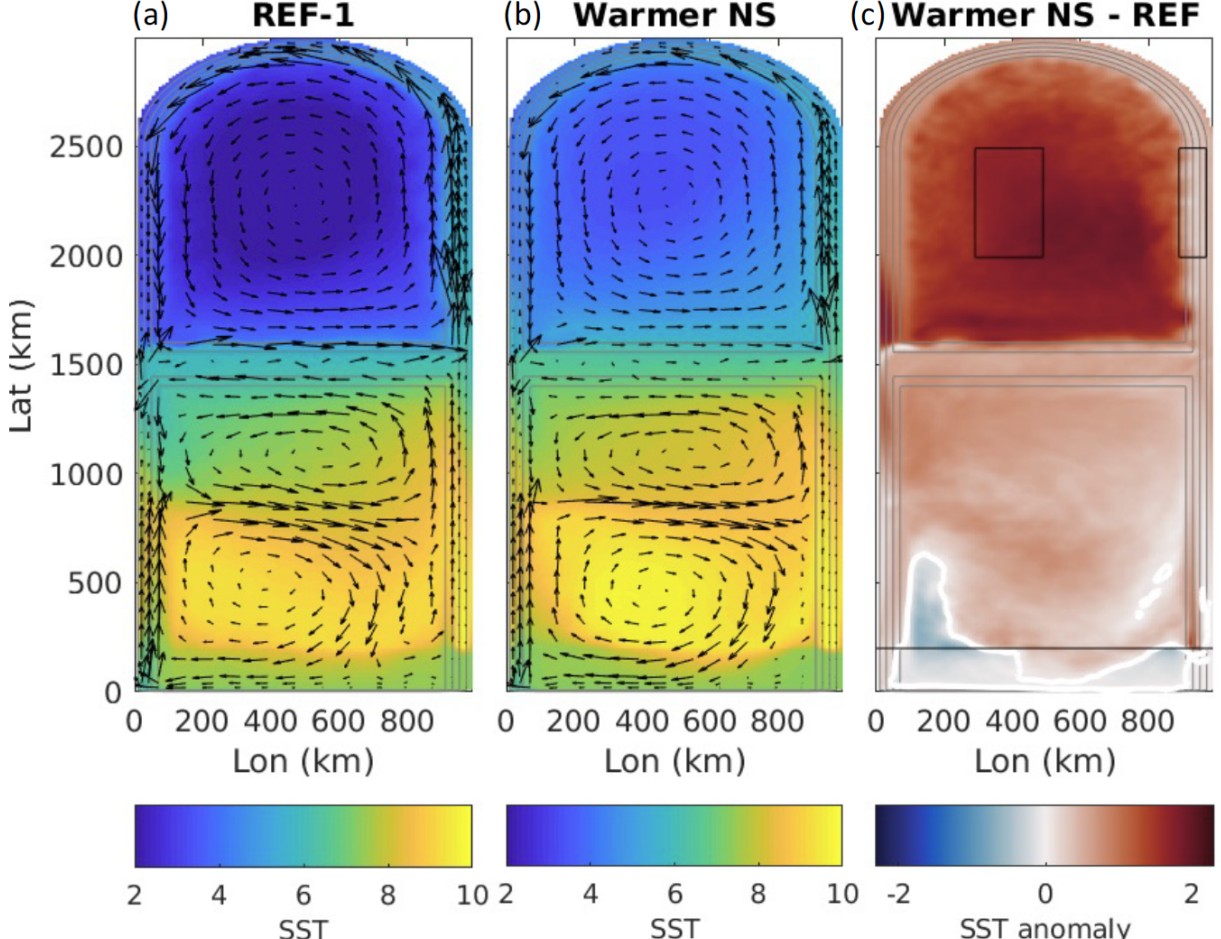

**Figure 5** SSTs (colours) and upper 500 m ocean circulation (black arrows) for experiments (a) REF-1 and (b) Warmer NS (G3, P1). (c) shows the difference between the two experiments. White mark the zero-contour line, and boxes show the areas used to calculate the SSTs of the Norwegian Sea (pink) and the Iceland Sea (blue) used in Table 1. The domains are set for the results to be comparable to the reconstructions, observations and CMIP6 results. Since the MITgcm is idealized the domains can, however, not be identical to the domains defined in Section 2.1. The North Atlantic restoring region equals the area south of the red line in (c).

## 3 Results

First, the relations between SST anomalies in the North Atlantic, Norwegian and Iceland Seas, as seen in HadlSST (multidecadal time scales), the low-emission future scenario runs (CMIP6 SSP126; multidecadal time scale) and Pliocene SST reconstructions (over several 100 kyr) are presented. Thereafter, we present the results of the idealized experiments testing the impact of changes in buoyancy forcing on the SST anomalies. The Pliocene reconstructions are sitespecific but considered to provide a reasonable representation of their respective regions while the observation and model data are regional averages. Somewhat larger amplitudes of the recorded SST anomalies may therefore be expected to be seen in the reconstructions.

### 3.1 SST anomaly relations in observation-based data: HadlSST

On multidecadal time scales the annual SSTanomaly, as seen in the HadlSST dataset, varies between -0.8 and +0.8°C (Fig. 6). As described in the introduction, the spatially noncoherent SST anomaly signal seen on shorter time scales should in theory no longer be of importance on multidecadal time scales, and we see a spatially coherent SST anomaly relationship between the North Atlantic, the Norwegian Sea and the Iceland Sea (Fig. 6). On multidecadal time scales,

these regions follow to a large extent the Atlantic Multidecadal Variability (AMV), with a warm phase in 1930-1970 and a cold phase in 1970-1990 (Knight et al., 2005).

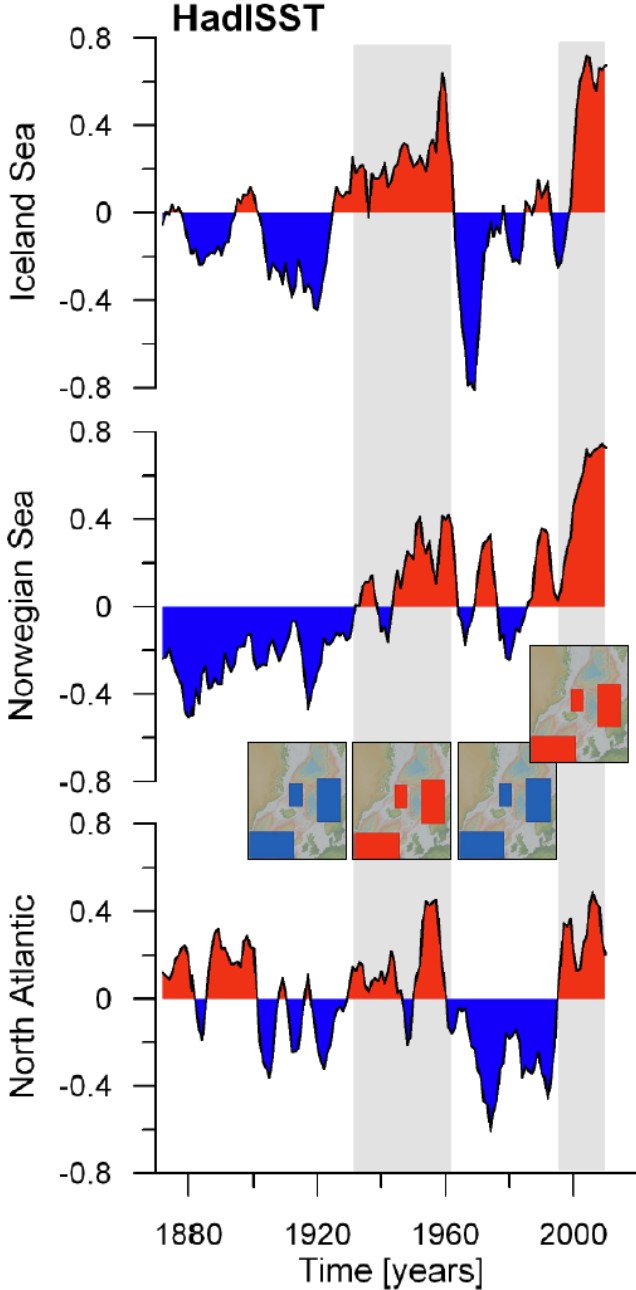

**Figure 6** Annual SST anomalies for the North Atlantic, Norwegian and Iceland Seas based on HadISST data. SSTs have been averaged
over the three box regions before calculating the anomalies: 49-57°N and 14-35°W (North Atlantic), 62.5-73°N and 0-16°E (Norwegian Sea), 66-72°N and 10-18°W (Iceland Sea). A running mean with a 5-year window has been applied on the time series and the anomalies are shown relative to the mean of the 5-year running mean filtered time series (1870-2012). The map inserts provide a conceptual representation of the results, with blue (red) boxes representing cold (warm) SST anomalies for the North Atlantic, Norwegian and Iceland Seas. The grey bars highlight the periods with positive spatially coherent SST anomalies. The base for the map inserts is made
with GeoMapApp (www.geomapapp.org) / CC BY / CC BY (Ryan et al., 2009)).

**3.2 Future SST anomaly relations - CMIP6exp: ssp126**

The three models, CNMR-ESM2-1, MPI-ESM1-2-LR and NorESM2-MM, show different results, both with respect to their SST climatology and the SST anomalies in each region of interest for the end of the 21$^{st}$ century (the last three

decades, 2068-2098) (Fig. 7). The mean SST of the 5-year running mean filtered time series (2023-2098) of the NE North Atlantic is fairly similar in the three models, but the Norwegian Sea differs to some extent with MPI-ESM1-2-LR being warmest (9.5°C), NorESM2-MM being coldest (6.8°C), and CNRM-ESM2-1 in between (7.8°C). The mean SST in the Iceland Sea differs to a large extent among the three models, again with MPI-ESM1-2-LR being warmest (5.6°C) and NorESM2-MM coldest (0°C).

CNRM-ESM2-1 shows larger SST anomalies for the Norwegian Sea and the Iceland Sea than the two other models (Fig. 7b). This is consistent with CNRM-ESM2-1 being the most sensitive, or more rapidly responding, model (as described in Section 2.2). Based on CNRM-ESM2-1, we find a dominantly cold SST anomaly in the North Atlantic and a warm SST anomaly in the Norwegian and the Iceland Seas towards the end of the 21$^{st}$ century (2068-2098). Thus, for the SSP126 scenario CNRM-ESM2-1 suggests that the North Atlantic SST anomaly will differ from the Norwegian and Iceland Seas SST anomalies at the end of the 21$^{st}$ century.

In contrast to the CNRM-ESM2-1 results, both MPI-ESM1-2-LR and NorESM2-MM show much smaller SST anomalies at the end of the 21$^{st}$ century relative to the models annual mean SST over the next century (Fig. 7b). Considering 10 different members from MPI-ESM1-2-LR, we find that the results from the individual members do not differ to a large extent (Fig. 7a); none of the members show anything but minor annual mean SST variability for any of the domains. The difference between the members is less than the amplitude of the changes in CNRM-ESM2-1. On the other hand, considering a more aggressive scenario (SSP585) for NorESM2-MM, we find a clear warm anomaly in the Norwegian and the Iceland Seas (Fig. 7b). A cold, but small, SST anomaly is seen for the North Atlantic. Hence, the sensitivity of the model impacts the result. Furthermore, we find that lowering the horizontal resolution in the atmosphere entails higher SSTs in the Iceland Sea at the end of the century relative to the results from the model version with a higher horizontal resolution in the atmosphere. A lowering horizontal resolution in the atmosphere does not, however, have a clear effect on the SSTs of the Norwegian Sea nor the North Atlantic (Fig. 7a).

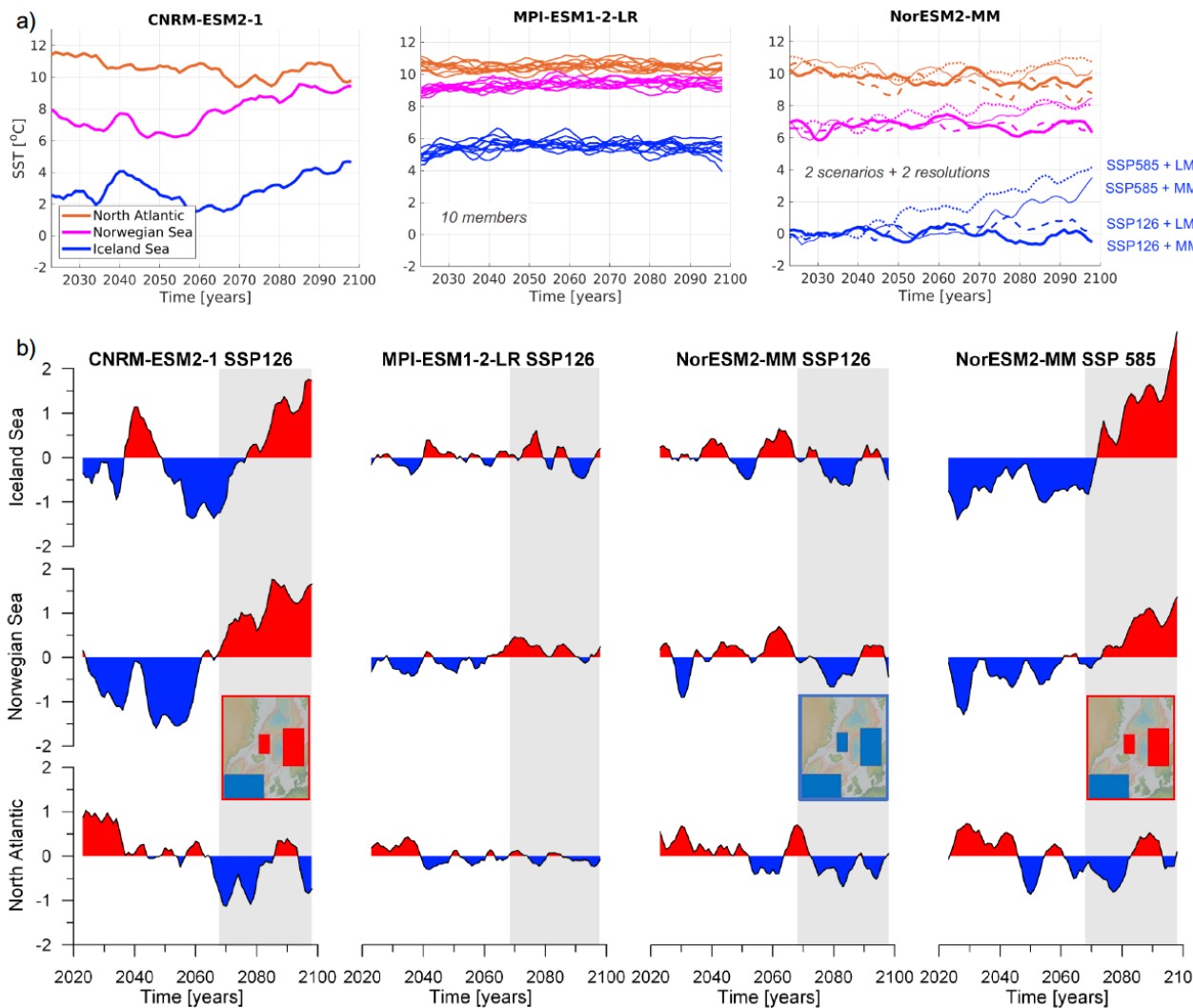

**Figure 7** a) Annual mean SST based on CMIP6 future scenario SSP126, representing the North Atlantic (red), Norwegian (magenta) and Iceland Seas (blue). SSTs have been averaged over the same box regions as described in Fig. 3. CNRM-ESM2-1 displays one member, MPI-ESM2-2-LR displays 10 members, and NorESM2-MM displays two different scenarios (SSP126/thick curves and SSP585/thin curves) and two different atmospheric resolutions (medium/solid curves and low/dashed curves). A running mean with a 5-year window has been applied on the time series. b) SST anomalies relative to the mean of the 5-year running mean filtered time series (2023 to 2098) from CNRM-ESM-2-1 (SSP126), one of the 10 MPI-ESM1-2-LR members, NorESM2-MM (SSP126) and NorESM2-MM (SSP585). We focus on the SST anomalies for the three domains and the relation between these at the end of the century (last three decades, 2068-2098, marked by the grey bars). The map inserts provide a conceptual representation of the results, with blue (red) boxes representing cold (warm) SST anomalies for the North Atlantic, Norwegian and Iceland Seas. The individual members of MPI-ESM1-2-LR SSP126 cannot be distinguished from each other. Therefore, we have not added a map insert for the member presented.  The base for the map inserts is made with GeoMapApp (www.geomapapp.org) / CC BY / CC BY (Ryan et al., 2009)).

### 3.3 SST anomaly relations in Pliocene SST reconstructions

The relation between SST anomalies of the North Atlantic, Norwegian Sea and Iceland Sea are not constant through the Pliocene. Two different spatially noncoherent SST anomaly relations are documented: 1) The Norwegian Sea SST anomaly differs from the SST anomalies of the North Atlantic and the Iceland Sea, either with a warm Norwegian Sea anomaly corresponding with a cold anomaly in the North Atlantic and Iceland Sea (first yellow period in Fig. 8; 3.63-3.93 Ma), or the opposite, a cold anomaly in the Norwegian Sea corresponds to a warm anomaly in the North Atlantic and Iceland Sea (second yellow period in Fig. 9; 4.73-4.93 Ma). 2) A cold anomaly in the North Atlantic and the

Norwegian Sea corresponds to a warm anomaly in the Iceland Sea (blue period in Fig. 9; 3.23-3.43 Ma). All these Pliocene SST anomaly relationships are different from what we see in the CMIP6 future runs (warm Norwegian Sea and Iceland Sea, cold North Atlantic). In addition to the spatially noncoherent SST anomaly relations, there are three time periods during the Pliocene that show a spatially coherent SST anomaly relationship, comparable to what we find for the observation-based data (grey time periods in Fig. 9; 4.03-4.33 Ma, 4.43-4.63 Ma and 5.03-5.23 Ma).

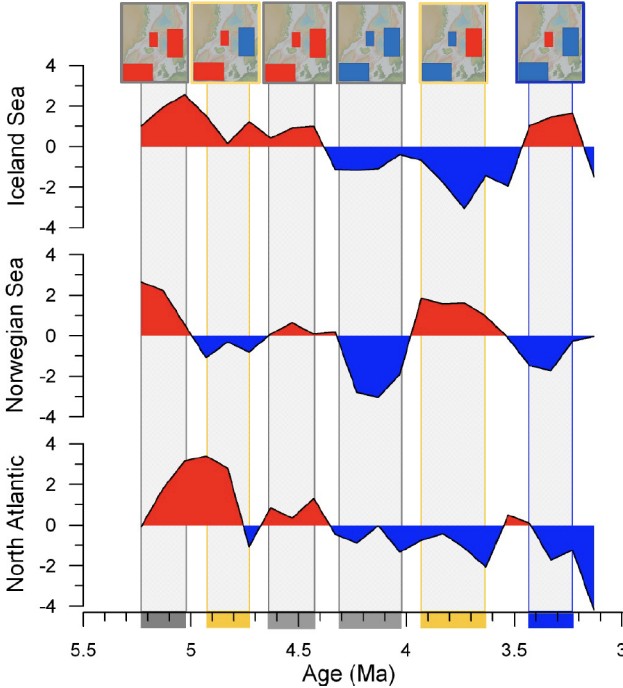

**Figure 8** Pliocene SST anomalies (°C, relative to the mean of the 100 kyr resampled records). The different SST anomaly relations identified are colour coded (grey boxes - spatial coherence; blue box - Iceland Sea SST anomaly differs from the North Atlantic and the Norwegian Sea SST anomalies; yellow boxes - the Norwegian Sea SST anomaly differs from the North Atlantic and the Iceland Sea SST anomalies). Conceptual illustrations of the respective SST anomaly relations for each interval are shown for all identified scenarios, with blue (red) boxes representing cold (warm) SST anomalies for the North Atlantic, Norwegian and Iceland Seas (from left to right: positive spatial coherence; cold Norwegian Sea, warm North Atlantic and Iceland Sea; positive spatial coherence; negative spatial coherence; warm Norwegian Sea, cold North Atlantic and Iceland Sea; and warm Iceland Sea, cold North Atlantic and Norwegian Sea). The base for the map inserts is made with GeoMapApp (www.geomapapp.org) / CC BY / CC BY (Ryan et al., 2009)).

**3.4 Buoyancy forced SST anomaly relationships - results from the MITgcm idealized experiments**

By changing the buoyancy forcing as shown in Fig. 4 and 5 we can produce three different spatial SST anomaly relationships in the idealized model: spatially coherent SST anomalies in all three regions (grey experiments, Table 1 and Fig. 9b); Norwegian Sea and Iceland Sea SST anomalies different from the North Atlantic (red experiments, Table 1 and Fig. 9); and an Iceland Sea SST anomaly that differ from the North Atlantic and the Norwegian Sea SST anomalies (blue experiments, Table 1 and Fig. 9). Hence, these idealized experiments can capture two of the three SST anomaly relations found during the Pliocene (grey and blue time periods in Fig. 8) and the SST anomaly relationship found in the CMIP6 future runs (Fig. 7). Table 1 summarizes the experiments.

The first set of four experiments have the same freshwater forcing (P1), but a decreasing SAT-gradient (i.e., increasing SAT over the Nordic Seas (G1-G4)) relative to REF-1. As SAT increases over the Nordic Seas the SST pattern shift from the Iceland Sea SST anomaly being different from the North Atlantic and the Norwegian Sea (G1) to the North Atlantic SST anomaly being different from the Norwegian and Iceland Seas (G2-G4) (Fig. 9a).


In the next set of experiments SAT is kept constant (G1) while the freshwater over the Nordic Seas is increased (P2-P5) relative to REF-2. With increasing freshwater over the Nordic Seas, the SST pattern shifts from spatial coherence (P2 and P3) to the Iceland Sea SST anomaly being different from the North Atlantic and the Norwegian Sea (P4) to the North Atlantic SST anomaly being different from the Norwegian and Iceland Seas (P5) (Fig. 9b).


In the last set of experiments, the decreasing SAT-gradient experiments (G1-G2) are repeated with a fresher Nordic Seas (P3) relative to REF-3. As for REF-1, the SST pattern shift from the Iceland Sea SST anomaly being different from the North Atlantic and the Norwegian Sea (G1) to the North Atlantic SST anomaly being different from the Norwegian and Iceland Seas (G2) when we increase the SAT over the Nordic Seas (Fig. 9c).


**Table 1** Selected output from the MITgcm idealized model experiments where buoyancy is changed by changing either SAT (G) or freshwater in form of precipitation (P). SAT is atmospheric temperature forcing (Fig. 4), FW is freshwater forcing (Fig. 4), $T_{nws}$ and $T_{ice}$ is sea surface temperature of the boundary current/Norwegian Sea and interior/Iceland Sea (Fig. 5) for the three reference experiments. $\Delta T_{nws}$-ref and $\Delta T_{ice}$-ref is the temperature difference between the experiment and the corresponding ref-experiment for

the boundary current/Norwegian Sea and interior/Iceland Sea, respectively. The numbers are marked in bold if the temperature difference exceeds the 2*std of the reference experiment. $\Delta D_{north-south}$ is the density difference between north (averaged over 2000-2500 km) and south (averaged over 500-1000 km) of the ridge within each experiment over the full depth. $V_{inflow}$ is the mean inflow velocity across the sill (cm/s) (at 1500 km), $V_{bc}$ is the mean velocity in the boundary current (cm/s) (average for the Norwegian Sea as defined by the pink box in Fig. 5), $HT_{sill}$ is the net heat transport across the sill (TW) (at 1500 km), and NMOC (Sv) is the maximum overturning

streamfunction at the sill (at 1500 km). Ref-exp is the corresponding reference experiment which the experiment is compared against. The three different SST anomaly relations identified is colour coded (grey - spatial coherence; red - Norwegian Sea and Iceland Sea SST anomaly different from North Atlantic; blue - a significant change in the Iceland Sea and no change in the Norwegian Sea and North Atlantic, considered to be a representation of an Iceland Sea SST anomaly different from North Atlantic and the Norwegian Sea).

| Ref exp / Colour code | SAT_FW to REF | $T_{nws}\pm$ 2std ($T_{nws}$) | $T_{ice}\pm$ 2std ($T_{ice}$) | $\Delta T_{nws}$-ref | $\Delta T_{ice}$-ref | $\Delta D_{north-south}$ | $V_{inflow}$ | $V_{bc}$ | $HT_{sill}$ | NMOC (Sv) |
|---|---|---|---|---|---|---|---|---|---|---|
| *REF -1 (G0 P1)* | | *5.18± 0.16* | *1.71±0.15* | | | *0.36* | *1.76* | *10.92* | *100.59* | *3.72* |
| Warmer SAT / **blue** | G1_P1 *to REF-1* | | | 0.10 | **0.51** | 0.33 | 1.62 | 9.95 | 82.33 | 3.33 |
| Warmer SAT / **red** | G2_P1 *to REF-1* | | | **0.28** | **1.11** | 0.30 | 1.44 | 8.76 | 59.24 | 2.89 |
| Warmer SAT / **red** | G3_P1 *to REF-1* | | | **0.44** | **1.74** | 0.25 | 1.54 | 7.19 | 39.24 | 2.49 |
| Warmer SAT / **red** | G4 P1 *to REF-1* | | | **0.69** | **2.42** | 0.2 | 1.05 | 5.58 | 24.57 | 1.99 |
| *REF-2 (G1 P1)* | | *5.28± 0.11* | *2.22± 0.10* | | | *0.33* | *1.62* | *9.95* | *82.33* | *3.33* |
| Fresher NS / **grey** | G1 P2 *to REF-2* | | | 0.02 | 0.00 | 0.32 | 1.59 | 9.67 | 82.25 | 3.34 |
| Fresher NS / **grey** | G1 P3 *to REF-2* | | | -0.04 | -0.08 | 0.31 | 1.45 | 8.94 | 76.78 | 3.26 |
| Fresher NS / **blue** | G1 P4 *to REF-2* | | | -0.05 | **-0.54** | 0.27 | 1.82 | 7.34 | 70.66 | 3.42 |
| Fresher NS / **red** | G1 P5 *to REF-2* | | | **-0.17** | **-0.64** | 0.23 | 1.66 | 5.86 | 59.52 | 2.86 |
| Saltier NS / **grey** | G1 P0 *to REF-2* | | | 0.04 | 0.07 | 0.33 | 1.47 | 10.10 | 82.17 | 3.24 |
| *REF-3 (G0 P3)* | | *5.16± 0.12* | *1.61± 0.09* | | | *0.34* | *2.00* | *10.34* | *94.21* | *3.64* |
| Warmer SAT / **blue** | G1 P3 *to REF-3* | | | 0.08 | **0.53** | 0.31 | 1.45 | 8.94 | 76.78 | 3.26 |
| Warmer SAT / **red** | G2 P3 *to REF-3* | | | **0.25** | **0.96** | 0.26 | 1.41 | 7.44 | 54.50 | 2.88 |


In addition to identifying the SST anomalies in the MITgcm experiments, information about the density difference over the ridge, mean inflow velocity across the sill, the mean velocity in the boundary current, the net heat transport over the sill and the maximum overturning streamfunction at the sill is extracted for each experiment (Table 1). This information will be used in the discussion, exemplifying oceanographic responses to specific buoyancy changes as seen in the MITgcm experiments.

Reducing the SAT-gradient, i.e., warming the atmosphere over the Nordic Seas, reduces the heat loss from the ocean and warms the SSTs in the Nordic Seas. Compared to the reference experiment, the following are smaller: the north-south density difference; the mean inflow velocity across the ridge; the boundary current velocity in the Nordic Seas; the net heat transport across the sill; the maximum overturning circulation across the sill; and the lateral eddy heat transport in the Nordic Seas (Table 1). The weaker ocean circulation transports less heat to the Norwegian Sea/Nordic Seas. In addition, as the Norwegian Sea boundary current is slower in warmer experiments, the water of the boundary current experiences more cooling as it travels the Nordic Seas, allowing for a larger heat loss in the Norwegian Sea region.

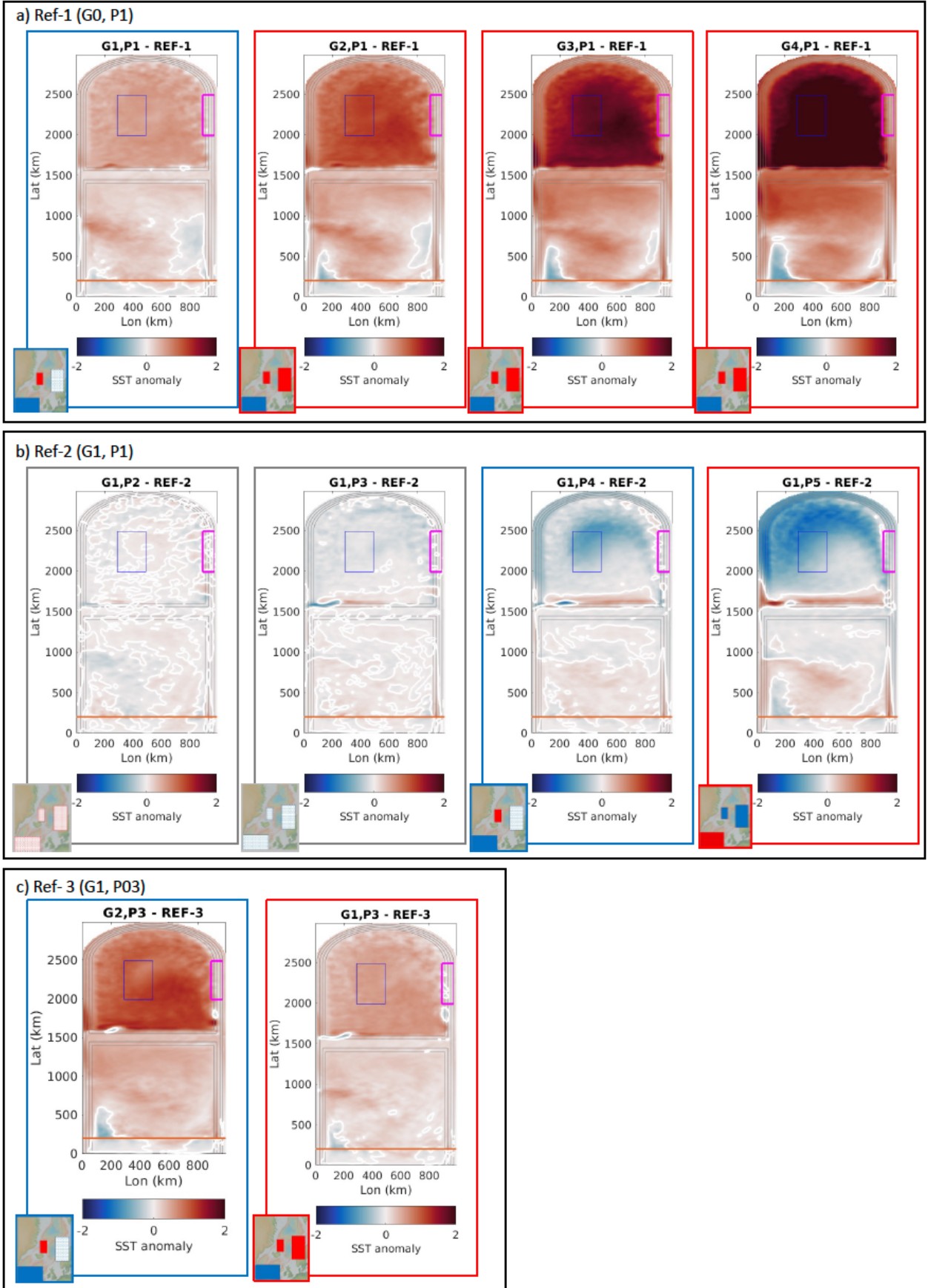

**Figure 9** SST anomalies seen in MITgcm idealized experiments. a) SST anomalies relative to REF- 1, where freshwater is kept constant at P1 while SAT is gradually increased (G1-G4). b) SST anomalies relative to REF-2, where SAT is kept constant at G1 while freshwater is gradually increased (P2-P5). c) SST anomalies relative to REF-3, where freshwater is kept constant at P3 while SAT is increased (G1 and G2). Surrounding blue boxes represent an Iceland Sea SST anomaly spatially incoherent with the North

Atlantic and the Norwegian Sea anomalies. The SST anomaly seen in the Iceland Sea exceeds 2*std of the relevant reference experiment. Surrounding red boxes represent a North Atlantic SST anomaly spatially incoherent with the Norwegian and Iceland Seas anomalies. The SST anomalies seen in the Norwegian and Iceland Seas exceeds 2*std of the relevant reference experiment. Surrounding grey boxes represent spatially coherent SST anomalies between the three regions (not significant responses; none of the SST anomalies exceeds 2*std of the relevant reference experiment). Conceptualized representation of the resulting SST anomaly relations is shown by the map inserts, where blue (red) boxes represent cold (warm) SST anomalies for the North Atlantic, Norwegian Sea and Iceland Sea. Lighter colours are used if the SST anomalies is less than 2*std of the relevant reference experiment. The base for the map inserts is made with GeoMapApp (www.geomapapp.org) / CC BY / CC BY (Ryan et al., 2009)).


For the experiments with a small SAT change (G1 relative to REF-1 and G1 relative to REF-3; blue experiments in Table 1 and Fig. 9a and c), the increased heat loss from the Norwegian Sea boundary current and decreased poleward heat transport are partly able to compensate for the increased atmospheric warming in the Norwegian Sea, and thus there is no significant temperature change in the Norwegian Sea. However, this is not the case for the Iceland Sea where the SSTs

increase. We consider this result to be representative for a situation when the SST anomaly of the Iceland Sea differs from the North Atlantic and Norwegian Sea SST anomalies, hence, an SST field that breaks the expectation of spatially coherent SST anomalies. The amplitude of the MITgcm Iceland Sea SST anomaly is in this case (G1 relative to REF-1 and G1 relative to REF-3) ca. 1/3 of the Iceland Sea SST anomaly reconstructed for the Pliocene (3.43 to 3.23 Ma) when the SST anomaly of the Iceland Sea differs from the North Atlantic and Norwegian Sea SST anomalies.


For larger SAT warming over the northern basin (G2-G4 relative to REF-1 and G2 relative to REF3; red experiments; Table 1 and Fig. 9a and c), both the Iceland and Norwegian Sea experience a significant temperature increase. There is more warming in the Iceland Sea than in the Norwegian Sea and the difference increases with larger SAT warming (i.e., weaker SAT-gradients). Thus, the absolute temperature for the Iceland Sea is more like the Norwegian Sea for a warmer

atmosphere over the Nordic Seas. The larger SST change in the Iceland Sea can be explained by a combination of reduced heat transport to the Norwegian Sea/Nordic Seas and slower Nordic Seas boundary current allowing for a larger heat loss in the Norwegian Sea region and more cooling of the water as it travels the Nordic Seas. The increased heat loss and decreased poleward heat transport counteracts the general atmospheric warming over the Norwegian Sea more so than over the Iceland Sea. The amplitude of the SST anomalies in the Iceland and Norwegian Sea as seen in CNRM-ESM2-1

ssp126 and NorESM-MM ssp 528 at the end of the century is within the range of the respective MITgcm SST anomalies.

Increasing the north-south salinity gradient across the sill (REF-2 experiments) gives similar results; no significant SST-change in the two regions for small changes in freshwater forcing (P2-P3 relative to REF2), an SST change only in the Iceland Sea for medium changes in freshwater forcing (P4 to REF-2), and SST changes in both the Iceland and Norwegian

Seas for the largest changes in freshwater forcing (P5 to REF-2, equal to a freshwater increase from 1 to 10 *1e-9 m/s (Fig. 4)) (Table 1 and Fig. 9b). A fresher Nordic Seas weakens the north-south density gradient which weakens the poleward heat transport and the boundary current velocity. The slower boundary current allows for more heat loss to the atmosphere in the Norwegian Sea, and the region cools. In these experiments, there is no prescribed warming of the atmosphere, so temperature change is only set by the changes in ocean circulation. The slower boundary current also

reduces the eddy heat fluxes from the boundary to the interior, and hence also the Iceland Sea region cools. As for the experiments where we increase the atmospheric temperature over the Nordic Seas, the Iceland Sea SST anomaly in the respective Pliocene case is larger than the MITgcm SST anomaly response to a medium freshwater change (P4 to REF-2). The MITgcm response to the largest freshwater forcing (P5 to REF-2) is at the lower end of the SST anomalies seen for the CMIP6 models at the end of the century.

Within the investigated parameter space, we have not found a situation where the increased heat loss and decreased poleward heat transport more than compensates for the increased atmospheric warming so that the Norwegian Sea temperature change is opposite to that of the other two regions (i.e., the situation seen for the yellow time periods in Fig. 8).

We have limited our study to the impact of buoyancy forcing on SST relationships. Using a similar model configuration, Spall (2011) and Spall (2012) report on the impact of changing sill depth on the temperature of the interior region (the inflowing temperature, i.e., Norwegian Sea temperature is assumed to equal that of the source region). A deeper sill increases the temperature of both the boundary current and the interior of the Nordic Seas. The difference in temperature between these two regions decreases for larger sill depths. The changes are associated with a strengthening of the meridional overturning circulation across the sill, and to a lesser extent to an increase in heat transport across the sill.

## 4 Discussion

In this study, we have identified four different SST anomaly relations (Fig. 10): 1) The North Atlantic, Norwegian and Iceland Seas SST anomalies being spatially coherent (at multidecadal time scales in the observation-based records and over hundreds of thousands of years for three Pliocene intervals (4.03-4.33 Ma, 4.43-4.63 Ma and 5.03-5.23 Ma; Fig. 6 and 8). 2) The Iceland Sea SST anomaly being different from the North Atlantic and the Norwegian Sea SST anomalies (over two hundred thousand years in the Pliocene, 3.23-3.43 Ma; Fig. 8). 3) The North Atlantic SST anomaly being different from the Norwegian and Iceland Seas SST anomalies (at multidecadal time scales at the end of the 21st century, Fig. 7). 4) The Norwegian Sea SST anomaly being different from the North Atlantic and the Iceland Sea SST anomalies (again over hundreds of thousands of years during the Pliocene, 3.63-3.93 Ma and 4.73-4.93 Ma; Fig. 8). From our idealized MITgcm experiments for the North-Atlantic - Nordic Seas region, three of the four different SST anomaly relations (1, 2 and 3) could be reproduced by changing the buoyancy forcing (atmospheric temperature and freshwater, Table 1; Fig. 9; Fig. 10). Our results thus suggest a key role for buoyancy forcing in setting the SST anomaly variability in the Northern North Atlantic.

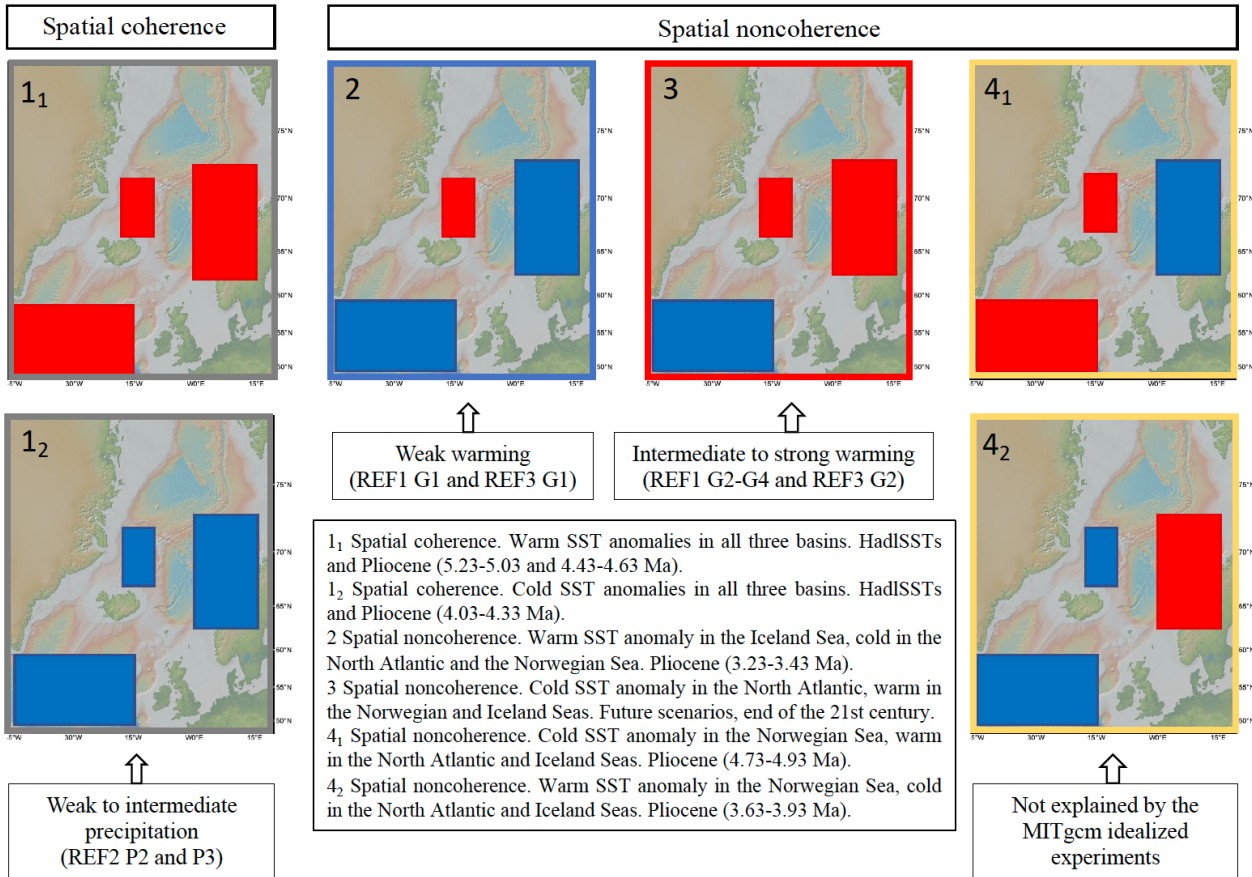

**Figure 10** Conceptual illustration of the identified SST anomaly relations, showing for which period the explicit SST anomaly relations are seen and which MITgcm idealized experiment resulted in the same type of SST relationships. Blue (red) boxes representing cold (warm) SST anomalies. The background maps are made with GeoMapApp (www.geomapapp.org) / CC BY / CC BY (Ryan et al., 2009)).

From the MITgcm experiments we see the SST anomaly responses to different degrees and causes of buoyancy forcing. In addition, we extract information on physical characteristics associated with the individual experiments and SST anomaly relations (Section 3.4). For the discussion we have searched for information on factors that may have impacted the Nordic Seas buoyancy during the individual Pliocene time periods and the future. For example, information on regional freshwater change is used as an indicator of buoyancy change. In addition, we have searched for information that informs on characteristics somewhat comparable to the physical characteristics extracted from the MITgcm experiments. These parameters include global SSTs, overturning circulation and ventilation of the Nordic Seas, the Atlantic Ocean equator-pole SST gradient and freshwater (Table 2). The content of Table 2 will, together with Table 1, form the basis for the discussion. For each SST anomaly relation identified in the Pliocene reconstructions or the CMIP6 results we will use the information from Table 1 and 2 to see if the SST anomaly change can be linked to a change in buoyancy, and if so, are the associated characteristics comparable to the MITgcm output for a similar SST anomaly relation.

**Table 2** An overview of reviewed information with respect to the Atlantic Ocean equator-pole SST gradient, ocean circulation changes (AMOC, relative proportion of North Atlantic Deep Water (NADW) and ventilation of the Nordic Seas), global SSTs, freshwater, and temperature over Norway, for the Pliocene and future. Pliocene information is extracted from available published reconstructions, after resampling every 100 kyr and in most cases presented as anomalies relative to the Pliocene resampled mean, hence, in the exact same manner as for the North Atlantic, Norwegian and Iceland Seas SST datasets analysed (Section 2.1), to secure methodological consistency on how information is extracted (further information and figures showing the relevant resampled datasets as anomaly plots are available in the Supplementary Information). All information related to the future is based on a literature review. The different SST

anomaly relations identified is given a colour code that reflects the colour code used throughout the paper (grey - spatial coherence; blue - Iceland Sea SST anomaly different from the North Atlantic and the Norwegian Sea; yellow - the Norwegian Sea SST anomaly different from the North Atlantic and the Iceland Sea; red - the North Atlantic SST anomaly different from the Norwegian and Iceland Seas).

| Time interval | 5.03-5.23 Ma | 4.73-4.93 Ma | 4.43-4.63 Ma | 4.03-4.33 Ma | 3.63-3.93 Ma | 3.23-3.43 Ma | CMIP6 (future trend) |
|---|---|---|---|---|---|---|---|
| SST anomaly relation | Warm spatial coherence (grey) | Cold NS, warm NA and IS (yellow) | Warm spatial coherence (grey) | Cold spatial coherence (grey) | Warm NS, cold NA and IS (yellow) | Warm IS, cold NA and NS (blue) | Cold NA, warm NS and IS (red) |
| Atlantic equator-pole T°C gradient [1] | Overall weaker than WOA annual mean 0-20 m (Locarini et al., 2018). | | | | | | |
| | Weak | Weak | Weak | Strong | Strong | Strong | |
| AMOC/ %NADW [2] | Strong (91) | Strong (69) | Strong (79) | No change (66) | No change (64) | Weak (23) | Weak in all scenarios (Weijer et al., 2020). |
| Nordic Seas deep ventilation [3] | No deep ventilation in the Nordic Seas (Jansen et al., 2000; Risebrobakken et al., 2016). | | | | | | No indication of reduced deep-water formation (Furevik et al., 2002^). |
| Norwegian Sea Ventilation, intermediate depth [4] | Weak | Weak | Weak | No change | Strong | Strong | |
| Norwegian Sea ventilation, upper water column [4] | Weak | Strong | Strong-to-weak | Weak | Strong | Strong | |
| Global SSTs [5] | Overall higher SSTs relative to today; weaker meridional gradients. Most sites experience cooling through the Pliocene, hence, moving from more positive to more negative anomalies relative to mean of self. | | | | | | Overall higher SST, except the subpolar North Atlantic in SSP126 (Kwiatkowski et al., 2020). |
| Freshwater/ Salinity [6] | No trace of sea ice. Extent unknown. | | IS: Indication of sea ice from 4.5 Ma | Traces of sea ice in the Iceland Sea and at the Yermak Plateau (maximum extent as for present summers) | | | No sea ice in the IS in September, but could be some in March in SSP126 (Wei et al., 2020; Derepentigny et al., 2020) |
| | - | - | - /+ | + | ++ | ++ | |
| | Closed Bering Str./closed CAA: less liquid freshwater into the Nordic Seas. | | Bering Strait transitioning from closed to open. | Open Bering Strait/closed CAA: more liquid freshwater into the Nordic Seas. | | | Fresher subpolar NA in SSP126 (Kwiatkowski et al., 2020), The liquid freshwater export increases in the Fram Strait in SSP245 (Zanowski et al., 2021). |
| Temperature over Norway [7] | No data | Warm | Warm | Cold | Warm | Cold | |

[1] See Fig. S1. Strength refers to relative relation between Pliocene states. [2] For Pliocene, the %NADW is calculated following (Bell et al., 2015). The indicated strengths are presented relative to Pliocene mean (ca 62%NADW). See Fig. S2 and Table S1 for Pliocene background information and dataset references. [3] Not from CMIP6. [4] Ventilation indicated relative to Pliocene mean. For further details see Fig. S2. [5] See Fig. S3 and S4 for Pliocene background data. [6] Information about Pliocene sea ice occurrence and extent is extracted from (Clotten et al., 2018; Clotten et al., 2019; Knies et al., 2014). Relative sea ice relation between intervals is indicated by +/-. The references to more/less liquid freshwater refer to a relative relation between Pliocene states. [7] Based on data from Panitz et al. (2018). See Fig. S6 for further information.

We note that minor geographical differences exist between the Pliocene and the present and future. The Greenland Scotland Ridge was deeper by a few hundred meters (Poore et al., 2006), the Canadian Arctic Archipelago was closed (Matthiessen et al., 2009) and the Barents Sea was most likely subaerial (Butt et al., 2002). These differences were, however, constant through the investigated time interval and would therefore not impact the interpretation of our results. In contrast, the Bering Strait opened during the investigated time interval and is suggested to have altered the Arctic freshwater balance and consequently the Nordic Seas oceanography (Table 2) (De Schepper et al., 2015; Otto-Bliesner et al., 2016). The consequences of the opening of the Bering Strait are therefore considered, as the change in freshwater balance will impact the Nordic Seas buoyancy.

The spatial coherence situation is addressed in Section 4.1. The Iceland Sea SST anomaly being different from the North Atlantic and Norwegian Sea SST anomalies is addressed in Section 4.2, and the situation when the North Atlantic SST anomaly differs from the Iceland and Norwegian Seas SST anomalies is addressed in Section 4.3. The situation where

the Norwegian Sea SST anomaly differs from both the North Atlantic and the Iceland Sea is not seen in any of the idealized experiments. This case will be discussed in Section 4.4.

### 4.1 Spatial coherent SST anomalies

Spatially coherent SST anomalies in the North Atlantic, Norwegian and Iceland Seas are seen at multidecadal time scales in the HadlSST dataset (Fig. 6) and over three Pliocene time intervals covering hundreds of thousands of years, from 5.23-5.03 Ma, 4.65-4.43 Ma and 4.33-4.03 Ma (Fig. 8). In the observation-based record the spatial coherence is linked to the continuous northward propagation of heat anomalies from the subpolar region and towards the Arctic, taking about 5-10 years to propagate north with a frequency of about 14 years (Årthun et al., 2017). Spatially coherent SST anomalies are also seen for more than half of the Pliocene time interval considered in this study, and therefore we consider spatial coherence to be the norm also for Pliocene climate, reflecting the connection of the three regions by ocean circulation (Fig. 1).

From the idealized experiments spatial coherence is seen under weak freshwater perturbations over the Nordic Seas, under constant SATs (Table 1; Fig. 9b). The responses, however, are small, and none of the selected output parameters from the idealized experiments show responses larger than 2*std relative to REF-2 (Table 1). During the cold Pliocene spatially coherent interval, neither the overturning circulation nor the intermediate depth ventilation of the Norwegian Sea deviated from its mean Pliocene conditions (Table 2), consistent with the weak responses seen in the MITgcm experiments for the comparable SST anomaly relation. During the warm spatially coherent intervals of the Pliocene the overturning circulation, as derived from the % North Atlantic Deep Water (%NADW) (Table 2), was however somewhat stronger than the Pliocene mean, while the intermediate depth ventilation of the Norwegian Sea was weaker than the Pliocene mean (Table 2). In line with a weaker overturning circulation during the cold relative to the warm spatially coherent intervals, a stronger North Atlantic meridional SST gradient and somewhat enhanced freshwater, or buoyancy, forcing characterized the cold relative to the warm spatially coherent SST anomalies of the Pliocene (Table 2). The characteristics associated with the warm spatially coherent SST intervals are thus less comparable to the MITgcm results than the characteristics associated with the cold spatially coherent SST interval.

The main difference between the Pliocene periods with warm rather than cold spatial coherent SST anomalies was a somewhat stronger freshwater influence during the cold interval, inferred from the occurrence of the sea ice marker IP$_{25}$ in the Iceland Sea (Clotten et al., 2019) and that the Bering Strait was fully opened (De Schepper et al., 2015) (Table 2). While there exists evidence for sea ice in the Arctic and the Iceland Sea, it is important to note that the Arctic sea ice extent was considerably smaller than today throughout the Pliocene (Clotten et al., 2019; Knies et al., 2014). Since the Canadian Arctic Archipelago (Fig. 1) was closed throughout the Pliocene (Matthiessen et al., 2009) and the mean Bering Strait throughflow is directed northwards into the Arctic Ocean (Woodgate and Aagaard, 2005), the Fram Strait was the only Arctic Ocean exit during the Pliocene. Opening the Bering Strait allowed for inflow of Pacific water with a lower salinity to the Arctic Ocean, and consequently, enhanced transport of freshwater from the Arctic Ocean to the Nordic Seas (Hu et al., 2015). Hence, both the increased freshwater influence through occurrence of sea ice in the Iceland Sea and the open Bering Strait enhanced the Nordic Seas buoyancy. Sensitivity experiments performed with CCSM4 have shown that a closed Canadian Arctic Archipelago entails colder SSTs in the North Atlantic, Norwegian and Iceland Seas relative to the PlioMIP1 experiments where both the Bering Strait and the Canadian Arctic Archipelago were open (Otto-Bliesner et al., 2016). In the idealized experiments a weak cold spatially coherent SST response (less than 2*std of REF-2) is found with weak to intermediate freshwater driven change (P3) in buoyancy. We therefore suggest that the increase

in freshwater reaching the Nordic Seas from the Arctic Ocean may have caused the cold spatial coherent SST anomaly case. We acknowledge that the source and distribution of freshwater is not directly comparable between the Pliocene and the idealized experiments: The freshwater perturbation in the MITgcm experiments is distributed equally over the Nordic Seas basin, while most of the sea ice and liquid freshwater transported from the Arctic Ocean to the Nordic Seas at any time during the Pliocene likely followed the boundary current along the east Greenland margin. Neither does the idealized mode set up include the Arctic gateways nor the Arctic sea ice cover. Hence, we stress again that knowledge about changes in the Bering Strait is used to infer changes in the freshwater balance, and hence buoyancy in the Nordic Seas, and similarly, the occurrence of more or less sea ice in the Iceland Sea is used to infer the likelihood that a buoyancy change took place. The data from Clotten et al., (2019) show however that some of this freshwater also reached the interior Iceland Sea. Less freshwater was available in the region during the Pliocene intervals with warm spatially coherent SST anomalies, also in line with the idealized experiments where a weak warm spatially coherent SST response (less than 2*std of REF-2) is seen for a weak salinity increase (Table 1; Fig. 9b).

The time scale considered for the observations and the Pliocene reconstructions are very different (multidecadal versus hundreds of thousands of years), however, we consider both to reflect equilibrium, or quasi-equilibrium, responses. The time scales involved in either case are longer than the propagation-driven lag that sets up the spatially noncoherent SST anomaly relation at interannual time scales. Compared to the future scenarios which undergo transient changes due to strong $CO_2$ forcing, we regard the era of instrumental observations to be in quasi-equilibrium. The Pliocene reconstructions represent the predominant situation over hundreds of thousands of years. Higher frequency variability did take place superimposed on long term Pliocene SST variability that we focus on, e.g., exemplified by orbital-scale variability visible in the original raw datasets (Fig. 2). Multidecadal variability is however not resolved for any of the relevant Pliocene sites, and the existing age constraints are not good enough to investigate SST anomaly relations at such time scales, or even at orbital scales. While not a focus for this study, we note that the amplitude of the Pliocene SST anomalies, from about 1°C to close to 3°C, iare larger than the observation-based anomalies that are mostly less than 0.5°C (Fig. 6 and 89). The difference in amplitude of the observed and Pliocene anomalies may in part be influenced by the different time scales.

### 4.2 An Iceland Sea SST anomaly different from the North Atlantic and the Norwegian Sea

A warm anomaly in the Iceland Sea corresponding with no change in the North Atlantic and the Norwegian Sea is documented for the Pliocene, between 3.43 and 3.23 Ma (Fig. 8). The idealized experiments show that buoyancy changes due to a weak atmospheric warming under constant freshwater forcing (G1 to REF-1 and G2 to REF-3) cause a warm Iceland Sea SST anomaly that differs from the North Atlantic and the Norwegian Seas, where strictly speaking no change is seen (Table 1; Fig. 9a and c). Pliocene air temperatures for the Nordic Seas domain are largely unknown. No information exists from Iceland (Verhoeven et al., 2013). Over Norway, colder air temperature is indicated between 3.43 and 3.23 Ma, mirroring the colder Norwegian Sea SSTs (Table 2) (Panitz et al., 2018). The data from Panitz et al. (2018) thus suggest a cooling over the Norwegian Sea, in contrast to how the idealized experiments result in a warm anomaly in the Iceland Sea corresponding with no change in the North Atlantic and the Norwegian Sea as a response to a weak warming over the full Nordic Seas. Available data therefore cannot confirm that buoyancy change due to warmer atmospheric temperatures over the Nordic Seas may have caused the Iceland Sea SST anomaly to differ from the North Atlantic and the Norwegian Sea.

The idealized experiments that reproduce this warm Iceland Sea / cold North Atlantic and Norwegian Sea SST anomaly relation are associated with a reduction in all selected output parameters relative to the respective reference experiment: the density gradient between the northern and southern basin, velocity of the inflow over the sill, velocity of the Norwegian Sea boundary current, heat transport over the sill and in the maximum overturning streamfunction at the sill. Existing Pliocene data indicate a weakened overturning circulation (smaller %NADW contribution) relative to the Pliocene mean between 3.43 and 3.23 Ma (Table 2), despite the Norwegian Sea being well ventilated down to intermediate depths (Risebrobakken et al., 2016) (Table 2). In line with a weakened overturning circulation, a strong equator-to-pole North Atlantic meridional SST gradient existed, due to the relatively cold North Atlantic and Norwegian Sea (Table 2; Fig. S1). While the Arctic Sea ice cover was still smaller than today (Knies et al., 2014), constant presence, rather than appearance, of $IP_{25}$ in the Iceland Sea (Clotten et al., 2018) suggests more available freshwater in the form of spring sea ice, relative to the intervals with spatially coherent SST anomalies (Table 2). There was however still much less sea ice than today. While not directly comparable, the overall reduction of the overturning at the sill in the idealized experiment seems consistent with a reduced %NADW for this period in the Pliocene (Table 2). The presence of seasonal sea ice, or sea ice transported to the Iceland Sea, suggests a slight freshening, and thereby strengthened stratification in the Iceland Sea. Such a change in stratification may lead to reduced dense water formation and may thereby weaken the NADW formation. The existence of sea ice in the Iceland Sea during the warm Pliocene suggests somewhat enhanced seasonal contrasts. Tests of different model sensitivities and freshwater forcing scenarios, using LOVECLIM, have shown the AMOC could be more sensitive to freshwater forcing under warm interglacial climate states with large seasonal contrasts for warm climate states with a weaker seasonal contrast (Blaschek et al., 2015). However, Blaschek et al. (2015) saw such an AMOC response only when freshwater reached the Labrador Sea convection site, not for future scenarios where less freshwater/sea ice is likely to impact the Labrador Sea. For parts of this Pliocene interval, 3.43-3.23 Ma, some seasonal sea ice existed in the Labrador Sea (Clotten, 2017). Hence, the results of Blaschek et al. (2015) shows the same direction of AMOC change as seen in the idealized experiments and from Pliocene data for the warm Iceland Sea cold North Atlantic and Norwegian Sea SST anomaly relation driven by a buoyancy change due to weak atmosphere warming. A weakened ocean circulation may again bring less heat and salt into the Norwegian Sea and by continuation the Iceland Sea, further strengthening the stratification.

In the idealized experiments, a noncoherent Iceland Sea SST anomaly scenario is also obtained through strongly increased freshwater driven buoyancy forcing under constant atmospheric temperatures (P4 to REF-2). However, the resulting response is then a cold SST anomaly in the Iceland Sea and warm anomalies in the North Atlantic and the Norwegian Sea, the opposite situation of what is seen by the Pliocene reconstructions (Fig. 6, 7 and 8).

Hence, from the idealized experiments we manage to set up the warm Iceland Sea / cold North Atlantic and Norwegian Sea anomaly scenario. However, the link between the reconstructed background climate and oceanography and the comparable output parameters from the idealized experiments is not straightforward, or relevant data do not exist, emphasizing the need for more data for further evaluation.

### 4.3 A North Atlantic SST anomaly different from the Norwegian and Iceland Seas

A positive SST change (warming) in the Norwegian and Iceland Seas corresponding with a small (close to zero) negative SST change (cooling) in the North Atlantic is seen at the end of the century (2068-2098) in CMIP6 future projections, depending on the scenario used (Fig. 7). NorESM2-MM is the least sensitive of the models, and for that model there are still spatially coherent SST anomalies at the end of the century for the SSP126 experiment, showing that the sensitivity

of the model plays a role. The SST anomaly relation with the Iceland Sea and Norwegian Sea being different from the North Atlantic found in CMIP6 projections is part of a transient response to an imposed $CO_2$ forcing. Strictly speaking this case is therefore not representative for an equilibrium or quasi-equilibrium situation, as discussed for all the other SST anomaly relations.

A similar SST anomaly relation, with the North Atlantic being different from the Norwegian and Iceland Seas, emerges in the idealized experiments for an intermediate to strong atmospheric warming over the Nordic Seas, under constant freshwater (Table 1 and Fig. 910; G2-G4 to REF-1 and G2 to REF-3). Changing the buoyancy by increasing the air temperature over the Nordic Seas reflects what may happen when the atmospheric $CO_2$ concentrations increase in the CMIP6 scenarios; Arctic amplification will entail a larger temperature change over the Nordic Seas than over the North Atlantic.

All these indealized experiments where buoyancy is changed by increasing the air temperature over the Nordic Seas are associated with a reduction in the heat transport over the sill, the maximum overturning streamfunction at the sill, the velocity of the Norwegian Sea boundary current and the density gradient between the northern and southern basin, relative to the respective reference experiments. The AMOC is also weakened compared to the historical level for all CMIP6 scenarios (Weijer et al., 2020) (Table 2). While global SSTs are overall higher for the SSP126 scenario (Table 2), the subpolar North Atlantic cool, and freshens (Kwiatkowski et al., 2020). For SSP245, increased export of liquid freshwater is seen in the Fram Strait (Zanowski et al., 2021).

The same SST anomaly relation as found in the CMIP6 future scenario runs were previously identified for CMIP5 future projections (Alexander et al., 2018; Nummelin et al., 2017) and in the Grand Ensemble of the MPI-ESM1.1 climate model (Keil et al., 2020). Keil et al. (2020) show that the heat import to the North Atlantic, associated with a weakening of the low latitude AMOC, decreases consistently, in parallel with an increased heat transport over the sill set up by a corresponding change in the high latitude overturning and the subpolar gyre circulation, in line with Alexander et al. (2018) and Nummelin et al. (2017). As described above, our idealized experiments show that the SST anomaly relation, with the North Atlantic being different from the other two regions, is associated with a reduced heat transport over the sill, which appears to be opposite to the results from simulated future projections. We stress again that the result from the idealized experiments refers to equilibrium conditions whereas the CMIP5/6 future projections show transient changes to enhanced $CO_2$ forcing, and thus, have not reached an equilibrium. This is exemplified by the results from Keil et al. (2020), where the ocean heat transport over the sill is first increasing and then slightly decreasing. Therefore, we find it hard to conclude on what is causing this SST anomaly relation.

Independent of the exact cause, the indirect effects of both a strong increase in atmospheric $CO_2$ concentrations (CMIP5/6 future projections) and increased atmospheric temperatures over the Nordic Seas (idealized experiments) may break the expectation of spatially coherent SST anomalies and set up a noncoherent SST anomaly relation between the North Atlantic (cold) and the Norwegian and Iceland Seas (warm).

This SST anomaly relation is not seen during the investigated Pliocene time interval. However, we find it interesting that despite the large differences in time scales involved, multidecadal versus hundreds of thousands of years, the strongest SST anomaly seen in the future (up to 2°C between 2068 and 2098 for the Norwegian and Iceland Seas in CNRM-ESM2-1; Fig. 7) is comparable to the amplitude of the Pliocene SST anomalies (Fig. 8), both larger than the amplitude of the

anomalies seen in the instrumental observations (less than 0.8°C; Fig. 6). The amplitude of the future changes depends on both the chosen model and scenario; these large anomalies are seen for SSP126 in CNRM-ESM2-1 and for SSP585 in NorESM. The $CO_2$ forcing of SSP126 (445 ppm by 2100) is comparable to the high end of the Pliocene $CO_2$ range (427 ppm) (Meinshausen et al., 2020; De La Vega et al., 2020), suggesting that the amplitude of the SST anomalies of the North Atlantic and the Nordic Seas is set by the atmospheric $CO_2$ level and that the response time may be as short as within a century.

**4.4 A Norwegian Sea SST anomaly different from the North Atlantic and the Iceland Sea**

At two times during the Pliocene, 3.63-3.93 Ma and 4.73-4.93 Ma, the Norwegian Sea SST anomaly differed from both the Iceland Sea and the North Atlantic (Fig. 8). Between 4.73 and 4.93 Ma a cold SST anomaly was seen in the Norwegian Sea, contemporaneous with a warm anomaly in the Iceland Sea and the North Atlantic. The later period, 3.63 to 3.93 Ma, represents the opposite situation, with a warm Norwegian Sea anomaly corresponding to a cold SST anomaly in the Iceland Sea and the North Atlantic (Fig. 8). None of our idealized experiments resulted in a SST anomaly relation where the Norwegian Sea is different from the other two regions.

During both these Pliocene periods, the %NADW was close to the Pliocene mean strength (Table 2; Fig. S2). For both cases the intermediate depth Norwegian Sea ventilation was stronger than in the periods with spatially coherent SST anomalies, and even stronger when the Norwegian Sea was warm, relative to the Pliocene mean (Table 2). The main difference between the cold (4.73 to 4.93 Ma) and warm (3.63 to 3.93 Ma) Norwegian Sea anomaly situations was that more freshwater entered the Nordic Seas from the Arctic Ocean in the warm scenario, following the opening of the Bering Strait, and that traces of sea ice were present in the Iceland Sea (Table 2). In addition, the Norwegian Sea intermediate and upper water column ventilation was stronger during the warm anomaly situation.

Since none of our idealized experiments show this SST anomaly relation we look further into this case from a conceptual point of view. Two main oceanographic features hold the potential to change the expected spatial coherence at equilibrium between the three locations; a change in ocean circulation and advective pathways that weaken the advective interlinkage, or a change in water column stratification in one of the regions, e.g., related to an anomalous influx of surface freshwater. A change in the advective pathway could for example be the case if the Iceland Sea is more under the influence of the Irminger Current bringing Atlantic water to the Iceland Sea directly through the Denmark Strait rather than the water eventually reaching the Iceland Sea via the Norwegian Atlantic Current. A change in the water column stratification, e.g., in the Iceland Sea, may take place if the surface water in the Iceland Sea is more under influence of the East Greenland Current and thus the Polar domain.

Following this conceptual framework, a cold SST anomaly in the Norwegian Sea corresponding with a warm North Atlantic and Iceland Sea SST anomaly may result from a weakened Norwegian Atlantic Current compensated by a strong Irminger Current. The dominant advective influence of the Nordic Seas is the eastern inflow via the Norwegian Atlantic Current. However, and even if anomalies tend to persist throughout the Nordic Seas advective loop, the water that at the end of this loop travels south via the Greenland and Iceland seas will qualitatively be cold, as the water flowing out of Nordic Seas through the Danmark Strait (e.g., Eldevik et al. 2009; Eldevik and Nilsen 2013). The Irminger Current, on the other hand, is a warm inflow directly influencing the Iceland Sea, where it largely overturns locally to overflow through the Denmark Strait where it entered (e.g., Våge et al. 2011). A stronger Irminger Current inflow can thus be expected to leave an anomalous warm signature in the Iceland Sea – simply more warm water brought directly into the

mix – independent of the anomalous state of the Norwegian Sea and the Norwegian Atlantic Current. Based on existing information we cannot verify if this was the case, or not, between 4.73 and 4.93 Ma.


In contrast, the existence of a warm anomaly in the Norwegian Sea corresponding with cold anomalies in the subpolar North Atlantic and in the Iceland Sea could in theory result from a strengthened or expanded East Greenland Current, increasing the fraction of cold polar water reaching the Iceland Sea (Rudels et al., 2005) and the North Atlantic (Dickson et al., 1988). Admittedly, the effect would need to be quite substantial to affect the North Atlantic proper (this is

nevertheless what is implied by the common attribution of the hydrographic impact of the "Great Salinity Anomaly"; (Dickson et al. 1988). In general, the state of the subpolar North Atlantic tends to relate more to the larger-scale forcing or subtropical–subpolar gyre features (e.g., Hátún et al. 2005; Reverdin 2010), which in this case would be aligned to leave the region anomalously cold. The existence of sea ice in the Iceland Sea (Clotten et al., 2019), combined with the effect of enhanced freshwater supply from the Arctic as a response to the fully opened Bering Strait (De Schepper et al.,

2015) (Table 2), may lend support to the occurrence of an strengthened or expanded East Greenland Current at the time of a warm Norwegian Sea SST anomaly and cold North Atlantic and Iceland Sea SST anomalies. This interval was quite similar to the cold spatial coherence situation (Section 4.1; Table 2), except that the cold spatial coherence case was associated with a higher %NADW and weaker ventilation of the upper Norwegian Sea water column. We suggest that changes in the Arctic freshwater balance, and consequently a strengthened East Greenland Current and/or stratification

of the Iceland Sea, may be a likely scenario for the cold Iceland Sea and North Atlantic and warm Norwegian Sea anomaly case.

In the MITgcm setup used, buoyancy is changed by a freshwater change evenly spread over the Nordic Seas and the ocean currents in the interior basin are not very well represented (e.g., the Irminger Current). Combined, this may explain why

we do not detect this SST anomaly relation through the idealized experiments. Alternatively, an even stronger buoyancy change might be needed to set up a similar response for the idealized experiments.

**5 Summary and future avenues**

Through our analysis of observation-based data (from year 1870 AD), CMIP6 projections of the next century and Pliocene

SST reconstructions covering the time interval between 5.23 and 3.13 Ma, we have identified four SST anomaly relations between the North Atlantic, Norwegian and Iceland Seas SSTs (Fig. 10): 1) Spatially coherent SST anomalies (observations and Pliocene; warm and cold spatially coherent anomalies). 2) The Iceland Sea SST anomaly being different from the North Atlantic and the Norwegian Sea (Pliocene; warm Iceland Sea and cold North Atlantic and Norwegian Sea). 3) The North Atlantic SST anomaly being different from the Norwegian and Iceland Seas (future scenarios; cold

North Atlantic and warm Norwegian and Iceland Seas). 4) The Norwegian Sea SST anomaly being different from the North Atlantic and the Iceland Sea (warm and cold Norwegian Sea corresponding with a cold and warm North Atlantic and Iceland Sea, respectively).

We show that a spatially noncoherent SST anomaly relation can exist in the low (ssp126) and intermediate (ssp585) future

emission scenarios and are not limited to just the high emission scenarios as previously reported. Whether the spatially noncoherent SST anomaly relation is seen in the low emission scenario or not is, however, dependent on the model's equilibrium climate sensitivity; for the least sensitive model, NorESM2-MM the spatially noncoherent SST anomaly relation is not seen in the ssp126 scenario but are seen in the ssp585 scenario. For the ssp126 scenario the spatially noncoherent SST anomaly is most pronounced in the most sensitive model (CNRM-ESM2-1).


Furthermore, we show that occurrence of spatially noncoherent SST anomaly pattern is not limited to the transient nature of the future scenario runs. Different spatially noncoherent SST anomaly relations occurred during the Pliocene, when the background climate is considered to have been in equilibrium with a $CO_2$ forcing comparable to the present atmospheric concentrations and the SSP126 scenario. While the documented SST anomaly relations take place over a

range of different time scales, the SST anomaly relations based on observation and reconstructions, as well as the idealized experiments, represent equilibrium, or quasi-equilibrium, situations. The future change is in that sense the odd case, reflecting a transient response to a given $CO_2$ forcing.

The idealized MITgcm experiments, set up to investigate the impact of buoyancy forcing, reproduce three out of four of

the documented SST anomaly relations, emphasizing the key role of buoyancy for setting the northern North Atlantic SST anomalies (Table 1, Fig. 9 and10). Spatially coherent SST anomalies are seen as a response to a weak to intermediate freshwater induced changes in buoyancy forcing under constant atmospheric temperatures. As the buoyancy forcing, either induced by a SAT or freshwater change, increases, the Iceland Sea SST anomaly becomes different from the North Atlantic and the Norwegian Sea. Under even stronger buoyancy forcing, both the Norwegian and Iceland Seas SST

anomalies are different from the North Atlantic. The situation with a warm SST anomaly in the Iceland Sea and cold anomalies in the North Atlantic and Norwegian Sea is the result of a weak atmospheric warming over the Nordic Seas. A stronger atmospheric warming over the Nordic Seas sets up the cold North Atlantic / warm Norwegian and Iceland Seas scenario.

Based on the idealized experiments and a literature review of existing Pliocene data, we find that spatially coherent SST anomalies are the norm relative to the mean background climate state under weak buoyancy forcing. The idealized experiments suggest that the situation where the Iceland Sea SST anomaly differs from the Norwegian Sea and the North Atlantic likely reflects a response to a weak increase in atmospheric temperatures under constant freshwater forcing. However, the existing data and data coverage are not good enough to verify this statement. The situation where the North

Atlantic SST anomaly differs from the Norwegian and Iceland Sea occur in association with warmer air temperatures caused by increased atmospheric $CO_2$ concentrations.

The case when the Norwegian Sea SST anomaly is different from the North Atlantic and the Iceland Sea, observed for two Pliocene intervals, cannot be explained by the idealized experiments. Here we suggest that a weakened NwAC

compensated by a strong Irminger Current in theory could set up a cold Norwegian Sea anomaly at the same time as the North Atlantic and Iceland Sea experienced a warm SST anomaly. The opposite situation, with a warm Norwegian Sea and cold North Atlantic and Iceland Sea SST anomalies, may be linked to an expanded East Greenland Current, increasing the fraction of cold polar water reaching the Iceland Sea. Both more data and further sensitivity studies are needed to settle the discussion on this specific SST anomaly relation.


The amplitude of the SST variability is overall larger during the Pliocene than in the observational record. The largest amplitude seen at the end of the 21st century in the Norwegian and Iceland Seas is more comparable to the amplitude of the Pliocene SST variability. Why this is the case is out of the scope of this paper and will need to be explored further in future studies. However, since both the Pliocene reconstructions and the future change occur under atmospheric $CO_2$

concentrations around 400 ppm or higher, these results suggest that the amplitudes of SST anomalies in the Nordic Seas depend on the radiative forcing. Furthermore, the results suggest that within the timeframe of the future scenarios SST

anomalies can reach amplitudes comparable to the SST anomaly amplitudes seen for Pliocene periods lasting hundreds of thousands of years and in equilibrium with the $CO_2$ forcing, emphasising how rapidly the earth system can react to increasing atmospheric $CO_2$ concentrations.


Building on this study, it would be interesting to do similar analyses of the SST anomalies over the investigated region for the last millennia, to see if the pattern documented for the observational record holds for a longer historical periods at a preindustrial $CO_2$ level. Furthermore, it would be of interest to do a series of sensitivity studies testing the effects of changing the winds over the region, since wind is the other main factor affecting the inflow to the Norwegian Sea.

However, the fact that we can explain most of the observed spatial SST patterns emphasises that buoyancy plays a key role for northern North Atlantic SST variability across the multiple time scales investigated.

**Acknowledgement**

We acknowledge support from the Centre for Climate Dynamics at the Bjerknes Centre for Climate Research and RCN

projects No 221712 and 229819. The model simulations were performed on resources provided by UNINETT Sigma2 - the National Infrastructure for High Performance Computing and Data Storage in Norway, project NN9709K. The CMIP6 analysis is part of the IS-ENES3 project that has received funding from the European Union's Horizon 2020 research and innovation program under grant agreement No 824084.

**Code/Data availability**
All reconstructions are previously published. Information on availability is given in the Supplement.
HadlSST data are available from
The CMIP6 runs analysed are available through
The MITgcm runs will be made available upon acceptance of the manuscript.


**Author contribution**
BR initiated the work, did the Pliocene data analysis, and led the work and the writing. All were involved in discussions setting the direction of the manuscript, discussions along the way and have provided input and feedback to the text. MFJ was responsible for the MITgcm experiments. HRL was responsible for the HadlSST and CMIP6 analyses. TE developed

the conceptual explanations.

**Competing interests**
We declare no competing interests.

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
