# Peer review of "Buoyancy forcing: A key role for northern North Atlantic SST variability across multiple time scales"

_EGUsphere, 2022_

## Referee Comment (RC1)

Review of the manuscript entitled "The role of buoyancy forcing for northern North Atlantic SST variability across multiple time scales"

The authors describe in the introduction that they aim to:
1) "document existing SST anomalies and phase relations in observational data, CMIP6 Shared Socioeconomic Pathways (SSP)126 experiments and Pliocene alkenone SST reconstruction from the North Atlantic, Norwegian and Iceland Seas."
2) "Address why different SST phase relations may emerge and exist across different climate states, time scales and forcing scenarios."
3) "Investigate impacts of changes in buoyancy forcing on the phase relationship between SSTs in the North Atlantic, Norwegian Sea, and Iceland Sea."

Combining information from past, present and future climates, and paleoclimate data, observational data and idealized ocean model simulations; and use all of that that to improve our understanding of spatial patterns of climate change across a wide range of temporal scales is a courageous undertaking with potentially large impacts. However, in my view the authors currently don't fully succeeded to logically combine all this information or derive underlying mechanisms and explanations for the observed patterns.
In the following I will detail all my major, minor and technical comments. I hope my comments will help the authors to improve the manuscript.

**Major comments:**

Underlying hypothesis and research questions:
The starting point of this study seems to be that on time scales longer than the advective time scale of SST anomalies between the three regions under consideration, the SST changes in the three regions should always be in-phase. I suppose this is theoretically true if all else remains equal. If on longer time-scales there are, however, changes in e.g. sea-ice, ocean currents, vertical mixing, boundary conditions etc, then this does not need to be true. Given that there is internal climate and Earth system variability on all possible time-scales plus forced changes, it seems to me that this theoretical starting point will very often not be met. I think this is also what the authors conclude on lines 687-688 "The in-phase situation is the norm relative to the mean background climate state under weak forcing". (Where 'forcing' in this context is to be understood as any change that is external to the coupled system of the northern North Atlantic, the Norwegian Sea and the Iceland Sea; in contrary to forcings external to the climate system in general.) To me it seems that this should in fact be the starting point of this study (for reasons explained above), not a conclusion. This could potentially lead to an overall manuscript structure that is much easer to read and follw. I would like to ask the authors to reflect on this.

Then there are all the cases in which the 'forcing is not weak' (again 'forcing' would be any change that is external to the coupled system of the northern North Atlantic, the Norwegian Sea and the Iceland Sea). Given the wide range of time scales that are considered here, it seems to me that there are potentially many, very different, causes of 'strong forcings' and thus reasons why SST's in the three regions under consideration would not vary in concert. The authors mention a number of them throughout the manuscript (changes in the Bering Strait, Arctic sea-ice export, strength of the different boundary currents in the region, gyre circulation, just to name a few). Is it the aim of this study to pin-point all these different causes for the time-scales that are considered in this study? Or to identify

commonalities between these causes? I think it should be more clearly described what the authors want to learn from the assessment they are presenting.

The third aim of this manuscript, according to the aims as I list them in the top of this review, is to use the MITgcm to study mechanisms underlying the observed 'phase relationship between SSTs'. Sensitivity studies allow one to test different hypothesis. The experiments presented here only focus on surface buoyancy changes. Why is that? Are there indications that on all the time scales under consideration here the drivers of the SST patterns were buoyancy changes? Is that the hypothesis of this study? Most of the explanations that the authors discuss for the observed patterns (changes in AMOC strength, changes in strength of the East Greenland Current, changes in salinity and/or sea-ice export via the East Greenland Current or a change in the connection between the Northern North Atlantic with the Nordic Sea, from the main connection being with the Norwegian Sea to the main connection being with the Iceland Sea) are not captured in the set of sensitivity experiments so I wonder if these experiments are the right ones to test the mechanisms underlying the observed, reconstructed and simulated patterns.

The usage of the terms 'in-phase' and 'out-of-phase':
The terms 'in-phase' and 'out-of-phase' for me are linked to temporal behavior. In the context of 'equilibrium' results (Pliocene time-slices, CMIP simulations and MITgcm results) I find these term rather confusing. If for instance a CMIP model shows cooling for the North Atlantic domain for 2070-2100 and warming in the Norwegian Sea they indeed have a different sensitivity to the forcing, but would you call that 'out-of-phase'? For the description of the observational record, I think usage of the terms 'in-phase' and 'out-of-phase' is appropriate. It is perhaps 'only a wording issue', but this for me confuses the whole concept behind the study. It raises questions like whether the authors mean to say that the in-phase Pliocene SST changes are part of an internal mode of variability? (like the described observational changes). Perhaps use words like 'spatially homogeneous versus spatially heterogeneous'? Coherent versus different?

Data sources:
Aims number 2 and 3, as described at the beginning of this review, are interesting research questions. However, aim number one is not a research question by itself. Describe more clearly why one would like to document SST anomalies and 'phase-relationships" in this particular combination of data-sets. What is the rationale of combining these data-sets?

Line 101: It seems to me that the easiest starting point to test the main hypothesis of this study (on time scales longer than the advective time scale of SST anomalies between the three regions under consideration, the SST changes in the three regions should always be in-phase) would be to look at CMIP/PMIP multi-millennial pre-industrial (control) simulations and past-2k simulations. Instead, future runs are used which introduce all kinds of complexities. Please explain the underlying reasoning.

The comparison with the CMIP6 results at the end of the century with the other data sources is difficult because, as the authors say, they show transient climate change, not equilibrium climate change. So why not include for instance experiments forced with a doubling of CO2, that would provide you with an equilibrium response that is more comparable. What is the added value of investigating the transient climate change at the end of this century? Please explain the underlying reasoning.

Line 101: why only look at Pliocene SST observations while Pliocene model simulations also exist.

Paleoclimate reconstructions:
Are the results for proxies local while the model results are large-scale averages? Does this impact the results? Did the authors test how the model results would look like if only data is used from the grid cells in which the proxy data is found?

Table 2: The information provided here for the Pliocene comes from many different sources. How good are the age constraints? Are they sufficient to assume that all of the presented climatic indicators happened during a single interval as defined by the authors?

MITgcm simulations:
The setup of the MITgcm is focusing on the connection between the North Atlantic, Norwegian Sea and Iceland Sea, but not the connection with the Arctic (sea-ice changes or opening of the Bering Strait). Nonetheless, those mechanisms are found to be important by the authors, questioning whether the setup of the idealized simulations is appropriate for the questions that are asked in this manuscript. Please explain why this approach is taken.

Line 206: why force buoyancy changes on the north of the ridge while the starting point of the manuscript is on the impact of advecting buoyancy anomalies from the region south of the ridge to the region north of the ridge?

Figures 1 and 5: Are the geographical regions used for the Iceland Sea and the Norwegian Sea in figure 1 (and so for the observational results and the CMIP results) comparable to the definition of the Iceland Sea and the Norwegian Sea in figure 5 (and so for the MITgcm results)? In the latter it seems really a comparison between a boundary current (Norwegian Sea) and the ocean interior (Iceland Sea), but is that a good representation of the other observational, modeling and proxy-based reconstruction results?

**Minor comments:**
Line 66: I understand what is being said here, but that is only because of the lines that follow. Please try to describe more clearly what is meant here. 'Relationships' between what? And why does the continuous northward transport of heat imply this?

Lines 208-215: I find the description of the modeling setup very confusing. You start by describing three different reference experiments. Why do we need three different reference experiments and how are they different? And then at some point in this text you continue describing the actual perturbation experiments. Why does the second set of experiments (G1-P1, G1-P2 etc) use G1 and not G0 as SAT forcing? Refer to table 1 at this point as the different experiments are nicely summarized in there, or include a new table to show this information?

Line 304: 'enhances the SST in the Iceland Sea', what does that mean? Please clarify.

Line 354: why would you call an insignificant response 'in phase' and why not simply an insignificant response? I guess there could also be a significant in phase response?

Lines 515-517: The opening of the Bering Strait could indeed play a role the Pliocene cooling over the whole North Atlantic, but what about other potential drivers? Higher CO2 levels for instance, how well are those constrained?

Lines 541-349: Many GCM experiments exist in which sea-ice melt increases and/or NADW is reduced. What do such experiments show? Is that in line with the findings described here for these specific Pliocene intervals? Please discuss.

Lines 576-577: I don't agree that the time-scales are comparable. The changes in the observational record play out on decadal time-scales while in the future runs, we are comparing roughly the period 2000-2050 with the period 2050-2100. Indeed figure 8 shows that on top of this 'long-term' variability, there is also decadal variability similar to the observational record for which the relationships between SST's in the three different regions are again different from what is described in this section (and perhaps more similar to the variability in the observational record). Please clarify.

Line 609: I don't quite understand this part. Do the authors mean that there is an indirect link between the radiative forcing and the 'out-of-phase' relationship between the North Atlantic and the Nordic Seas? And if so, what kind of link would this be?

Lines 632-633: Increased influence of the East Greenland Current can explain a cooling of the Iceland Sea, but how does it explain the corresponding cooling of the North Atlantic?

Lines 635-639: Wouldn't you still need to weaken advection between the Norwegian and Iceland Seas? Otherwise wouldn't warm North Atlantic water enter the Norwegian Sea via the Iceland Sea and still result in a warming in both regions?

Lines 688-690: what is meant here with a 'weakened ocean circulation'? Please clarify.

Lines 705-706: Isn't this what one would expect? That the amplitude of SST changes depends on the radiative forcing?

Figure 1: How are the domains depicted in figure 1 determined and what is their influence on the presented results?

Figure 7: clarify in the figure caption that these are all multi-decadal variations. What does it mean that only the bandpass filtered data are significant and the running mean data are not?

Figure 10: The SST response seems very small in many cases. How significant are these results and how does the magnitude of these responses compare to the magnitude found in the observational results, CMIP models and proxy-based reconstructions?
And on line 241 you mention that the North Atlantic is set to constant, so why are anomalies simulated for that region in figure 10? Clarify in a little more detail how the experimental forcing is defined.

Table 2: Why is information from the observational period, one of the three main periods discussed in this manuscript, not included in this table? Please explain.

**Technical comments:**
Figure 1: include letters a,b and c in the figure. Also I don't see the described blue and red boxes in figure 1. Please clarify.

Figure 6: What do the grey and white vertical bands mean in this figure? The 'multi-decadal' periods over which the anomalies in the inserted maps are calculated?

In general I find figure 6 not a very clear representation of the different in-phase and anti-phase relationships on interannual-to-decadal and on multi-decadal time-scales as is described on lines 258-263.

Figure 8B: why no inserted maps for the middle two models?

Table 1: The experiment name that is given in this table (exp) seems different from what is used in the main text and in figure 10. Consider changing for clarity.

---

## Author Comment (AC1)

**Responses to Reviewer 1 – Pepijn Bakker**

Below you will find our responses to Pepijn Bakker's review of our manuscript "The role of buoyancy forcing for northern North Atlantic SST variability across multiple time scales", egusphere-2022-959. His comments are provided in blue while our responses are shown in black, directly following the individual reviewer comments.

We appreciate his positive view on the overall goal of the paper and his very thorough, constructive, and useful feedback that clearly will help improve the manuscript. We acknowledge the main points about coherence, terminology etc. of the manuscript and will put major efforts into clarifying these.

Best regards,
Bjørg Risebrobakken, on behalf of all co-authors

Review of the manuscript entitled "The role of buoyancy forcing for northern North Atlantic SST variability across multiple time scales"

The authors describe in the introduction that they aim to:
1) "document existing SST anomalies and phase relations in observational data, CMIP6 Shared Socioeconomic Pathways (SSP)126 experiments and Pliocene alkenone SST reconstruction from the North Atlantic, Norwegian and Iceland Seas."
2) "Address why different SST phase relations may emerge and exist across different climate states, time scales and forcing scenarios."
3) "Investigate impacts of changes in buoyancy forcing on the phase relationship between SSTs in the North Atlantic, Norwegian Sea, and Iceland Sea."

Combining information from past, present and future climates, and paleoclimate data, observational data and idealized ocean model simulations; and use all of that that to improve our understanding of spatial patterns of climate change across a wide range of temporal scales is a courageous undertaking with potentially large impacts. However, in my view the authors currently don't fully succeeded to logically combine all this information or derive underlying mechanisms and explanations for the observed patterns.

In the following I will detail all my major, minor and technical comments. I hope my comments will help the authors to improve the manuscript.

**Major comments:**

Underlying hypothesis and research questions:
The starting point of this study seems to be that on time scales longer than the advective time scale of SST anomalies between the three regions under consideration, the SST changes in the three regions should always be in-phase. I suppose this is theoretically true if all else remains equal. If on longer time-scales there are, however, changes in e.g. sea-ice, ocean currents, vertical mixing, boundary conditions etc, then this does not need to be true. Given that there is internal climate and Earth system variability on all possible time-scales plus forced changes, it seems to me that this theoretical starting point will very often not be met. I think this is also what the authors conclude on lines 687-688 "The in-phase situation is the norm relative to the mean background climate state under weak forcing". (Where 'forcing' in this context is to be

understood as any change that is external to the coupled system of the northern North Atlantic, the Norwegian Sea and the Iceland Sea; in contrary to forcings external to the climate system in general.) To me it seems that this should in fact be the starting point of this study (for reasons explained above), not a conclusion. This could potentially lead to an overall manuscript structure that is much easier to read and follow. I would like to ask the authors to reflect on this. Then there are all the cases in which the 'forcing is not weak' (again 'forcing' would be any change that is external to the coupled system of the northern North Atlantic, the Norwegian Sea and the Iceland Sea). Given the wide range of time scales that are considered here, it seems to me that there are potentially many, very different, causes of 'strong forcings' and thus reasons why SST's in the three regions under consideration would not vary in concert. The authors mention a number of them throughout the manuscript (changes in the Bering Strait, Arctic sea-ice export, strength of the different boundary currents in the region, gyre circulation, just to name a few). Is it the aim of this study to pinpoint all these different causes for the time-scales that are considered in this study? Or to identify commonalities between these causes? I think it should be more clearly described what the authors want to learn from the assessment they are presenting.

We thank the reviewer for making us aware of the need to clarify the aim of the study, the hypothesis to be tested and the need to look carefully at the phrasings to avoid misunderstandings. We will put major efforts into addressing these issues.

We have considered the suggestion to change the starting point of the study from expected coherency to expected non coherency. We will, however, like to keep the observational record as our starting point and to set the expectation based on how oceanography of the focus region is known from observations. To do so, we want to focus and tighten the introduction to express more clearly what we want to do and why, including the reasoning for the choices made.

We agree that on long time scales, many factors, e.g. changes in continental configuration, may impact the large scale oceanographic features. Keeping that in mind, we see that how we have used the background knowledge from literature (Table 2) might be confusing. We will go carefully through the text to clarify how we use the information given. While many of the factors mentioned in Table 2 can be seen as forcing in themselves, they are more often used to infer how buoyancy may have been different under given conditions. E.g. sea ice may work as a force directly, however, more sea ice can also be seen as an indicator for more freshwater in the system, and hence a buoyancy change.

Furthermore, we will go carefully through the text with the aim to pick up other potential unclear statements. One example of how the text will benefit from clarifications is the statement mentioned above in original lines 687-688; here we should refer to buoyancy forcing specifically (as the sentence refers to the MITgcm results), not any change to the earth system. This has now been specified.

The third aim of this manuscript, according to the aims as I list them in the top of this review, is to use the MITgcm to study mechanisms underlying the observed 'phase relationship between SSTs'. Sensitivity studies allow one to test different hypothesis. The experiments presented here only focus on surface buoyancy changes. Why is that? Are there indications that on all the time scales under consideration here the drivers of the SST patterns were buoyancy changes? Is that the hypothesis of this study?

Yes, we hypothesise that buoyancy plays a key role at all the time scales considered. We will make a clear statement in the introduction about why we focus on buoyancy.

Two key factors impact the inflow of Atlantic Water from the North Atlantic, over the Greenland Scotland Ridge and into the Nordic Seas; wind and buoyancy. From a present day physical oceanography point of view, buoyancy is considered most important at multidecadal and longer time scales, while wind is more important at interannual to decadal time scales. We acknowledge that wind may still be important, however, in this study we have chosen to focus on buoyancy. We expected buoyancy changes to be of major importance and wanted to see to what degree the observed SST patterns could be explained solely through changing buoyancy. While all cases may not be explained through these experiments, most cases can, emphasising the importance of buoyancy.

In the revised manuscript we will stress that wind may also impact the inflow, even though it is not a focus of this study to investigate the effects of wind changes. It could be an interesting future study to test the potential effects of changes in wind forcing on SST patterns. We may add a sentence on this in the outlook section.

Most of the explanations that the authors discuss for the observed patterns (changes in AMOC strength, changes in strength of the East Greenland Current, changes in salinity and/or sea-ice export via the East Greenland Current or a change in the connection between the Northern North Atlantic with the Nordic Sea, from the main connection being with the Norwegian Sea to the main connection being with the Iceland Sea) are not captured in the set of sensitivity experiments so I wonder if these experiments are the right ones to test the mechanisms underlying the observed, reconstructed and simulated patterns.

Thanks for making us aware of the need to clarify the purpose of Table 2 and how we use the information given there. We do not intend to investigate all possible factors that may have an impact. Rather, we wanted to provide background information on factors that may have had an impact on buoyancy over the region at a given time, or factors that may compare to responses seen in the idealised experiments to changes in buoyancy. We will consider the phrasings used throughout, to avoid misunderstandings on how we consider the relevant information at any point in the manuscript. We acknowledge that some of these factors may work as forces themselves, and this will be clarified.

The usage of the terms 'in-phase' and 'out-of-phase':
The terms 'in-phase' and 'out-of-phase' for me are linked to temporal behavior. In the context of 'equilibrium' results (Pliocene time-slices, CMIP simulations and MITgcm results) I find these term rather confusing. If for instance a CMIP model shows cooling for the North Atlantic domain for 2070-2100 and warming in the Norwegian Sea they indeed have a different sensitivity to the forcing, but would you call that 'out-of-phase'? For the description of the observational record, I think usage of the terms 'in-phase' and 'out-of-phase' is appropriate. It is perhaps 'only a wording issue', but this for me confuses the whole concept behind the study. It raises questions like whether the authors mean to say that the in-phase Pliocene SST changes are part of an internal mode of variability? (like the described observational changes). Perhaps use words like 'spatially homogeneous versus spatially heterogeneous'? Coherent versus different?

We see that using the terms in and out of phase may cause confusion and will change the wording throughout the manuscript to spatially coherent and dissimilar or noncoherent.

Data sources:
Aims number 2 and 3, as described at the beginning of this review, are interesting research questions. However, aim number one is not a research question by itself. Describe more clearly

We totally agree that to document the anomalies is not a research question. This sentence was included to describe the different steps of the paper and is in that sense just a description of what we do to address what type of SST anomaly patterns that exist in the selected data sets. We will delete this sentence to avoid further confusion.

The reason for looking into the future projections is twofold:
1. The spatially incoherent SST pattern, with a cold North Atlantic/warm Nordic Seas was detected in the RCP 8.5 CMIP5 runs but is not fully understood. As specified in the introduction, we wanted to see if this still holds in the lower emission scenario CMIP6 runs. Our results show that this is the case for the most sensitive model.
2. At the end of the century the $CO_2$ concentration of the ssp126 scenario is in the range of the Pliocene $CO_2$ level. Therefore, we wanted to look at the long-term SST pattern of the Pliocene, to see how long-term equilibrium SST patterns compares to the short-term transient response to a comparable atmospheric $CO_2$ concentration.

When the work started out, we did consider looking at the longer historical runs in addition to the future runs, however, at that time the available runs were few and from different models than the ones we had selected for the analysing the projections. Since it would not have been possible to keep consistency between analysed models' consistency and to avoid adding even further complexity to an already complex study we decided not to continue in this direction.

Part of our motivation came from how the high emission scenarios seemed to break the expectation of spatially coherent SST anomalies over the North Atlantic, Norwegian and Iceland Seas. The question that came up then was if this signal was restricted to the high emission scenarios or if similar responses could be seen under lower emission scenarios, or in a past comparable climate state.

We see that a logical continuation of this study might be to look at the CMIP/PMIP multi-millennial pre-industrial (control) simulations and past-2k simulations and will add this as a potential future outlook in the summary section of the paper.

We appreciate the suggestion to look at a 2*CO2 equilibrium response rather than the CMIP6 scenario experiments. As mentioned above, part of our motivation came from how the high emission scenarios seemed to break the expectation of spatially coherent SST anomalies over the North Atlantic, Norwegian and Iceland Seas, as set by the observational record. The question that came up then was if this signal was restricted to the high emission scenarios or if

similar responses could be seen under lower emission scenarios. The next question then is if the response relates to the transient nature of the ssp experiments, or if similar spatially noncoherent SST anomalies also existed during a past climate state with comparable CO2 but in equilibrium. We could not have been able to address these questions by a doubling of CO2 experiment.

Line 101: why only look at Pliocene SST observations while Pliocene model simulations also exist.

When we started out, we did consider the possibility to look at PlioMIP experiments as well. However, in the end we decided not to investigate the Pliocene experiments. The reason why we did not analyse PlioMIP simulations is that they are time slice experiments. Analysing those could have given us the SST anomalies relative to pre-industrial, while the SST reconstructions (and the SST anomalies of the other time intervals) are all looked at as the SST anomalies relative to the mean of the respective investigated time series. Hence, the results would not have been comparable to our other results.

Paleoclimate reconstructions: Are the results for proxies local while the model results are large-scale averages? Does this impact the results? Did the authors test how the model results would look like if only data is used from the grid cells in which the proxy data is found?

The proxy records provide information from the individual sites, however, the available sites are from locations considered representative for the area that they represent. If any impact on the results when comparing with the model results that are averaged over a larger area, one may expect the proxy results to show a somewhat larger amplitude of change. We have by purpose avoided looking at the model results from individual grid points. It is not given that the grid points surrounding a specific site provides the best representation of that site, or that the representation would be constant between the different models. To address this question from the reviewer we will check the impact of using single grid points versus averaged domain.

Table 2: The information provided here for the Pliocene comes from many different sources. How good are the age constraints? Are they sufficient to assume that all of the presented climatic indicators happened during a single interval as defined by the authors?

The shortest interval we look at lasts for 200 000 years and we only look at the mean state of the individual intervals. Looking at the mean conditions over such long time intervals, we do consider the age constraints to be sufficiently good for the use. Pliocene chronologies are mostly constrained by tuning to LR04 and/or using tie points from magnetic reversals. The tuning error is generally considered to be no more than a few thousand years but may exceed 10 ka prior to 4.3 Ma due to a less certain obliquity variance (Lisiecki and Raymo, 2005). If we had looked at shorter orbital time scales the 10 kyr uncertainty would have been an issue. We will add somewhat more details about this in Section 2.3. When feeding information to Table 2, we are consistent in treating the data in the same way as we have treated the North Atlantic/Norwegian Sea/Iceland Sea records.

MITgcm simulations:

The setup of the MITgcm is focusing on the connection between the North Atlantic, Norwegian Sea and Iceland Sea, but not the connection with the Arctic (sea-ice changes or opening of the Bering Strait). Nonetheless, those mechanisms are found to be important by the authors, questioning whether the setup of the idealized simulations is appropriate for the questions that are asked in this manuscript. Please explain why this approach is taken.

The model set-up is indeed idealized (but with full physical equations), including sea-ice and a possble key gateway missing, However, we have some experience in using the model in

similar set-ups before – jointly to disentangle cause-and-effect and, broadly, regionalities (e.g., Våge et al. 2011; Jensen et el. 2018). And this we have found useful.

The approach has in general been spearheaded by Mike Spall, using it in numerous applications/process studies for the North Atlantic, Nordic Seas, and, in some cases, including the Arctic, on the nature of flow, exchanges, and external forcing (buoyancy, wind). Including making regional inference (with care).

We will clarify better in the revised manuscript how comparison is or can be made between the idealized – ("case study" – model, and both the other data and research questions of our mansusctipt. In particular, we will acknowledge the missing part of sea-ice and a northern gateway, and how that may reflect on our results.

Line 206: why force buoyancy changes on the north of the ridge while the starting point of the manuscript is on the impact of advecting buoyancy anomalies from the region south of the ridge to the region north of the ridge?

This goes to the nature of the approach/the model set-up. The restoring boundary condition in the south is also similarly a forcing – it is both the representation of the (infinite) source of Atlantic Water and the experiment's energy source (heat and buoyancy input). This water mass then experience the surface forcing applied, water mass transformation takes place, and a consistent ocean circulation is set up, incl setting the hydrography of the different regions. (The southern boundary energy input and northern surface heat loss balance when the model have reached (quasi-)equilibrium; they are accordingly equally important for the experiment.)

We will explain this better in the revised version of the manuscript.

Figures 1 and 5: Are the geographical regions used for the Iceland Sea and the Norwegian Sea in figure 1 (and so for the observational results and the CMIP results) comparable to the definition of the Iceland Sea and the Norwegian Sea in figure 5 (and so for the MITgcm results)? In the latter it seems really a comparison between a boundary current (Norwegian Sea) and the ocean interior (Iceland Sea), but is that a good representation of the other observational, modeling and proxy-based reconstruction results?

The reviewer is right that the Norwegian and Iceland Seas domains are slightly different when looking at the output from the MITgcm experiments compared to the CMIP output. Thanks for making us aware that we have missed adding information about this and the reasoning behind. In the revised manuscript we will add information on the reasoning for this choice in Section 2.4. We will also correct the caption for Figure 1. The MITgcm setup is idealized, therefore it is not possible to set exactly the same domains and expect comparability to the observations and CIMIP results.

The reviewer is also correct that MITgcm output is for the Norwegian Sea representative of the boundary current and the Iceland Sea the interior of Iceland basis. This is done as we consider these areas to be best representative for what the proxy data from the core sites represent. Site 642 in the Norwegian Seas is under direct influence of the Norwegian Atlantic Current represented by the eastern boundary current, while Site 907 is from the interior part of the Iceland basin, and hence not within the core of the western boundary current.

We believe regional inference broadly can be made from these experiments, but we will take greater care in outlining the basis/caveats for interpretation and comparison (see also response

above). The reviewer's point regarding highly advective regions (Norwegian Sea) versus more "stagnant" interior basins (Iceland Sea), is a good one.

Minor comments:
Line 66: I understand what is being said here, but that is only because of the lines that follow. Please try to describe more clearly what is meant here. 'Relationships' between what? And why does the continuous northward transport of heat imply this?

We see that this could be unclear and will rephrase the start of this paragraph. Thanks for pointing it out.

Lines 208-215: I find the description of the modeling setup very confusing. You start by describing three different reference experiments. Why do we need three different reference experiments and how are they different? And then at some point in this text you continue describing the actual perturbation experiments. Why does the second set of experiments (G1-P1, G1-P2 etc) use G1 and not G0 as SAT forcing? Refer to table 1 at this point as the different experiments are nicely summarized in there, or include a new table to show this information?

We thank the reviewer here and in general to identify where our overall presentation and lines-of-argument are in need of improvement, incl making specific and constructive suggestions. We will clarify and in particular use Table 1 more actively to structure the presentation of experiments/results as suggested by the reviewer.

Line 304: 'enhances the SST in the Iceland Sea', what does that mean? Please clarify.

We will rephrase this sentence to clarify the content.
In the model version with a lower horizontal resolution in the atmosphere the SST of the Iceland Sea is higher at the end of the century than in the model version with a higher horizontal resolution in the atmosphere. Such an effect is not seen in the Norwegian Sea or the North Atlantic.

Line 354: why would you call an insignificant response 'in phase' and why not simply an insignificant response? I guess there could also be a significant in phase response?

From the MITgcm experiments, all in phase SST changes are less than 2*std of the reference experiment, hence the notation insignificant in-phase. We will consider possible ways to clarify the message, e.g. by marking the inserts differently when the response is considered insignificant.

Lines 515-517: The opening of the Bering Strait could indeed play a role the Pliocene cooling over the whole North Atlantic, but what about other potential drivers? Higher CO2 levels for instance, how well are those constrained?

Increasing or decreasing atmospheric $CO_2$ concentrations will have an impact on the atmospheric temperatures over the region. For example, one might imagine that with arctic amplification a potentially increased atmospheric $CO_2$ concentration may entail a larger temperature change over the Nordic Seas than over the North Atlantic, hence, comparable to idealized experiments relative to REF1 and REF3 where we change the buoyancy forcing by increasing the SAT over the Nordic Seas.

The $CO_2$ level is unfortunately not well enough constrained before ca. 3.5 Ma to evaluate what role $CO_2$ changes might have been for the individual intervals.

 Many GCM experiments exist in which sea-ice melt increases and/or NADW is reduced. What do such experiments show? Is that in line with the findings described here for these specific Pliocene intervals? Please discuss.

Thanks for the suggestion. We will look at the literature and consider how to implement information from such studies before submitting the revised manuscript.

 I don't agree that the time-scales are comparable. The changes in the observational record play out on decadal time-scales while in the future runs, we are comparing roughly the period 2000-2050 with the period 2050-2100. Indeed figure 8 shows that on top of this 'long-term' variability, there is also decadal variability similar to the observational record for which the relationships between SST's in the three different regions are again different from what is described in this section (and perhaps more similar to the variability in the observational record). Please clarify.

We will clarify this point in the revised manuscript.

The SST anomalies of the observational record take place over multiple decades, hence multi decadal time scales. For the future we focus on the conditions at the end of the century, 2068-2098 relative to the mean of the future runs (2023-2098), hence, a 30 year long period, or a multi decadal signal.

For the end of the century (2068-2098), CNRM-ESM2-1 SSP126 and NorESM2-MM ssp585 show the discussed phase relation, with a cold anomaly in the North Atlantic and a warm anomaly in the Norwegian and Iceland Seas. For MPI-ESM1-2-LR SSP126 you cannot really distinguish the different members from each other. Therefore, we did not include the insert there (however, the same cold anomaly in the North Atlantic and a warm anomaly in the Norwegian and Iceland Seas is seen in the member presented in Figure 8b). NorESM2-MM is the least sensitive of the models, and for that model there are still spatially coherent SST anomalies at the end of the century for the SSP126 experiment, showing how the sensitivity of the model plays a role.

 I don't quite understand this part. Do the authors mean that there is an indirect link between the radiative forcing and the 'out-of-phase' relationship between the North Atlantic and the Nordic Seas? And if so, what kind of link would this be?

Thanks for making us aware of this unclear statement. We will rephrase this sentence when we revise the manuscript.

The $CO_2$ forcing of SSP126 is comparable to the high end of the Pliocene $CO_2$ range (Meinshausen et al., 2020; De La Vega et al., 2020), suggesting that the amplitude of the SST anomalies of the North Atlantic and the Nordic Seas are set by the atmospheric CO2 level and that the response time may be as short as within a century.

 Increased influence of the East Greenland Current can explain a cooling of the Iceland Sea, but how does it explain the corresponding cooling of the North Atlantic?

The latter is, admittedly, not trivial from how we present the case. A stronger East Greenland Current – more cold water – could do both. We will elaborate this issue better in the revised manuscript.

 Wouldn't you still need to weaken advection between the Norwegian and Iceland Seas? Otherwise wouldn't warm North Atlantic water enter the Norwegian Sea via the Iceland Sea and still result in a warming in both regions?

We see that this is not very well explained. The argument would go something like this: the AW-source region is in a warm state. It is provided to the Iceland Sea more directly by a stronger Irminger Current, and the former warms. For the Norwegian Sea, fed by the Norwegian Atlantic Current (NwAC), the weakened NwAC decreases the heat input more than what is contributed from the AW source being warmer (the less volume transport, the more cooling from a given heat loss). And evetual influence of the Irminger Current on the Norwegian Sea via the Iceland Sea would be margial compared to the larger and more direct impact of NwAC.

We will elaborate this issue better in the revised manuscript.

Lines 688-690: what is meant here with a 'weakened ocean circulation'? Please clarify.
This relates to similar arguments as outlined in our response above. We will elaborate this issue better in the revised manuscript.

Lines 705-706: Isn't this what one would expect? That the amplitude of SST changes depends on the radiative forcing?
We agree with the reviewer that it may be expected that the amplitude of SST change depends on the radiative forcing. However, we still find it interesting that we see the same amplitude at the end of the century, after quite a short time of increasing CO2, as for Pliocene anomalies representative of hundreds of thousands of years. It emphasises how quickly change may take place. We will add a specification here.

Figure 1: How are the domains depicted in figure 1 determined and what is their influence on the presented results?
The domains are selected to be representative of the three sites where we have Pliocene data. We do consider that SST averaged over the domains to better represent the variability than a single grid point (see reply to comment above). More details on the selection of the domains will be added to section 2.1.

Figure 7: clarify in the figure caption that these are all multi-decadal variations. What does it mean that only the bandpass filtered data are significant and the running mean data are not?
Figure 7 was included to show the anti-phase relations at interannual to decadal scales. We see that it may be confusing the way it was included and we will remove Figure 7 from the revised manuscript. The text will be edited accordingly.

Figure 10: The SST response seems very small in many cases. How significant are these results and how does the magnitude of these responses compare to the magnitude found in the observational results, CMIP models and proxy-based reconstructions?
In Table 1 it is shown which of the MITgcm results are significant. The magnitude of the anomalies for each case is also defined in Table 1. In addition, we will add information about significance in the figure caption of Figure 10. All "red" experiments see significant anomalies in both the Norwegian and Iceland Seas. The "blue" experiments see a significant change in the Norwegian Sea, but not in the Iceland Sea. The "grey" experiments see no significant change in either the Norwegian or Iceland Seas.

The significant SST anomalies seen from the MITgcm experiments are in the range of 0.17 to 0.69°C for the Norwegian Sea and 0.51 to 2.42°C for the Iceland Sea; the insignificant MITgcm responses are 0.0-0.1 and 0.0 to 0.08, respectively (Tabel 1). The mean anomalies as seen in HadlSSTs, CMIP6 models and Pliocene reconstructions are within the same range for the

Iceland Sea. In the Norwegian Sea the mean anomaly exceeds the MITgcm range in the most sensitive CMIP6 model (CNRM-ESM2-1 ssp126), and in four of the six Pliocene cases. We will elaborate in the revised manuscript.

And on line 241 you mention that the North Atlantic is set to constant, so why are anomalies simulated for that region in figure 10? Clarify in a little more detail how the experimental forcing is defined.

We will check this and clarify in the revised manuscript.

Table 2: Why is information from the observational period, one of the three main periods discussed in this manuscript, not included in this table? Please explain.

Related climate change for warm and cold periods in the observational period have been documented previously in studies about AMV. However, in this study, information from the instrumental period is primarily used to set the expectation, and confirm that the result previously shown by others (e.g. Årthun et al., 2017) for specific stations along the pathway of Atlantic Water transport also is valid when looking at the domains that we focus on in this study. We will add an explanation to the text.

Technical comments:

Figure 1: include letters a,b and c in the figure. Also I don't see the described blue and red boxes in figure 1. Please clarify.

The letters a, b and c will be added to the figure. The reference to blue and red boxes was added to inform on how the boxes/defined domains are expressed in following figures. We see that the phrasing may be confusing and will delete this part of the sentence.

Figure 6: What do the grey and white vertical bands mean in this figure? The 'multi-decadal' periods over which the anomalies in the inserted maps are calculated

The grey bars are inserted to highlight the periods with positive spatially coherent SST anomalies. We will add a specification to the figure caption.

In general I find figure 6 not a very clear representation of the different in-phase and anti-phase relationships on interannual-to-decadal and on multi-decadal time-scales as is described on lines 258- 263.

The focus of Fig. 6 is to show in-phase relationships on multidecadal time scales, not the anti-phase relation at shorter time scales. Hence, you should not expect to see the anti-phase reflation. As mentioned above, figure 7 was included for the purpose of showing the anti-phase at interannual to decadal time scales, but to avoid confusion we will delete figure 7 from the revised manuscript and rephrase the text.

Figure 8B: why no inserted maps for the middle two models?

Inserted maps are not shown for the two middle models due to a weak signal and/or that there was not possible to say anything from between the different model members. We will consider adding inserts here as well, with specifications in the caption.

Table 1: The experiment name that is given in this table (exp) seems different from what is used in the main text and in figure 10. Consider changing for clarity.

Thanks for making us aware that this may be a source of confusion. We will consider restructuring the information about the exp names in Table 1.

**References**

de la Vega, E., Chalk, T. B., Wilson, P. A., Bysani, R. P., & Foster, G. L. (2020). Atmospheric CO2 during the Mid-Piacenzian Warm Period and the M2 glaciation. *Scientific Reports*, *10*(1), 11002, https://doi.org/10.1038/s41598-020-67154–67158.

Jensen, M. F., Nisancioglu, K. H., & Spall, M. A. (2018). Large changes in sea ice triggered by small chnages in Atlantic water temperature. *Journal of Climate*, *31*, 4847–4863.

Lisiecki, L. E., & Raymo, M. E. (2005). A Pliocene-Pleistocene stack of 57 globally distributed benthic δ18O records. *Paleoceanography*, *20*, PA1003, doi:10.1029/2004PA001071.

Meinshausen, M., Nicholls, Z. R. J., Lewis, J., Gidden, M. J., Vogel, E., Freund, M., Beyerle, U., Gessner, C., Nauels, A., Bauer, N., Canadell, J. G., Daniel, J. S., John, A., Krummel, P. B., Luderer, G., Meinshausen, N., Montzka, S. A., Rayner, P. J., Reimann, S., … Wang, R. H. J. (2020). The shared socio-economic pathway (SSP) greenhouse gas concentrations and their extensions to 2500. *Geosci. Model Dev.*, *13*, 3571–3605, https://doi.org/10.5194/gmd-13-3571–202.

Våge, K., Pickart, R. S., Spall, M. A., Valdimarsson, H., Jónsson, S., Torres, D. J., Østerhus, S., & Eldevik, T. (2011). Significant role of the North Icelandic Jet in the formation of Denmark Strait overflow water. *Nature Geoscience*, *4*, 723–727, doi:10.1038/NGEO1234.

Årthun, M., Eldevik, T., Viste, E., Drange, H., Furevik, T., Johnson, H. L., & Keenlyside, N. (2017). Skillful prediction of northern climate provided by the ocean. *8*, 15875, doi:10.1038/ncomms15875.

---

## Author Comment (AC2)

**Responses to Reviewer 2 – anonymous**

Bellow you will find our responses to the second review of our manuscript "The role of buoyancy forcing for northern North Atlantic SST variability across multiple time scales", egusphere-2022-959. The reviewer comments are provided in blue while our responses are shown in black, directly following the individual reviewer comments. We acknowledge how addressing the reviewers' comments will help us to improve the manuscript.

Best regards,
Bjørg Risebrobakken, on behalf of all co-authors

This manuscript presents an excellent combination of observational data, paleoclimate data and modeling. The authors tried to provide an explanation for the different scenarios observed in the Pliocene data (from sedimentary records), and, also, to understand the mechanisms that may bring different scenarios in the future. This manuscript illustrates perfectly how the paleoclimate data may be extremely useful to obtain a better understanding of the climate forcings and to predict future scenarios, as stressed in the last IPCC report.

The manuscript is well written and can be easily followed. Figures and tables are clear and illustrate the text clarifying some of the descriptions. I particularly liked figure 11 and Table 2, which summarize the main results of the experiments.

We thank Reviewer 2 for an overall positive review.

I am not an expert in modeling, and, therefore, I cannot evaluate if there are any flaws in the modeling experiments performed. From my perspective the manuscript presents a set of experiments that allowed the authors to assess the role of buoyancy forcing on different scenarios. According to the observations of the Pliocene data, the authors investigated the main drivers of the phase relationships observed between the sea surface temperatures in the North Atlantic, Norwegian Sea and Iceland Sea. It is very interesting that those experiments investigating the role of buoyancy forcing in different scenarios, only provided a robust explanation for 3 of the 4 scenarios. For the scenario in which the Norwegian Sea is out phase the authors propose 2 alternative changes in ocean circulation and/or in water column stratification.

Since the experiments are only evaluating the role of buoyancy forcing, I wonder if there are other mechanisms that could affect the SST during those intervals. If so, could you just mention what other factors may be causing that kind of phasing? I understand that you will probably need a new set of experiments to evaluate those other factors, but it will be nice to acknowledge that maybe buoyancy forcing is not the only forcing.

Yes, the other primary factor impacting the inflow to the Nordic Seas is wind. This will be more clearly stated in the introduction. In addition, we will add more details on why we focus on buoyancy (considered important at longer time scales, based on the knowledge from present day physical oceanography of the region, unless impacted by significant topographical changes). Running more simulations is beyond the scope of this paper, however, we will in the summary section add a sentence on how this may form the base for future studies.

Also, maybe a sketch with the 2 alternative explanations for the Norwegian Sea out of phase will clarify the proposed hypothesis.

We see that it may be useful and will consider the possibility of adding such a figure in the revised manuscript.

Minor comments:
Figure 1. The figure caption indicates a, b and c panels but those are not indicated in the figure, please add the letters in each panel.
The letters a, b and c will be added to the figure.

Line 593: Have not reached
Corrected.

---

## Author Response (AR1)

Dear David Thornally, editor for Climate of the Past,

Below you will find information on the revisions done to address the two reviews of our manuscript "The role of buoyancy forcing for northern North Atlantic SST variability across multiple time scales", egusphere-2022-959. For the revised version we have changed the title to "Buoyancy forcing: A key role for northern North Atlantic SST variability across multiple time scales". The reviewer comments are given in blue while our responses are shown in black, directly following the individual reviewer comments.

In the same manner you also find our response to your request to include more information on the caveats/assumptions/uncertainties regarding the use of alkenones for an SST proxy.

We very much appreciate the thorough and constructive feedback provided through review and how addressing these comments have helped us to significantly improve the manuscript.

Best regards,
Bjørg Risebrobakken
On behald of all co-authors

**Response to the Editor - David Thornalley**
In addition to the reviewer comments, please may I ask that you also include, in your section 2.3, a short section/few sentences where you briefly describe the caveats/assumptions/uncertainties regarding the use of alkenones for an SST proxy. This could reference other review type papers, and comparative proxy studies such as Andersson et al 2009 CoP. I think it is important that the reader is at least made aware that a full interpretation of alkenone SST (like all proxies) needs to consider other environmental (and biological) factors (for example seasonality, changes due to freshwater input and stratification etc). It needs to be flagged to non-proxy readers that some caution is needed and these 'SSTs' may be recording different things at different times. I do not think additional revisions are needed later in the manuscript (discussion etc), of course, if you would like to consider this issue further, in terms of the later interpretation of the data, great, but I don't think that is the focus of your paper here and is thus not necessary.

We appreciate the suggestion and the added value of providing more background information on the use of alkenones as an SST proxy and have integrated such information in Section 2.3.

[Revised paragraph, including background information on alkenones: The $U^K_{37}$' index records the relative abundance of specific lipids (alkenones) synthesized by selected unicellular haptophyte algae living at or near the sea surface (e.g. Marlowe et al., 1984). Through the study of cultures, water samples and surface sediments, it has been shown that the $U^K_{37}$' index changes with temperature (Prahl and Wakeham, 1987; Müller et al., 1998; Conte et al., 2006; Tierney and Tingley, 2018). All records are presented here as previously published (Herbert et al., 2016; Lawrence et al., 2009; Bachem et al., 2016; Bachem et al., 2017), using established age models and the Müller et al. (1998) $U^K_{37}$'-SST calibration. The near-global and linear relationship between surface-sediment UK'37 values and mean annual SSTs (Muller et al., 1998) aligns closely to a culture study (Prahl and Wakeham, 1987), and has been used to calibrate and reconstruct mean annual SSTs. The standard error of estimate using this calibration is ±1.5°C (Müller et al., 1998). As a biological temperature proxy, it is important to consider both the environmental and biological influences over this $U^K_{37}$'-SST relationship. Marked local or regional differences in the timing of alkenone production and flux to the seafloor may impart a seasonal bias to the sedimentary record (e.g. Rosell-Mele and Prahl, 2013). In a recent expansion and Bayesian analysis of the global surface sediment calibration, a stronger correlation to August-October SSTs was identified in the North Atlantic (Tierney and Tingley, 2018) i.e. in the region of our study, which may be supported by overlap between reconstructed SSTs and autumn multi-model means for ODP Site 982 and ODP Site 642 during the KM5c interglacial at 3.205 Ma (McClymont et al., 2020). In the Nordic Seas, low salinity or high sea ice have been linked to elevated production of the C37:4 alkenone (e.g. Bendle and Rosell-Mele, 2004; Wang et al., 2019). However, this alkenone is not included in the $U^K_{37}$' index, and was not recorded at values of concern at ODP Site 642 (Bachem et al., 2017).]

**Responses to Reviewer 1 – Pepijn Bakker**
Review of the manuscript entitled "The role of buoyancy forcing for northern North Atlantic SST variability across multiple time scales"

The authors describe in the introduction that they aim to:
1) "document existing SST anomalies and phase relations in observational data, CMIP6 Shared Socioeconomic Pathways (SSP)126 experiments and Pliocene alkenone SST reconstruction from the North Atlantic, Norwegian and Iceland Seas."
2) "Address why different SST phase relations may emerge and exist across different climate states, time scales and forcing scenarios."
3) "Investigate impacts of changes in buoyancy forcing on the phase relationship between SSTs in the North Atlantic, Norwegian Sea, and Iceland Sea."

Combining information from past, present and future climates, and paleoclimate data, observational data and idealized ocean model simulations; and use all of that that to improve our understanding of spatial patterns of climate change across a wide range of temporal scales is a courageous undertaking with potentially large impacts. However, in my view the authors currently don't fully succeeded to logically combine all this information or derive underlying mechanisms and explanations for the observed patterns.

In the following I will detail all my major, minor and technical comments. I hope my comments will help the authors to improve the manuscript.

We thank Pepijn Bakker for his thorough review and constructive feedback. We appreciate his overall positive view on the overall goal of the paper and his very useful feedback that has helped us improve the manuscript significantly.

**Major comments:**

Underlying hypothesis and research questions:
The starting point of this study seems to be that on time scales longer than the advective time scale of SST anomalies between the three regions under consideration, the SST changes in the three regions should always be in-phase. I suppose this is theoretically true if all else remains equal. If on longer time-scales there are, however, changes in e.g. sea-ice, ocean currents, vertical mixing, boundary conditions etc, then this does not need to be true. Given that there is internal climate and Earth system variability on all possible time-scales plus forced changes, it seems to me that this theoretical starting point will very often not be met. I think this is also what the authors conclude on lines 687-688 "The in-phase situation is the norm relative to the mean background climate state under weak forcing". (Where 'forcing' in this context is to be understood as any change that is external to the coupled system of the northern North Atlantic, the Norwegian Sea and the Iceland Sea; in contrary to forcings external to the climate system in general.) To me it seems that this should in fact be the starting point of this study (for reasons explained above), not a conclusion. This could potentially lead to an overall manuscript structure that is much easier to read and follow. I would like to ask the authors to reflect on this. Then there are all the cases in which the 'forcing is not weak' (again 'forcing' would be any change that is external to the coupled system of the northern North Atlantic, the Norwegian Sea and the Iceland Sea). Given the wide range of time scales that are considered here, it seems to me that there are potentially many, very different, causes of 'strong forcings' and thus reasons why SST's in the three regions under consideration would not vary in concert. The

authors mention a number of them throughout the manuscript (changes in the Bering Strait, Arctic sea-ice export, strength of the different boundary currents in the region, gyre circulation, just to name a few). Is it the aim of this study to pinpoint all these different causes for the time-scales that are considered in this study? Or to identify commonalities between these causes? I think it should be more clearly described what the authors want to learn from the assessment they are presenting.

We thank the reviewer for making us aware of the need to clarify the aim of the study, the hypothesis to be tested and the need to look carefully at the phrasings to avoid misunderstandings. We have tightened the introduction and clarified the aim, questions and hypothesis. We have, however, kept the observational record as our starting point.

[New text in introduction: We question whether the spatially noncoherent SST response seen in the CMIP5 studies is restricted to the high emission scenario, or if spatially noncoherent SST responses may also occur under less extreme atmospheric $CO_2$ concentrations. If it turns out that spatially noncoherent SST anomalies are seen also under less extreme atmospheric $CO_2$ concentrations, the validity of our expectation of spatial coherence may be limited to the observational period. To investigate whether spatially noncoherent SST responses may also occur under less extreme atmospheric $CO_2$ concentrations, we will use CMIP6 Shared Socioeconomic Pathways (SSP)126 experiments and Pliocene alkenone SST reconstruction from the North Atlantic, Norwegian and Iceland Seas. During the Pliocene (5.3 to 2.6 Ma), atmospheric $CO_2$ concentrations were close to 400 ppm in average (De La Vega et al., 2020; Bartoli et al., 2011), comparable to the present (ca. 410 ppm) and the future low emission scenarios such as ssp126 (445 ppm by the end of the century) (Meinshausen et al., 2020; IPCC, 2021). However, it is important to keep in mind is that the Pliocene climate was not forced by an abrupt $CO_2$ increase, as the future scenarios are. Rather, relatively high $CO_2$ values existed for millions of years through the Pliocene. The SST anomaly relations in the North Atlantic, Norwegian and Iceland Seas during the Pliocene may therefore be seen as various equilibrium responses to an atmospheric $CO_2$ content comparable to todays, in contrast to the transient responses given by the CMIP model scenarios. Analysing both the SSP126 experiments and the Pliocene reconstructions therefore allow us to explore potential differences in equilibrium versus transient SST responses to a ca. 400 ppm $CO_2$ forcing.

Furthermore, we address why spatially noncoherent SST relations may emerge and exist across different climate states, time scales and atmospheric $CO_2$ forcing scenarios. As mentioned above, at multidecadal and longer timescales the inflow of Atlantic Water to the Norwegian Sea over the Greenland Scotland Ridge is tightly connected to the density difference between the two basins (Furevik et al., 2007; Smedsrud et al., 2022; Talley et al., 2011), while wind forcing may dominate at shorter time scales (Bringedal et al., 2018). We hypothesise that, for the time scales of interest here, changing the buoyancy may be enough to push the system from spatially coherent to spatially noncoherent SST anomalies. To test this hypothesis, we perform a range of idealized sensitivity experiments using the Massachusetts Institute of Technology general circulation model (MITgcm) to investigate impacts of changes in buoyancy forcing on the SSTs in the North Atlantic, Norwegian Sea, and Iceland Sea. These idealized experiments provide potential physical explanations for the different spatial SST relations see in the investigated region.]

When we say strong forcing, we refer either to the MITgcm experiments and the temperature or freshwater induced buoyancy change, or the high emission scenario runs. We have gone carefully through the text and checked that whenever we use the word forcing we have specified which particular type of forcing we refer to. For example, for the mentioned statement from original lines 687-688 it is now specified that it refers to buoyancy forcing.

We agree that on long time scales, many factors, e.g., changes in continental configuration, may impact the large scale oceanographic features. Keeping that in mind, we see that how we have used the background knowledge from literature (Table 2) might be confusing. We have gone carefully through the text to clarify how we use the information given. While factors mentioned in Table 2 can be seen as forcing in themselves, they are used to infer how buoyancy may have been different under given conditions. E.g., sea ice may work as a force directly, however, more sea ice can also be seen as an indicator for more freshwater in the system, and hence a buoyancy change. We have specified in the start of section 4 how we use the

information provided in Table 2. We have also checked throughout the discussion how we refer to this information and in part restructured the discussion, to avoid further confusion.

[New text, introduction Section 4: From the MITgcm experiments we see the SST anomaly responses to different degrees and causes of buoyancy forcing. In addition, we extract information on physical characteristics associated with the individual experiments and SST anomaly relations (Section 3.4). For the discussion we have searched for information on factors that may have impacted the Nordic Seas buoyancy during the individual Pliocene time periods and the future. For example, information on regional freshwater change is used as an indicator of buoyancy change. In addition, we have searched for information that informs on characteristics somewhat comparable to the physical characteristics extracted from the MITgcm experiments. These parameters include global SSTs, overturning circulation and ventilation of the Nordic Seas, the Atlantic Ocean equator-pole SST gradient and freshwater (Table 2). The content of Table 2 will, together with Table 1, form the basis for the discussion. For each SST anomaly relation identified in the Pliocene reconstructions or the CMIP6 results we will use the information from Table 1 and 2 to see if the SST anomaly change can be linked to a change in buoyancy, and if so, are the associated characteristics comparable to the MITgcm output for a similar SST anomaly relation.
.

In addition, Section 4.1-4.3 have to varying degrees been restructured.]

We have also specified the purpose of providing some of the information in Table 1.

[New text, section 3.4: In addition to identifying the SST anomalies in the MITgcm experiments, information about the density difference over the ridge, mean inflow velocity across the sill, the mean velocity in the boundary current, the net heat transport over the sill and the maximum overturning streamfunction at the sill is extracted for each experiment (Table 1). This information will be used in the discussion, exemplifying oceanographic responses to specific buoyancy changes as seen in the MITgcm experiments.]

The third aim of this manuscript, according to the aims as I list them in the top of this review, is to use the MITgcm to study mechanisms underlying the observed 'phase relationship between SSTs'. Sensitivity studies allow one to test different hypothesis. The experiments presented here only focus on surface buoyancy changes. Why is that? Are there indications that on all the time scales under consideration here the drivers of the SST patterns were buoyancy changes? Is that the hypothesis of this study?

In the introduction we have clarified why we focus on buoyancy, and state the hypothesis that buoyancy may in itself be enough to push the system from spatially coherent to spatially noncoherent SST anomalies. Furthermore, we make it clear that wind also is a key factor impacting the inflow to the Nordic Seas, but since wind is more important at shorter time scales (seasonal and interannual) and we investigate longer time scales (multidecadal and longer), our study put an emphasis on the role of buoyancy.

[Specifications from the introduction:

2nd paragraph: Wind and buoyancy are the two key factors that impact the inflow of Atlantic Water to the Norwegian Sea. Wind forcing is important for the inflow of Atlantic Water across the Greenland Scotland Ridge at seasonal and interannual time scales (Bringedal et al., 2018). However, buoyancy forcing, changing seawater density due to heat (heating/cooling) and/or freshwater (evaporation/precipitation/runoff) fluxes and associated production of dense overflow water that must be compensated, is key at longer time scales (Furevik et al., 2007; Smedsrud et al., 2022; Talley et al., 2011). We will investigate SST anomaly relations in the North Atlantic, Norwegian and Iceland Seas region, at multidecadal and longer time scales, hence, timescales when buoyancy is considered most important. Therefore, our focus is on how northern North Atlantic SST anomalies are impacted by changes in buoyancy.

6th paragraph: Furthermore, we address why spatially noncoherent SST relations may emerge and exist across different climate states, time scales and atmospheric $CO_2$ forcing scenarios. As mentioned above, at multidecadal and longer timescales the inflow of Atlantic Water to the Norwegian Sea over the Greenland Scotland Ridge is tightly connected to the density difference between the two basins (Furevik et al., 2007; Smedsrud et al., 2022; Talley et al., 2011), while wind forcing may dominate at shorter time scales (Bringedal et al., 2018). We hypothesise that, for the time scales of interest here, changing the buoyancy may be enough to push the system from spatially coherent to spatially noncoherent SST anomalies. To test this hypothesis, we perform a range of idealized sensitivity experiments using the Massachusetts Institute of Technology general circulation model (MITgcm) to investigate impacts of changes in buoyancy forcing on the SSTs in the North Atlantic, Norwegian

Sea, and Iceland Sea. These idealized experiments provide potential physical explanations for the different spatial SST relations see in the investigated region.]

Most of the explanations that the authors discuss for the observed patterns (changes in AMOC strength, changes in strength of the East Greenland Current, changes in salinity and/or sea-ice export via the East Greenland Current or a change in the connection between the Northern North Atlantic with the Nordic Sea, from the main connection being with the Norwegian Sea to the main connection being with the Iceland Sea) are not captured in the set of sensitivity experiments so I wonder if these experiments are the right ones to test the mechanisms underlying the observed, reconstructed and simulated patterns.

Thanks for making us aware of the need to clarify the purpose of Table 2 and how we use the information given there. We did not intend to investigate all possible factors that may have an impact. Rather, we wanted to provide background information on factors that may have had an impact on buoyancy over the region at a given time, or factors that may compare to responses seen in the idealised experiments to changes in buoyancy. As mentioned above, we have clarified throughout the manuscript how we use the information from Tables 1 and 2, and hence how we see the information provided as relevant in relation to each other. We also specify that the information is not directly comparable but reflects what can be extracted from available Pliocene datasets. To avoid further confusion, we have also changed the phrasing throughout the manuscript so that we no longer use the word precipitation, but rather freshwater.

The usage of the terms 'in-phase' and 'out-of-phase':
The terms 'in-phase' and 'out-of-phase' for me are linked to temporal behavior. In the context of 'equilibrium' results (Pliocene time-slices, CMIP simulations and MITgcm results) I find these term rather confusing. If for instance a CMIP model shows cooling for the North Atlantic domain for 2070-2100 and warming in the Norwegian Sea they indeed have a different sensitivity to the forcing, but would you call that 'out-of-phase'? For the description of the observational record, I think usage of the terms 'in-phase' and 'out-of-phase' is appropriate. It is perhaps 'only a wording issue', but this for me confuses the whole concept behind the study. It raises questions like whether the authors mean to say that the in-phase Pliocene SST changes are part of an internal mode of variability? (like the described observational changes). Perhaps use words like 'spatially homogeneous versus spatially heterogeneous'? Coherent versus different?

We have changed the phrasing used throughout the manuscript. We are now using phrases like SST anomaly relations, spatially coherent, noncoherent, different from, dissimilar or noncoherent.

Data sources:
Aims number 2 and 3, as described at the beginning of this review, are interesting research questions. However, aim number one is not a research question by itself. Describe more clearly why one would like to document SST anomalies and 'phase-relationships" in this particular combination of data-sets. What is the rationale of combining these data-sets?
We totally agree that to document the anomalies is not a research question. This sentence was included to describe the different steps of the paper and is in that sense just a description of what we do to address what type of SST anomaly patterns that exist in the selected data sets. We have deleted this sentence to avoid further confusion.

Line 101: It seems to me that the easiest starting point to test the main hypothesis of this study (on time scales longer than the advective time scale of SST anomalies between the three regions under consideration, the SST changes in the three regions should always be in-phase) would

be to look at CMIP/PMIP multi-millennial pre-industrial (control) simulations and past-2k simulations. Instead, future runs are used which introduce all kinds of complexities. Please explain the underlying reasoning.

The reason for looking into the future projections is twofold:

1. The spatially incoherent SST pattern, with a cold North Atlantic/warm Nordic Seas was detected in the RCP 8.5 CMIP5 runs but is not fully understood. As specified in the introduction, we wanted to see if this still holds in the lower emission scenario CMIP6 runs. Our results show that this is the case for the most sensitive model.
2. At the end of the century the CO2 concentration of the ssp126 scenario is in the range of the Pliocene CO2 level. Therefore, we wanted to look at the long-term SST pattern of the Pliocene, to see how long-term equilibrium SST patterns compares to the short-term transient response to a comparable atmospheric CO2 concentration.

When the work started out, we did consider looking at the longer historical runs in addition to the future runs, however, at that time the available runs were few and from different models than the ones we had selected for the analysing the projections. Since it would not have been possible to keep consistency between analysed models consistency and to avoid adding even further complexity to an already complex study we decided not to continue in this direction.

Part of our motivation came from how the high emission scenarios seemed to break the expectation of spatially coherent SST anomalies over the North Atlantic, Norwegian and Iceland Seas. The question that came up then was if this signal was restricted to the high emission scenarios or if similar responses could be seen under lower emission scenarios, or in a past comparable climate state.

We see that a logical continuation of this study might be to look at the CMIP/PMIP multi-millennial pre-industrial (control) simulations and past-2k simulations, and we have added this as a future outlook in the summary section of the paper.

[New text, last paragraph of the summery: Building on this study, it would be interesting to do similar analyses of the SST anomalies over the investigated region for the last millennials, to see if the pattern documented for the observational record holds for a longer historical period a preindustrial $CO_2$ level.]

The comparison with the CMIP6 results at the end of the century with the other data sources is difficult because, as the authors say, they show transient climate change, not equilibrium climate change. So why not include for instance experiments forced with a doubling of CO2, that would provide you with an equilibrium response that is more comparable. What is the added value of investigating the transient climate change at the end of this century? Please explain the underlying reasoning.

We appreciate the suggestion to look at a 2*CO2 equilibrium response rather than the CMIP6 scenario experiments. As mentioned above, part of our motivation came from how the high emission scenarios seemed to break the expectation of spatially coherent SST anomalies over the North Atlantic, Norwegian and Iceland Seas, as set by the observational record. The question that came up then was if this signal was restricted to the high emission scenarios or if similar responses could be seen under lower emission scenarios. The next question then is if the response relates to the transient nature of the ssp experiments, or if similar spatially noncoherent SST anomalies also existed during a past climate state with comparable CO2 but in equilibrium. We could not have been able to address these questions by a doubling of CO2 experiment.

Line 101: why only look at Pliocene SST observations while Pliocene model simulations also exist.

When we started out we did consider the possibility to look at PlioMIP experiments as well. However, in the end we decided not to do so, because the PlioMIP experiments are time slice experiments. Analysing those would have given us the SST anomalies relative to pre-industrial, while the SST reconstructions (and the SST anomalies of the other time intervals) are all looked at as the SST anomalies relative to the mean of the respective investigated time series. Hence, the results would not have been comparable to our other results.

Paleoclimate reconstructions: Are the results for proxies local while the model results are large-scale averages? Does this impact the results? Did the authors test how the model results would look like if only data is used from the grid cells in which the proxy data is found?
The proxy records provide information from the individual sites, however, the available sites are from locations considered representative for the area that they represent. If any impact on the results when comparing with the model results that are averaged over a larger area, one may expect the proxy results to show a somewhat larger amplitude of change. We have by purpose avoided looking at the model results from individual grid points. It is not given that the grid points surrounding a specific site provides the best representation of that site, or that the representation would be constant between the different models. We have added a specification on how the domain averaging may impact the amplitude of the signal.
[New text, first paragraph Section 3: The Pliocene reconstructions are site specific but considered to provide a reasonable representation of their respective regions while the observation and model data are regional averages. Somewhat larger amplitudes of the recorded SST anomalies may therefore be expected to be seen in the reconstructions.]

To address this question from the reviewer we have now tested the impact of using single grid points versus averaged domain (using data from the CNRM model). We find that time series from single grid points show similar variability as the averaged time series, but the amplitude might change. Thus, the warm and cold anomalies displayed in Fig. 8b for CNRM would still be the case using single grid points, but the amplitudes might be enhanced or depressed. This is the case for all grid points in the Norwegian Sea and the NE North Atlantic domain, and the case for the grid points in the western Iceland Sea (between 18-15W). Including the grid points in the eastern Iceland Sea (between 15-10W) show some time series with a cooling (instead of a warming) after 2070 (see figures below).

[Figure]

Table 2: The information provided here for the Pliocene comes from many different sources. How good are the age constraints? Are they sufficient to assume that all of the presented climatic indicators happened during a single interval as defined by the authors?

The shortest interval we look at lasts for 200 000 years and we only look at the mean state of the individual intervals. Looking at the mean conditions over such long time intervals, we do consider the age constraints to be of sufficiently good for the use. Pliocene chronologies are mostly constrained by tuning to LR04 and/or using tie points from magnetic reversals. The tuning error is generally considered to be no more than a few thousand years, but may exceed 10 ka prior to 4.3 Ma due to a less certain obliquity variance (Lisiecki and Raymo, 2005). If we had looked at orbital time scales the 10 kyr uncertainty would have been an issue. We have rephrased and added more details about this in Section 2.3. Furthermore, when feeding information to Table 2, we are consistent in treating the data in the same way as we have treated the North Atlantic/Norwegian Sea/Iceland Sea records.

[New text, end last paragraph in section 2.3: Furthermore, focusing on the mean state of longer intervals (the shortest time interval is 200 000 years), rather than point to point comparison, also minimizes the impact of uncertainties from age models. Pliocene chronologies are mostly constrained by tuning to LR04 and/or using tie points from magnetic reversals. The tuning error is generally considered to be no more than a few thousand years, but may exceed 10 ka prior to 4.3 Ma due to less certain obliquity variance (Lisiecki and Raymo, 2005). At the time scales considered here such errors are acceptable.]

MITgcm simulations:
The setup of the MITgcm is focusing on the connection between the North Atlantic, Norwegian Sea and Iceland Sea, but not the connection with the Arctic (sea-ice changes or opening of the Bering Strait). Nonetheless, those mechanisms are found to be important by the authors, questioning whether the setup of the idealized simulations is appropriate for the questions that are asked in this manuscript. Please explain why this approach is taken.

The model set-up is indeed idealized (but with full physical equations), including sea-ice and a possible key gateway missing, However, we have some experience in using the model in similar set-ups before – jointly to disentangle cause-and-effect and, broadly, regionalities (e.g., Våge et al. 2011; Jensen et el. 2018). And this we have found useful.

The approach has in general been spearheaded by Mike Spall, using it in numerous applications/process studies for the North Atlantic, Nordic Seas, and, in some cases, including the Arctic, on the nature of flow, exchanges, and external forcing (buoyancy, wind). Including making regional inference (with care).

As specified in other places and in the revised text, we consider the changes in the Bering Strait as it will eventually change the freshwater balance in the Nordic Seas, and in that sense represent a change in buoyancy. The sea ice changes discussed to be somewhat smaller or larger for the different Pliocene periods refers to sea ice data from the same Iceland Sea site that is used for SST data (ODP Site 907). A specification is added, acknowledging that the idealised setup does not include the Arctic gateways nor the Arctic Sea ice.
[New specification specifically on the lack of Arctic gateways/Arctic sea ice in the MITgcm model setup, Section 4.1: Neither does the idealized mode set up include the Arctic gateways nor the Arctic sea ice cover. Hence, we stress again that the knowledge about changes in the Bering Strait is used to infer changes in the freshwater balance, and hence buoyancy in the Nordic Seas, and similarly, the occurrence of more or less sea ice in the Iceland Sea is used to infer the likelihood that a buoyancy change took place.]

Line 206: why force buoyancy changes on the north of the ridge while the starting point of the manuscript is on the impact of advecting buoyancy anomalies from the region south of the ridge to the region north of the ridge?
We have added a paragraph on why the buoyancy change is induced north of the ridge.
[New text, Section 2.4, 4th paragraph: Buoyancy changes are forced north of the ridge due to the nature of the model set-up. The restoring boundary conditions in the south is also a forcing, both representing the (infinite) source of Atlantic Water and the experiment's energy source (heat and buoyancy input). As the surface forcing is applied north of the ridge, water mass transformation takes place and a consistent ocean circulation is set up, including setting the hydrography of the different regions. The southern boundary energy input and northern surface heat loss balance when the model has reached (quasi-)equilibrium. The northern and southern regions are accordingly equally important for the experiments.]

Figures 1 and 5: Are the geographical regions used for the Iceland Sea and the Norwegian Sea in figure 1 (and so for the observational results and the CMIP results) comparable to the definition of the Iceland Sea and the Norwegian Sea in figure 5 (and so for the MITgcm results)? In the latter it seems really a comparison between a boundary current (Norwegian Sea) and the ocean interior (Iceland Sea), but is that a good representation of the other observational, modeling and proxy-based reconstruction results?
The reviewer is right that the Norwegian and Iceland Seas domains are slightly different when looking at the output from the MITgcm experiments compared to the CMIP output. The MITgcm setup is idealized, therefore it is not possible to set exactly the same domains and expect comparability to the observations and CIMIP results. Thanks for making us aware that we have missed adding information about this and the reasoning behind. In the revised manuscript we have specified in Section 2.4 that the domains are not exactly the same, and added reasoning for why this is the case. We have also added information on why the domains are set to the boundary current and the interior Iceland Sea. Site 642 in the Norwegian Seas is under direct influence of the Norwegian Atlantic Current represented by the eastern boundary current, while Site 907 is from the interior part of the Iceland basin. The domain used for observations/CMIP6 is also set to target the same areas (section 2.1). The caption for Figure 1 has been corrected and a specification is added to the caption for Figure 5.

[New text, Section 2.4, 7th paragraph: When presenting MITgcm results, the North Atlantic domain is defined as the North Atlantic restoring region (Fig. 5c), set to be 6°C for all experiments (Fig. 4a). The Norwegian Sea domain is defined as a box in the eastern boundary current region, while the Iceland Sea domain is represented by the interior ocean north of the ridge (Fig. 5c). The definition of these domains, as the domains used for the observations and CMIP6 results (Section 2.1; Fig. 1), are directed by the location of the Pliocene sites, representing the Norwegian boundary current and interior Iceland Sea. Since the MITgcm setup is idealized it is not possible to set the exact same domains as used for the observations and CMIP6 results. However, within the limitations set by the individual data sources, all information extracted from the reconstructions, observations, CMIP6 and MITgcm experiments represents the Norwegian Sea boundary current and the interior Iceland Sea. Regional inference can therefore be made from the MITgcm experiments.

New text from caption to Figure 5: The domains are set for the results to be comparable to the reconstructions, observations and CMIP6 results. Since the MITgcm is idealized the domains can, however, not be identical to the domains defined in Section 2.1.]

Minor comments:
Line 66: I understand what is being said here, but that is only because of the lines that follow. Please try to describe more clearly what is meant here. 'Relationships' between what? And why does the continuous northward transport of heat imply this?
We see that this could be unclear and have rephrased the paragraph.
[New text: Heat is continuously transported northwards from the North Atlantic towards the Arctic. Due to the continuous northward transport, it is expected that a warm North Atlantic will entail warm SSTs both in the Norwegian and Iceland Seas. Alternatively, if it is cold in the North Atlantic, the Norwegian and Iceland Seas are also expected to be cold. It takes 3-4 years for Sea Surface Temperature (SST)/heat anomalies to travel from the North Atlantic through the Norwegian Sea (Holliday et al., 2008). Therefore, spatially incoherent SST anomalies between the seas may exist at interannual-to-decadal time scales. This feature has been documented in observations and Earth System Models (Årthun and Eldevik, 2016; Årthun et al., 2017). Beyond decadal time scales, however, this propagation-driven lag should in theory no longer be of importance, and the default expectation is of a spatially coherent SST relationship between the North Atlantic, the Norwegian Sea and the Iceland Sea, in line with observations (Årthun and Eldevik, 2016; Årthun et al., 2017).]

Lines 208-215: I find the description of the modeling setup very confusing. You start by describing three different reference experiments. Why do we need three different reference experiments and how are they different? And then at some point in this text you continue describing the actual perturbation experiments. Why does the second set of experiments (G1-P1, G1-P2 etc) use G1 and not G0 as SAT forcing? Refer to table 1 at this point as the different experiments are nicely summarized in there, or include a new table to show this information?
We have restructured the text and added information on why we set up three reference experiments. A reference to Table 1 is added.
[Updated text Section 2.4 (3rd and 5th paragraph): Since one of the key drivers for inflow of Atlantic Water to the Norwegian Sea is buoyancy forcing and production of dense overflow water that must be compensated (Furevik et al., 2007), we change the SAT (G) and the freshwater (in form of precipitation) (P) north of the ridge to study the impact of buoyancy forcing on the relationships between SSTs in the North Atlantic, Norwegian Sea, and Iceland Sea. The SAT and freshwater are changed as shown in Fig. 4 and Table 1. Note that the SAT over the restoring region is the same for all experiments. The idealized model is run for 30 years, to near steady-state, and we present results from the last 5 years of the runs which are compared to the results from the relevant reference experiment. We are therefore not studying transient changes, but differences between equilibrium states.

We want to investigate the responses to changes in buoyancy caused by either a SAT or a freshwater change. In addition, we want to see if the initial state of the ocean impacts the response to a SAT change, specifically testing if the response differs if we start out from a fresher Nordic Seas. Therefore, we define three reference experiments, REF-1 (G0 and P1), REF-2 (G1 and P1) and REF-3 (G0 and P3) (Table 1). REF-1 is set up to investigate the oceanographic responses to a gradually decreasing SAT gradient between the North Atlantic and the Nordic Seas, under constant freshwater forcing, by increasing the SATs over the Nordic Seas. For the REF-2 experiments, the buoyancy is changed by gradually increasing the freshwater over the Nordic Seas while SAT is kept constant. The REF-3 experiments are similar to the REF-1 experiments in the sense that SAT over the Nordic Seas are increased, however, the initial state of the Nordic Seas is fresher for REF-3 than for REF-1. Hence the REF-3 experiments

are set up to see how the initial state of the ocean may impact the responses to increased SAT over the Nordic Seas.]

REF-2 was set up using G1 rather than G0 because G0 provided smaller responses.

**Line 304: 'enhances the SST in the Iceland Sea', what does that mean? Please clarify.**

We have rephrased this part of the text to clarify the content. In the model version with a lower horizontal resolution in the atmosphere the SST of the Iceland Sea is higher at the end of the century than in the model version with a higher horizontal resolution in the atmosphere. Such an effect is not seen in the Norwegian Sea or the North Atlantic.

[New text, last paragraph Section 3.2: Furthermore, we find that lowering the horizontal resolution in the atmosphere entails higher SSTs in the Iceland Sea at the end of the century relative to the results from the model version with a higher horizontal resolution in the atmosphere. A lowering horizontal resolution in the atmosphere does not, however, have a clear effect on the SSTs of the Norwegian Sea nor the North Atlantic (Fig. 7a).]

**Line 354: why would you call an insignificant response 'in phase' and why not simply an insignificant response? I guess there could also be a significant in phase response?**

From the MITgcm experiments, all in phase SST changes are less than 2*std of the reference experiment. We have changed the phrasing used in line 354 and a couple of other places. In Figure 10 (now 9) we have changed the colour used in the inserts for any case where the MITgcm SST anomaly response is less than 2std.

[New phrasings:

Section 3.4, 2nd paragraph: With increasing freshwater over the Nordic Seas, the SST pattern shifts from spatial coherence (P2 and P3) to the Iceland Sea SST anomaly being different from the North Atlantic and the Norwegian Sea (P4) to the North Atlantic SST anomaly being different from the Norwegian and Iceland Seas (P5) (Fig. 9b).

Section 4.1, 3rd paragraph: In the idealized experiments a weak cold spatially coherent SST response (less than 2*std of REF-2) is found with weak to intermediate freshwater driven change (P3) in buoyancy

Section 4.1, 3rd paragraph: Less freshwater was available in the region during the Pliocene intervals with warm spatially coherent SST anomalies, also in line with the idealized experiments where a weak warm spatially coherent SST response (less than 2*std of REF-2) is seen for a weak salinity increase (Table 1; Fig. 9b).

Figure 9 caption: Lighter colours are used if the SST anomalies is less than 2*std of the relevant reference experiment.]

**Lines 515-517: The opening of the Bering Strait could indeed play a role the Pliocene cooling over the whole North Atlantic, but what about other potential drivers? Higher CO2 levels for instance, how well are those constrained?**

Increasing or decreasing atmospheric CO2 concentrations will have an impact on the atmospheric temperatures over the region. For example one might imagine that with arctic amplification a potentially increased atmospheric CO2 concentration may entail a larger temperature change over the Nordic Seas than over the North Atlantic, hence, comparable to idealized experiments relative to REF1 and REF3 where we change the buoyancy forcing by increasing the SAT over the Nordic Seas.

The CO2 level is unfortunately not well enough constrained before ca. 3.5 Ma to evaluate the role of CO2 changes might have been for the individual intervals.

**Lines 541-349: Many GCM experiments exist in which sea-ice melt increases and/or NADW is reduced. What do such experiments show? Is that in line with the findings described here for these specific Pliocene intervals? Please discuss.**

We have added reference to GCM relevant GCM experiments and expanded the discussion here.

[New text in Section 4.2, 2nd paragraph: While not directly comparable, the overall reduction of the overturning at the sill in the idealized experiment seems consistent with a reduced %NADW for this period in the Pliocene (Table 2). The presence of seasonal sea ice, or sea ice transported to the Iceland Sea, suggests a slight freshening, and thereby strengthened stratification in the Iceland Sea. Such a change in stratification may lead to reduced dense water formation and may thereby weaken the NADW formation. The existence of sea ice in the Iceland Sea during the warm Pliocene suggests somewhat enhanced seasonal contrasts. Tests of different model sensitivities and freshwater forcing scenarios, using LOVECLIM, have shown the AMOC could be more sensitive to freshwater forcing under warm interglacial climate states with large seasonal contrasts for warm climate states with a weaker seasonal contrast (Blaschek et al., 2015). However, Blaschek et al. (2015) saw such an AMOC response only when freshwater reached the Labrador Sea convection site, not for future scenarios where less freshwater/sea ice is likely to impact the Labrador Sea. For parts of this Pliocene interval, 3.43-3.23 Ma, some seasonal sea ice existed in the Labrador Sea (Clotten, 2017). Hence, the results of Blaschek et al. (2015) shows the same direction of AMOC change as seen in the idealized experiments and from Pliocene data for the warm Iceland Sea cold North Atlantic and Norwegian Sea SST anomaly relation driven by a buoyancy change due to weak atmosphere warming. A weakened ocean circulation may again bring less heat and salt into the Norwegian Sea and by continuation the Iceland Sea, further strengthening the stratification.]

Lines 576-577: I don't agree that the time-scales are comparable. The changes in the observational record play out on decadal time-scales while in the future runs, we are comparing roughly the period 2000-2050 with the period 2050-2100. Indeed figure 8 shows that on top of this 'long-term' variability, there is also decadal variability similar to the observational record for which the relationships between SST's in the three different regions are again different from what is described in this section (and perhaps more similar to the variability in the observational record). Please clarify.

The sentence the reviewer refers to has been deleted.

The SST anomalies of the observational record take place over multiple decades, hence multi decadal time scales. For the future we focus on the conditions at the end of the century, 2068-2098 relative to the mean of the future runs (2023-2098), hence, a 30 year long period, or a multi decadal signal.

For the end of the century (2068-2098), CNRM-ESM2-1 SSP126 and NorESM2-MM ssp585 show the discussed phase relation, with a cold anomaly in the North Atlantic and a warm anomaly in the Norwegian and Iceland Seas. For MPI-ESM1-2-LR SSP126 you can not really distinguish the different members from each other. Therefore, we did not include the insert there (however, the same cold anomaly in the North Atlantic and a warm anomaly in the Norwegian and Iceland Seas is seen in the member presented in Figure 8b). NorESM2-MM is the least sensitive of the models, and for that model there are still spatially coherent SST anomalies at the end of the century for the SSP126 experiment, showing how the sensitivity of the model plays a role.

We have rephrased the first sentence of Section 4.3 so that it is clear that what we discuss is the signal seen at the end of the century (2023-2098).

[New start Section 4.3, 1st paragraph: A positive SST change (warming) in the Norwegian and Iceland Seas corresponding with a small (close to zero) negative SST change (cooling) in the North Atlantic is seen at the end of the century (2068-2098) in CMIP6 future projections, depending on the scenario used (Fig. 7).]

Line 609: I don't quite understand this part. Do the authors mean that there is an indirect link between the radiative forcing and the 'out-of-phase' relationship between the North Atlantic and the Nordic Seas? And if so, what kind of link would this be?

Thanks for making us aware of this unclear statement. The sentence has been rephrased.

[New text, last sentence Section 4.3: The $CO_2$ forcing of SSP126 (445 ppm by 2100) is comparable to the high end of the Pliocene $CO_2$ range (427 ppm) (Meinshausen et al., 2020; De La Vega et al., 2020), suggesting that the

amplitude of the SST anomalies of the North Atlantic and the Nordic Seas is set by the atmospheric $CO_2$ level and that the response time may be as short as within a century.]

**Lines 632-633: Increased influence of the East Greenland Current can explain a cooling of the Iceland Sea, but how does it explain the corresponding cooling of the North Atlantic?**

We have added information on this in Section 4.4 where this conceptual explanation is put forward.

[Revised paragraph in Section 4.4, 5th paragraph: In contrast, the existence of a warm anomaly in the Norwegian Sea corresponding with cold anomalies in the subpolar North Atlantic and in the Iceland Sea could in theory result from a strengthened or expanded East Greenland Current, increasing the fraction of cold polar water reaching the Iceland Sea (Rudels et al., 2005) and the North Atlantic (Dickson et al., 1988). Admittedly, the effect would need to be quite substantial to affect the North Atlantic proper (this is nevertheless what is implied by the common attribution of the hydrographic impact of the "Great Salinity Anomaly"; (Dickson et al. 1988). In general, the state of the subpolar North Atlantic tends to relate more to the larger-scale forcing or subtropical–subpolar gyre features (e.g., Hátún et al. 2005; Reverdin 2010), which in this case would be aligned to leave the region anomalously cold.]

**Lines 635-639: Wouldn't you still need to weaken advection between the Norwegian and Iceland Seas? Otherwise wouldn't warm North Atlantic water enter the Norwegian Sea via the Iceland Sea and still result in a warming in both regions?**

We have clarified this is Section 4.4 where this conceptual explanation is put forward.

[Revised paragraph in Section 4.4, 4th paragraph: Following this conceptual framework, a cold SST anomaly in the Norwegian Sea corresponding with a warm North Atlantic and Iceland Sea SST anomaly may result from a weakened Norwegian Atlantic Current compensated by a strong Irminger Current. The dominant advective influence of the Nordic Seas is the eastern inflow via the Norwegian Atlantic Current. However, and even if anomalies tend to persist throughout the Nordic Seas advective loop, the water that at the end of this loop travels south via the Greenland and Iceland seas will qualitatively be cold, as the water flowing out of Nordic Seas through the Danmark Strait (e.g., Eldevik et al. 2009; Eldevik and Nilsen 2013). The Irminger Current, on the other hand, is a warm inflow directly influencing the Iceland Sea, where it largely overturns locally to overflow through the Denmark Strait where it entered (e.g., Våge et al. 2011). A stronger Irminger Current inflow can thus be expected to leave an anomalous warm signature in the Iceland Sea – simply more warm water brought directly into the mix – independent of the anomalous state of the Norwegian Sea and the Norwegian Atlantic Current. Based on existing information we cannot verify if this was the case, or not, between 4.73 and 4.93 Ma.]

**Lines 688-690: what is meant here with a 'weakened ocean circulation'? Please clarify.**

This was not a correct statement; thanks for making us aware of the mistake here. The statement has been corrected.

[New text, Section 5, 5th paragraph: The idealized experiments suggest that the situation where the Iceland Sea SST anomaly differs from the Norwegian Sea and the North Atlantic likely reflects a response to a weak increase in atmospheric temperatures under constant freshwater forcing. However, the existing data and data coverage are not good enough to verify this statement.]

**Lines 705-706: Isn't this what one would expect? That the amplitude of SST changes depends on the radiative forcing?**

We agree with the reviewer that it may be expected that the amplitude of SST change depends on the radiative forcing. However we still find it interesting that we see the same amplitude at the end of the century, after quite a short time of increasing CO2, as for Pliocene anomalies representative of hundreds of thousands of years. It emphasises how quickly change may take place. We have added a specification.

[New text, Section 5, 7th paragraph: However, since both the Pliocene reconstructions and the future change occur under atmospheric $CO_2$ concentrations around 400 ppm or higher, these results suggest that the amplitudes of SST anomalies in the Nordic Seas depend on the radiative forcing. Furthermore, the results suggest that within the timeframe of the future scenarios SST anomalies can reach amplitudes comparable to the SST anomaly amplitudes seen for Pliocene periods lasting hundreds of thousands of years and in equilibrium with the $CO_2$ forcing, emphasising how rapidly the earth system can react to increasing atmospheric $CO_2$ concentrations.]

**Figure 1: How are the domains depicted in figure 1 determined and what is their influence on the presented results?**

The domains are selected to be representative of the three sites where we have Pliocene data. We do consider that SST averaged over the domains to better represent the variability than a single grid point (see reply to comment above). More details on the selection of the domains have been added to section 2.1.

[New text Section 2.1: The data is averaged over three box domains, as shown in Fig. 1, to represent the three sites in the Pliocene reconstructions (Section 2.3). The same domains are used in the CMIP6 model analysis (Section 2.2). We consider that SST averaged over the domains better represents the variability than a single grid point. The domains are chosen as follows: to represent the site in the NE North Atlantic, we use a domain covering the northeastern part of the Subpolar North Atlantic (49-57°N, 35-14°W); to represent the site along the NwAC, we use a box over the eastern Norwegian Sea (62.5-73°N, 0-16°E), and finally, to represent the site in the Iceland Sea, we use a box covering the major part of the Iceland Sea (66-72°N, 18-10°W).]

**Figure 7: clarify in the figure caption that these are all multi-decadal variations. What does it mean that only the bandpass filtered data are significant and the running mean data are not?**

Figure 7 was included to show the anti-phase relations at interannual to decadal scales. We see that it may be confusing the way it was included and the figure has been deleted from the revised manuscript and the text has been edited accordingly.

[New text, Section 3.1, first paragraph: On multidecadal time scales the annual SSTanomaly, as seen in the HadlSST dataset, varies between -0.8 and +0.8°C (Fig. 6). As described in the introduction, the spatially noncoherent SST anomaly signal seen on shorter time scales should in theory no longer be of importance on multidecadal time scales, and we see a spatially coherent SST anomaly relationship between the North Atlantic, the Norwegian Sea and the Iceland Sea (Fig. 6).]

**Figure 10: The SST response seems very small in many cases. How significant are these results and how does the magnitude of these responses compare to the magnitude found in the observational results, CMIP models and proxy-based reconstructions?**

In Table 1 it is shown which of the MITgcm results are significant. The magnitude of the anomalies for each case is also defined in Table 1. In addition, we have now added information about significance in the figure caption of Figure 10. All "red" experiments see significant anomalies in both the Norwegian and Iceland Seas. The "blue" experiments see a significant change in the Norwegian Sea, but not in the Iceland Sea. The "grey" experiments see no significant change in either the Norwegian or Iceland Seas.

The significant SST anomalies seen from the MITgcm experiments are in the range of 0.17 to 0.69°C for the Norwegian Sea and 0.51 to 2.42°C for the Iceland Sea; the insignificant MITgcm responses are 0.0-0.1 and 0.0 to 0.08, respectively. The mean anomalies as seen in HadlSSTs, CMIP6 models and Pliocene reconstructions are within the same range for the Iceland Sea. In the Norwegian Sea the mean anomaly exceeds the MITgcm range in the most sensitive CMIP6 model (CNRM-ESM2-1 ssp126), and in four of the six Pliocene cases.

More details have been given in the text.

[New caption for Figure 10: SST anomalies seen in MITgcm idealized experiments. a) SST anomalies relative to REF- 1, where precipitation is kept constant at P1 while SAT is gradually increased (G1-G4). b) SST anomalies relative to REF-2, where SAT is kept constant at G1 while precipitation is gradually increased (P2-P5). c) SST anomalies relative to REF-3, where precipitation is kept constant at P3 while SAT is increased (G1 and G2). Surrounding blue boxes represent an Iceland Sea SST anomaly spatially incoherent with the North Atlantic and the Norwegian Sea anomalies. The SST anomaly seen in the Iceland Sea exceeds 2*std of the relevant reference experiment. Surrounding red boxes represent a North Atlantic SST anomaly spatially incoherent with the Norwegian and Iceland Seas anomalies. The SST anomalies seen in the Norwegian and Iceland Seas exceeds 2*std of the relevant reference experiment. Surrounding grey boxes represent spatially coherent SST anomalies between the three regions (not significant responses; none of the SST anomalies exceeds 2*std of the relevant reference experiment). Conceptualized representation of the resulting phase relations is shown by the map inserts, where blue (red) boxes represent cold (warm) SST anomalies for the North Atlantic, Norwegian Sea and Iceland Sea. The base for the map inserts is made with GeoMapApp (www.geomapapp.org) / CC BY / CC BY (Ryan et al., 2009)).]

[New details added to section 3.4:
- End paragraph 7: The amplitude of the MITgcm Icealand Sea SST anomaly is in this case (G1 relative to REF-1 and G1 relative to REF-3) ca. 1/3 of the Iceland Sea SST anomaly reconstructed for the Pliocene (3.43 to 3.23 Ma) when the SST anomaly of the Iceland Sea differs from the North Atlantic and Norwegian Sea SST anomalies.
- End paragraph 8: The amplitude of the SST anomalies in the Iceland and Norwegian Sea as seen in CNRM-ESM2-1 ssp126 and NorESM-MM ssp 528 at the end of the century is within the range of the respective MITgcm SST anomalies.
- End paragraph 9: As for the experiments where we increase the atmospheric temperature over the Nordic Seas, the Iceland Sea SST anomaly in the respective Pliocene case is larger than the MITgcm SST anomaly response to a medium freshwater change (P4 to REF-2). The MITgcm response to the largest freshwater forcing (P5 to REF-2) is at the lower end of the SST anomalies seen for the CMIP6 models at the end of the century.]

And on line 241 you mention that the North Atlantic is set to constant, so why are anomalies simulated for that region in figure 10? Clarify in a little more detail how the experimental forcing is defined.

Line 241 was not formulated very well, and we have changed it to "The North Atlantic is restored to constant temperatures". In addition, we have added some specifications to the method-section.

The North Atlantic is restored toward constant temperatures with a given restoring strength and a time scale of 1 month. This means that the region is not necessarily constant in temperature, only restored towards it. The restoring will dampen the potential temperature changes, and that is why we do not see a significant temperature change in that region.

[New specification Section 2.4, 8th paragraph: The North Atlantic is restored to constant temperatures. Temperatures are not necessarily constant in the restoring region but restores towards constant temperatures. The restoring will dampen the potential temperature change, and therefore no significant temperature change is ever seen for this region.]

Table 2: Why is information from the observational period, one of the three main periods discussed in this manuscript, not included in this table? Please explain.

Related climate change for warm and cold periods in the observational period have been documented previously in studies about AMV. However, in this study, information from the instrumental period is primarily used to set the expectation, and confirm that the result previously shown by others (e.g. Årthun et al., 2017) for specific stations along the pathway of Atlantic Water transport also is valid when looking at the domains that we focus on in this study. We have added information on this in Section 2.1.

[New text, start Section 2.1: The expectation of spatially coherent SST anomalies between the North Atlantic and the Norwegian and Iceland Seas is rooted in the observational period and investigations of SST anomalies at specific stations along the pathway of the North Atlantic Current (Årthun and Eldevik, 2016; Årthun et al., 2017). A comparable analysis of the observational record is done here to confirm if the expectation of spatial coherence holds when looking at averages over larger domains encompassing the North Atlantic, Norwegian and Iceland Seas.]

Technical comments:
Figure 1: include letters a,b and c in the figure. Also I don't see the described blue and red boxes in figure 1. Please clarify.

The letters a, b and c have been added to the figure. The reference to blue and red boxes was added to inform on how the boxes/defined domains are expressed in following figures. We see that the phrasing may be confusing and have deleted this part of the sentence.

Figure 6: What do the grey and white vertical bands mean in this figure? The 'multi-decadal' periods over which the anomalies in the inserted maps are calculated

We have added a specification to the figure caption.

[Specification added to Figure 6 caption: The grey bars highlight the periods with positive spatially coherent SST anomalies.]

In general I find figure 6 not a very clear representation of the different in-phase and anti-phase relationships on interannual-to-decadal and on multi-decadal time-scales as is described on lines 258- 263.

The focus of Fig. 6 is to show in-phase relationships on multidecadal time scales, not the anti-phase relation at shother time scales. Hence, you should not expect to see the anti-phase reflation. As mentioned above, figure 7 was included for the purpose of showing the anti-phase at interannual to decadal time scales, but to avoid confusion we have deleted figure 7 from the revised manuscript and rephrased the text.

Figure 8B: why no inserted maps for the middle two models?

Inserted maps are not shown for the two middle models due to a weak signal and/or that there was not really possible to say anything from between the different model members. We have added an insert to the NorESM-MM SSP126 model (and a specification in the text on how this results emphasise the importance of the model sensitivity for the results). We have also added a specification in the figure caption on why no insert map is shown for the second model, MPI-ESM1-2-LR SSP126.

[Specification added to Figure 8 caption: The individual members of MPI-ESM1-2-LR SSP126 cannot be distinguished from each other. Therefore we have not added a map insert for the member presented. .

Section 4.3 specification about the NorESM-MM SSP126 results shown in Figure 8: NorESM2-MM is the least sensitive of the models, and for that model there are still spatially coherent SST anomalies at the end of the century for the SSP126 experiment, showing that the sensitivity of the model plays a role.]

Table 1: The experiment name that is given in this table (exp) seems different from what is used in the main text and in figure 10. Consider changing for clarity.

Thanks for making us aware that this may be a source of confusion. We have revised how we present the information about the exp names in Table 1.

**Reviewer 2 – anonymous**

This manuscript presents an excellent combination of observational data, paleoclimate data and modeling. The authors tried to provide an explanation for the different scenarios observed in the Pliocene data (from sedimentary records), and, also, to understand the mechanisms that may bring different scenarios in the future. This manuscript illustrates perfectly how the paleoclimate data may be extremely useful to obtain a better understanding of the climate forcings and to predict future scenarios, as stressed in the last IPCC report.

The manuscript is well written and can be easily followed. Figures and tables are clear and illustrate the text clarifying some of the descriptions. I particularly liked figure 11 and Table 2, which summarize the main results of the experiments.

We thank Reviewer 2 for an overall positive review that have helped us improve the manuscript.

I am not an expert in modeling, and, therefore, I cannot evaluate if there are any flaws in the modeling experiments performed. From my perspective the manuscript presents a set of experiments that allowed the authors to assess the role of buoyancy forcing on different scenarios. According to the observations of the Pliocene data, the authors investigated the main drivers of the phase relationships observed between the sea surface temperatures in the North Atlantic, Norwegian Sea and Iceland Sea. It is very interesting that those experiments

investigating the role of buoyancy forcing in different scenarios, only provided a robust explanation for 3 of the 4 scenarios. For the scenario in which the Norwegian Sea is out phase the authors propose 2 alternative changes in ocean circulation and/or in water column stratification.

Since the experiments are only evaluating the role of buoyancy forcing, I wonder if there are other mechanisms that could affect the SST during those intervals. If so, could you just mention what other factors may be causing that kind of phasing? I understand that you will probably need a new set of experiments to evaluate those other factors, but it will be nice to acknowledge that maybe buoyancy forcing is not the only forcing.

Yes, the other primary factor impacting the inflow to the Nordic Seas is wind. This is now more clearly stated in the introduction. We have also added details on why we focus on buoyancy. Running more simulations is beyond the scope of this paper, however, we have in the summary section added a sentence on how this may be a future step to follow up on.

[New text from the introduction, 2$^{nd}$ paragraph: Wind and buoyancy are the two key factors that impact the inflow of Atlantic Water to the Norwegian Sea. Wind forcing is important for the inflow of Atlantic Water across the Greenland Scotland Ridge at seasonal and interannual time scales (Bringedal et al., 2018). However, buoyancy forcing, changing seawater density due to heat (heating/cooling) and/or freshwater (evaporation/precipitation/runoff) fluxes and associated production of dense overflow water that must be compensated, is key at longer time scales (Furevik et al., 2007; Smedsrud et al., 2022; Talley et al., 2011). We will investigate SST anomaly relations in the North Atlantic, Norwegian and Iceland Seas region, at multidecadal and longer time scales, hence, timescales when buoyancy is considered most important. Therefore, our focus is on how northern North Atlantic SST anomalies are impacted by changes in buoyancy.]

[New paragraph on future avenues, last paragraph Section 5: Building on this study, it would be interesting to do similar analyses of the SST anomalies over the investigated region for the last millennia, to see if the pattern documented for the observational record holds for a longer historical periods at a preindustrial CO$_2$ level. Furthermore, it would be of interest to do a series of sensitivity studies testing the effects of changing the winds over the region, since wind is the other main factor affecting the inflow to the Norwegian Sea. However, the fact that we can explain most of the observed spatial SST patterns emphasises that buoyancy plays a key role for northern North Atlantic SST variability across the multiple time scales investigated.]

Also, maybe a sketch with the 2 alternative explanations for the Norwegian Sea out of phase will clarify the proposed hypothesis.

We have optioned not to add a new figure here, but have instead added more information in the text to clarify conceptual explanations, also addressing comments from reviewer 1.

[Revised paragraph in Section 4.4, 4$^{th}$ paragraph: Following this conceptual framework, a cold SST anomaly in the Norwegian Sea corresponding with a warm North Atlantic and Iceland Sea SST anomaly may result from a weakened Norwegian Atlantic Current compensated by a strong Irminger Current. The dominant advective influence of the Nordic Seas is the eastern inflow via the Norwegian Atlantic Current. However, and even if anomalies tend to persist throughout the Nordic Seas advective loop, the water that at the end of this loop travels south via the Greenland and Iceland seas will qualitatively be cold, as the water flowing out of Nordic Seas through the Danmark Strait (e.g., Eldevik et al. 2009; Eldevik and Nilsen 2013). The Irminger Current, on the other hand, is a warm inflow directly influencing the Iceland Sea, where it largely overturns locally to overflow through the Denmark Strait where it entered (e.g., Våge et al. 2011). A stronger Irminger Current inflow can thus be expected to leave an anomalous warm signature in the Iceland Sea – simply more warm water brought directly into the mix – independent of the anomalous state of the Norwegian Sea and the Norwegian Atlantic Current. Based on existing information we cannot verify if this was the case, or not, between 4.73 and 4.93 Ma.

Revised paragraph in Section 4.4, 5$^{th}$ paragraph: In contrast, the existence of a warm anomaly in the Norwegian Sea corresponding with cold anomalies in the subpolar North Atlantic and in the Iceland Sea could in theory result from a strengthened or expanded East Greenland Current, increasing the fraction of cold polar water reaching the Iceland Sea (Rudels et al., 2005) and the North Atlantic (Dickson et al., 1988). Admittedly, the effect would need to be quite substantial to affect the North Atlantic proper (this is nevertheless what is implied by the common attribution of the hydrographic impact of the "Great Salinity Anomaly"; (Dickson et al. 1988). In general, the state of the subpolar North Atlantic tends to relate more to the larger-scale forcing or subtropical–subpolar gyre features

(e.g., Hátún et al. 2005; Reverdin 2010), which in this case would be aligned to leave the region anomalously cold.]

Minor comments:
Figure 1. The figure caption indicates a, b and c panels but those are not indicated in the figure, please add the letters in each panel.
The letters a, b and c have been added to the figure.

Line 593: Have not reached
Corrected.

---

## Referee Report (RR1)

Review 2 of the manuscript entitled "The role of buoyancy forcing for northern North Atlantic SST variability across multiple time scales"

The manuscript is much improved. I do, however, have some further minor comments and technical comments.

**Minor comments:**

In section 2.4 the model setup and results for the reference experiments is described. I think this is a good and clear setup. A bit confusing that figure 5 and accompanying text, which seems to be part of section 2.4, does in fact already present results from a non-reference experiment. Consider making a clear seperation between the methods and the results.

Figures 6 & 7: How sensitive are these results (spatially coherent versus spatially non-coherent) on the definition of the multi-decadal analysis period? It seems to me that in figure 6 you can also pick periods of the same duration for which you will find spatially non-coherent results? Similarly for figure 7, if you would not be forced to use 2068-2098, but could shift this frame back and forth in time by several decades, would you find very different results? Some discussion on this sensitivity (and perhaps thus the effect of decadal variability on multi-decadal averages) would make the manuscript stronger.

For the analysis of the pliocene data, observations and CMIP results, the North Atlantic region is defined as a region roughly between Ireland and the southern tip of Greenland. This would roughly correspond to the norther half of the subpolar gyre. In the results of the MITgcm, the same region is however defined as the very southern part of the model domain. This seems to correspond to the southern side of the subtropical gyre. Looking at figure 9, it seems that this definition could make a lot of difference. What is the rationale of using this definition of the North Atlantic region in the MITgcm and how does it impact the results?

Lines 625-626: Doesn't this line imply that the changes are related to transient changes rather than equilibrium states?

**Technical comments:**
Line 260: 'Has'
Line 303: How is sigma defined? Calculated after annually averaging the model SSTs?
Line 304: 'restores'
Line 326: 'SST anomaly'
Figure 9: in the panel of G1,P4-REF-2, I'm wondering if the scematic showing cooling in the North Atlantic, warming over the iceland basin and no change over the Norwegian current, is indeed correct. Looking at the anomaly map these signals are not apparent.
Line 624: 'iare'
Line 794: mention the period that the observational data covers, not just the starting year.
Code/Data availability: it seems that this section is not yet complete.

---

## Author Response (AR2)

Dear David Thornally,

Here you will find our responses to the second round of review of our manuscript egusphere-2022-959. We appreciate the work you and the reviewers have put in, helping us to improve the manuscript.

The reviewer comments are shown in black, and our responses are shown in blue.

We have also corrected some minor inconsistences and typos found when while reading through the text again.

Best regards,
Bjørg Risebrobakken
On behalf of all co-authors

Review 2 of the manuscript entitled "The role of buoyancy forcing for northern North Atlantic SST variability across multiple time scales"

The manuscript is much improved. I do, however, have some further minor comments and technical comments.

We thank Pepijn Bakker for his second review of our manuscript. We appreciate his positive feedback and are glad that he sees significant improvements of the manuscript following the first round of review.

**Minor comments:**
In section 2.4 the model setup and results for the reference experiments is described. I think this is a good and clear setup. A bit confusing that figure 5 and accompanying text, which seems to be part of section 2.4, does in fact already present results from a non-reference experiment. Consider making a clear seperation between the methods and the results.
We see that it may cause confusions that Figure 5 included some of the results. We have made a new Fig 5 which only shows a reference experiment and the defined domains, to better separate the methods and results. The text/figure caption has been corrected accordingly.

Figures 6 & 7: How sensitive are these results (spatially coherent versus spatially non-coherent) on the definition of the multi-decadal analysis period? It seems to me that in figure 6 you can also pick periods of the same duration for which you will find spatially non-coherent results?
Information on the robustness has been added to section 3.1.

The spatially coherent SST anomaly relationship between the North Atlantic, the Norwegian Sea and the Iceland Sea (Figure 6) are robust. Correlations between the time series from the three regions are positive and significant. This relationship holds for different filtering of the time series (running mean with a 5-year window, 10-year window, and 15-year window). The correlation increases, the more the time series are filtered.

Please see the attached figures that show:
1) The filtered anomaly time series for different filters: the more smoothing, the clearer is the positive correlation between the time series.
2) The correlation between the anomaly time series (description below each plot tells which time series and which filter that has been applied).

[Figure]

| | |
|---|---|
| hadisst_anomalies_timeseries_filt_10yr_det rended | hadisst_anomalies_timeseries_filt_15yr_det rended |
| hadisst_corr_filt_5yr_aIS_bNA | hadisst_corr_filt_5yr_aNS_bNA |
| hadisst_corr_filt_10yr_aIS_bNA | hadisst_corr_filt_10yr_aNS_bNA |

[Figure]

| hadisst_corr_filt_15yr_aIS_bNA | hadisst_corr_filt_15yr_aNS_bNA |

New text section 3.1: The spatially coherent SST anomaly relationship between the North Atlantic, the Norwegian Sea and the Iceland Sea is robust, showing positive and significant correlations between the detrended time series (both the Norwegian Sea and the Iceland Sea is correlated with the North Atlantic). The relationship also holds for different filtering of the time series, i.e., running mean with a 5-year window, 10-year window, and 15-year window (not shown).

Similarly for figure 7, if you would not be forced to use 2068-2098, but could shift this frame back and forth in time by several decades, would you find very different results? Some discussion on this sensitivity (and perhaps thus the effect of decadal variability on multi-decadal averages) would make the manuscript stronger.

In contrast to the results in Figure 6, the results in the last three decades in Figure 7 are not robust across the whole time period. Thus, the spatially incoherent SST anomaly relationship, found in CNRM-ESM2-1 SSP126 and and NorESM2-MM SSP585 in the last three decades, is sensitive to the time period chosen. This means that choosing another time period, e.g., 2038-2068, would not give the same results. Correlations between the time series from the three regions are non-significant, and this is found for different filtering of the time series (running mean with a 5-year window, 10-year window, and 15-year window). The same results are found for all four experiments, except for NorESM2-MM SSP126 (indication of a positive correlation between the North Atlantic and the Iceland Sea) and NorESM2-MM SSP585 (indication of a positive correlation between all regions only if time series are detrended, but then the spatially incoherent SST anomaly relationship is lost). A new paragraph addressing this has been added to Section 3.2.

New paragraph, Section 3.2: The spatially incoherent SST anomaly relationship found in CNRM-ESM2-1 SSP126 and NorESM2-MM SSP585 in the last three decades (2068-2098) (Fig. 7) is sensitive to the chosen period and are not robust over the whole projected period (2023-2098). Correlations are overall non-significant between the time series from the three regions. Positive correlations are indicated for NorESM2-MM SSP585 (both the Norwegian Sea and the Iceland Sea is correlated with the North Atlantic), but only if time series are detrended. In the latter case, the spatially incoherent SST anomaly relationship in the last three decades is not found in NorESM2-MM SSP585.

For the analysis of the pliocene data, observations and CMIP results, the North Atlantic region is defined as a region roughly between Ireland and the southern tip of Greenland. This would roughly correspond to the norther half of the subpolar gyre. In the results of the MITgcm, the same region is however defined as the very southern part of the model domain. This seems to

The reviewer is correct from a geographical point of view. To better motivate and explain the rationale for the MITgcm experiments including and how we set the three domains we have rephrased how this is explained in Section 2.4.

Updated text replacing the last two paragraphs in the previous version of the manuscript:

For the MITgcm, the Norwegian Sea domain is defined as a box in the eastern boundary current region, while the Iceland Sea domain is represented by the interior ocean north of the ridge (Fig. 5). The definition of these domains, as the domains used for the observations and CMIP6 results (Section 2.1; Fig. 1), are directed by the location of the Pliocene sites, representing the Norwegian boundary current and interior Iceland Sea.

The definition of the North Atlantic domain is somewhat different; it is for simplicity defined as the North Atlantic restoring region (Fig. 5), restored to 6°C for all experiments (Fig. 4a). With this restoring, the state of the North Atlantic source water eventually becoming the inflow to the Nordic Seas, is essentially known. Also, the model North Atlantic is much less directly impacted by the prescribed changes in buoyancy forcings than the Nordic Seas (Fig. 4; and also less impacted in consequence, as evident in Fig. 9). Directly related, it is the relative temperature (density) difference of the model ocean that constrains the flow and thus the results. (The nonlinearity of the equation of state will only be in effect for large excursions in the absolute values of the restoring temperature and salinity between the different experiments.)

In short, the MITgcm experiments assess the state of the Nordic Seas, and including that of the Norwegian and Iceland seas, relative to that of the North Atlantic. The summary of experiments in Table 1 reflects this relative perspective.

We identify spatially coherent/noncoherent SST anomaly relationships between the North Atlantic, Norwegian and Iceland Seas by comparing the temperature of the sensitivity experiment with the relevant reference experiment. Change in a region is classified as an SST anomaly when the temperature change between two experiments exceeds $2\sigma(SST_{reference\_experiment})$, with $\sigma$ calculated after temporally averaging the model SSTs. The North Atlantic is restored to constant temperatures as mentioned above. Even if it deviates (slightly) within what is allowed for by the restoring (see Section 2.4), it remains essentially constant and this non-anomalous. Thus, as also alluded to above, change in SST anomaly relationship between the three regions exists if there is a temperature change in either the Norwegian Sea or the Iceland Sea, or both, larger than $2\sigma$.

Lines 625-626: Doesn't this line imply that the changes are related to transient changes rather than equilibrium states?

No, it was not the intention to give the impression that the differences in amplitude for the SST anomalies between decadal scale observations and hundreds of thousands of years is related to transient change rather than equilibrium. To avoid confusion, we have deleted this sentence.

**Technical comments:**
Line 260: 'Has'
Corrected

Line 303: How is sigma defined? Calculated after annually averaging the model SSTs?
Yes, sigma is calculated after averaging the model SSTs. A specification is added to the text.

Line 304: 'restores'
Corrected

Line 326: 'SST anomaly'
Corrected

Figure 9: in the panel of G1,P4-REF-2, I'm wondering if the scematic showing cooling in the North Atlantic, warming over the iceland basin and no change over the Norwegian current, is indeed correct. Looking at the anomaly map these signals are not apparent.

Thanks for noticing this mistake, the schematic used here was not the correct one. The figure has been updated. We changed the inserts so nonsignificant changes is shown consistently in the same manner, realizing that there were a small room for misunderstanding between the figure (as it was) and the text.

Line 624: 'iare'
Corrected

Line 794: mention the period that the observational data covers, not just the starting year.
Corrected

Code/Data availability: it seems that this section is not yet complete.
You are right that part of the information was missing here. This has been corrected.
Updated text on data and code availability:
All reconstructions are previously published. Information on availability is given in the Supplement.
The observation-based SST data set from the Met Office Hadley Centre can be accessed from the following link: https://www.metoffice.gov.uk/hadobs/hadisst/.
Data from the current generation of global climate model simulations is available through the most recent Coupled Model Intercomparison Project Phase 6 (CMIP6), and can be accessed from Earth System Grid Federation (https://esgf-node.llnl.gov/search/cmip6/).
The MITgcm source code is an open sourced, freely available model code (https://mitgcm.readthedocs.io/en/latest/getting_started/getting_started.html). Due to the size of the model results, simulation data is avaialabl upon request.